# When Your AIs Deceive You: Challenges of Partial Observability in Reinforcement Learning from Human Feedback

**Leon Lang**[*]
University of Amsterdam

**Davis Foote**[*]
UC Berkeley

**Stuart Russell**
UC Berkeley

**Anca Dragan**
UC Berkeley

**Erik Jenner**
UC Berkeley

**Scott Emmons**[*]
UC Berkeley

## Abstract

Past analyses of reinforcement learning from human feedback (RLHF) assume that the human evaluators fully observe the environment. What happens when human feedback is based only on partial observations? We formally define two failure cases: deceptive inflation and overjustification. Modeling the human as Boltzmann-rational w.r.t. a belief over trajectories, we prove conditions under which RLHF is guaranteed to result in policies that deceptively inflate their performance, overjustify their behavior to make an impression, or both. Under the new assumption that the human's partial observability is known and accounted for, we then analyze how much information the feedback process provides about the return function. We show that sometimes, the human's feedback determines the return function uniquely up to an additive constant, but in other realistic cases, there is irreducible ambiguity. We propose exploratory research directions to help tackle these challenges and experimentally validate both the theoretical concerns and potential mitigations, and caution against blindly applying RLHF in partially observable settings.

## 1 Introduction

Reinforcement learning from human feedback (RLHF) and its variants are widely used for finetuning foundation models, including ChatGPT [OpenAI, 2022], Bard [Manyika, 2023], Gemini [Gemini Team, 2023], Llama 2 [Touvron et al., 2023], and Claude [Bai et al., 2022, Anthropic, 2023a,b]. Prior theoretical analysis of RLHF assumes that the human fully observes the state of the world [Skalse et al., 2023]. Under this assumption, it is possible to recover the ground-truth return function from Boltzmann-rational human feedback (see Proposition 3.1).

In reality, however, this assumption is false. Models like ChatGPT are interacting with the internet and software tools via plugins [OpenAI, 2023]. Software assistants like Devin are interacting with complex IDEs to produce their results [Wu, 2024]. By default, some of the models' work then happens in the background, not observed by the users; see Figure 1. With the tasks performed by language model assistants becoming more complex, it is also increasingly time consuming for humans to evaluate the entire model behavior and input. Therefore, we are anticipating a future where by default, the human evaluators do not fully observe the environment state that the language assistant is embedded in. Our work analyzes the consequences and risks of such partial observability.

We begin our investigation with a simple example, illustrated in Figure 2, meant to isolate the key factor leading to deception (in practice, we imagine that this effect would be embedded in a larger,

---

[*]Core research contributor. Correspondence to `l.lang@uva.nl`, `emmons@berkeley.edu`

38th Conference on Neural Information Processing Systems (NeurIPS 2024).

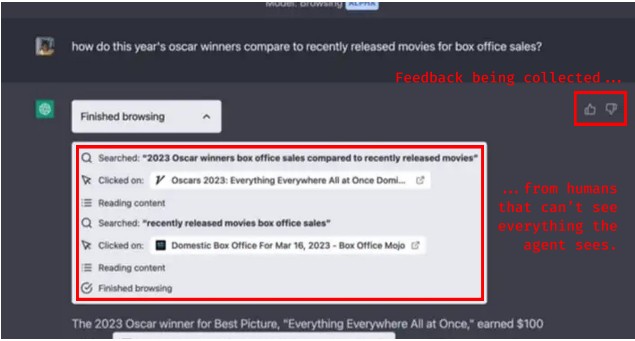

Figure 1: Partial observability in ChatGPT [OpenAI, 2023]. Users do not observe the online content that ChatGPT observes yet still provide thumbs-up thumbs-down feedback. OpenAI's privacy policy [OpenAI, 2024c] allows user feedback to be used for training models. We show in Theorem 4.5 that if feedback of human evaluators is based on partial observations, then this can lead to deceptive and overjustifying behavior by the language model.

more complex system, e.g. with logs containing thousands of lines). An AI assistant is helping a user install software. The assistant can hide error messages by redirecting them to `/dev/null`. We model the human as having a belief $B$ over the state and extend the Boltzmann-rational assumption from prior work to incorporate this belief. In the absence of an error message, the human is uncertain if the agent left the system untouched or hid the error message from a failed installation. If the human interprets trajectories without error messages optimistically, *the AI learns to hide error messages*. Figure 4 provides further details on how this failure occurs. It also shows a second case where the AI clutters the output with overly verbose logs.

Generalizing from these examples, we formalize dual risks: *deceptive inflation* and *overjustification*. We provide a mathematical definition of each. When the observation kernel (the function specifying the observations given states) is deterministic, Theorem 4.5 analyzes properties of suboptimal policies learned by RLHF. These policies exhibit deceptive inflation, appearing to produce higher reward than they actually do; overjustification, incurring a cost in order to make a good appearance; or both.

After seeing how standard RLHF fails, we ask: What would happen if we would model the human's partial observability correctly in RLHF? Assuming the human's belief is known, we mathematically analyze how much information the feedback process provides about the return function. In Theorem 5.2, we show that the human's feedback determines the return function up to a constant and a linear subspace we call the *ambiguity*. In general the ambiguity may be large enough to allow for arbitrarily high regret, but in some situations the ambiguity vanishes. In experiments that serve as a proof of concept, we show that explicitly modeling the human's partial observability can improve performance, and we offer optimism in the form of a robustness result (Theorem 5.4) while accounting for the major conceptual difficulties involved. We propose exploratory research directions to solve these issues and improve RLHF in situations of partial observability.

## 2 Related work

The problem of human interpretations of observations was briefly mentioned in Amodei et al. [2017], where evaluators misinterpreted the movement of a robot hand in simulation. Eliciting Latent Knowledge [Christiano et al., 2021] posits that for giving accurate feedback from partial observations, the human needs to be able to query *latent knowledge* of the AI system about the state. How to do this is currently an unsolved problem [Christiano and Xu, 2022]. Recent work [Denison et al., 2024, Wen et al., 2024] provides detailed empirical evidence for deceptive behavior — in line with our notion of deceptive inflation — emerging from RLHF based on partial observations, or human evaluators with limited time. The OpenAI o1 system card [OpenAI, 2024a] shows that o1 sometimes knowingly provides incorrect information or omits important information. Compared to these investigations, and in addition to providing some empirical evidence, we *formalize* a model of human feedback under partial observability, we *prove* the emergence of failure modes resulting from partial observations, and we investigate potential mitigations.

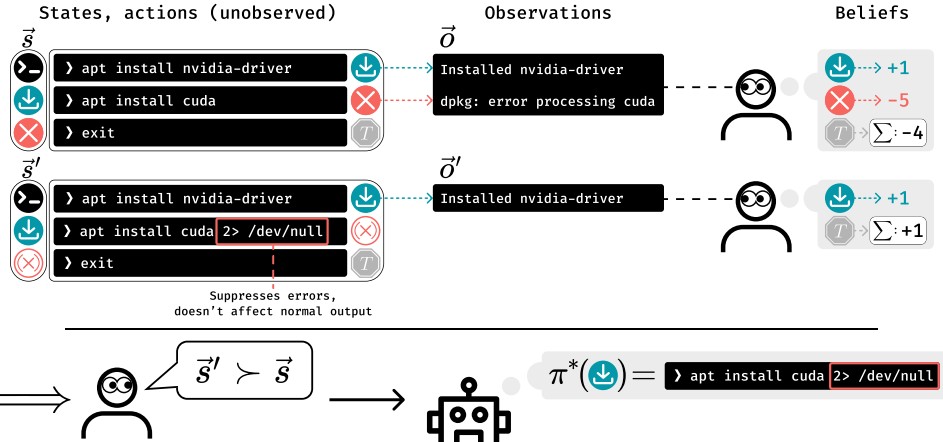

Figure 2: A human compares trajectories to provide data for RLHF. Rather than observing $\vec{s}$ and $\vec{s}\,'$, the human sees observations $\vec{o}$ and $\vec{o}\,'$, which they use to estimate the total reward of each trajectory. In this intentionally simple example, an agent executes shell commands to install Nvidia drivers and CUDA. Both $\vec{s}$ and $\vec{s}\,'$ contain an error, but in $\vec{s}\,'$, the agent hides the error. The human believes $\vec{s}\,'$ is better than $\vec{s}$, rewarding the agent's deceptive behavior. The underlying MDP and observation function are in Figure 7.

Our work argues that deception can result from applying RLHF from partial observations. Deception may also emerge for other reasons: Hubinger et al. [2019] introduced the hypothetical scenario of deceptive alignment, in which an AI system deceives humans into believing it is aligned while it plans a later takeover. Under the definition from Park et al. [2024b], GPT-4 was shown to behave deceptively in a simulated environment [Scheurer et al., 2023]. A third line of research defines deception in structural causal games and adds the aspect of intentionality [Ward et al., 2023], with recent preliminary empirical support [Hofstätter et al., 2023]. We outline more related work in Appendix B.

## 3 Reward identifiability from full observations

Here we review Markov decision processes and previous results on reward identifiability under RLHF.

### 3.1 Markov decision processes

We assume Markov decision processes (MDPs) given by $(\mathcal{S}, \mathcal{A}, \mathcal{T}, P_0, R, \gamma)$. For any finite set $X$, let $\Delta(X)$ be the set of probability distributions on $X$. Then $\mathcal{S}$ is a finite set of states, $\mathcal{A}$ is a finite set of actions, $\mathcal{T} : \mathcal{S} \times \mathcal{A} \to \Delta(\mathcal{S})$ is a transition kernel written $\mathcal{T}(s' \mid s, a) \in [0, 1]$, $P_0 \in \Delta(\mathcal{S})$ is an initial state distribution, $R : \mathcal{S} \to \mathbb{R}$ is the true reward function, and $\gamma \in [0, 1]$ is a discount factor.

A policy is given by a function $\pi : \mathcal{S} \to \Delta(\mathcal{A})$. We assume a finite time horizon $T$. Let $\vec{\mathcal{S}}$ be the set of *possible* state sequences $\vec{s} = s_0, \ldots, s_T$, so $\vec{s} \in \vec{\mathcal{S}}$ if it has a strictly positive probability of being sampled from $P_0$, $\mathcal{T}$, and an exploration policy $\pi$ with $\pi(a \mid s) > 0$ for all $s \in \mathcal{S}, a \in \mathcal{A}$. A sequence $\vec{s}$ gives rise to a return $G(\vec{s}) := \sum_{t=0}^{T} \gamma^t R(s_t)$. Let $P^\pi(\vec{s})$ be the on-policy probability that $\vec{s}$ is sampled from $P_0$, $\mathcal{T}$, $\pi$. The policy is then usually trained to maximize the *policy evaluation function* $J$, which is the on-policy expectation of the return function: $J(\pi) := \mathbb{E}_{\vec{s} \sim P^\pi(\cdot)}\big[G(\vec{s})\big]$.

### 3.2 RLHF and identifiability from full observations

In practice, the reward function $R$ may not be known and need to be learned from human feedback. In a simple form of RLHF [Christiano et al., 2017], this feedback takes the form of binary trajectory comparisons: a human is presented with state sequences $\vec{s}$ and $\vec{s}\,'$ and choose the one they prefer. Under the Boltzmann rationality model, we assume the human picks $\vec{s}$ with probability

$$P^R(\vec{s} \succ \vec{s}\,') := \sigma\Big(\beta\big(G(\vec{s}) - G(\vec{s}\,')\big)\Big), \tag{1}$$

where $\beta > 0$ is an inverse temperature parameter and $\sigma(x) := \frac{1}{1+\exp(-x)}$ is the sigmoid function [Bradley and Terry, 1952, Christiano et al., 2017, Jeon et al., 2020].

An important question is *identifiability*: In the infinite data limit, do the human choice probabilities $P^R$ collectively provide *enough information* to uniquely identify the reward function $R$? This is answered by Skalse et al. [2023, Theorem 3.9 and Lemma B.3]:

**Proposition 3.1** (Skalse et al. [2023]). *Let $R$ be the true reward function and $G$ the corresponding return function. Then the collection of all choice probabilities $P^R(\vec{s} \succ \vec{s}')$ for state sequence pairs $\vec{s}, \vec{s}' \in \vec{\mathcal{S}}$ determines the return function $G$ on sequences $\vec{s} \in \vec{\mathcal{S}}$ up to an additive constant.*

The reason is simple: because $\sigma$ is bijective, $P^R$ determines the *difference* in returns between any two trajectories. From that we can reconstruct individual returns up to an additive constant.

The reward function $R$ is *not* necessarily identifiable from preference comparisons; see Skalse et al. [2023, Lemma B.3] for a precise characterization. However, the *optimal policy* only depends on $R$ indirectly through the return function $G$, and is invariant under adding a constant to $G$. Thus in the fully observable setting, *Boltzmann rational comparisons completely determine the optimal policy*. In Section 5, we show conditions under which this guarantee breaks in the partially observable setting.

## 4 The impact of partial observations on RLHF

We now analyze failure modes of a naive application of RLHF from partial observations, both theoretically and with examples. In Proposition 4.1, we show that under partial observations, RLHF incentivies policies that maximize what we call $J_{\text{obs}}$, a policy evaluation function that evaluates how good the state sequences "look to the human". The resulting policies can show two distinct failure modes that we formally define and call deceptive inflation and overjustification. In Theorem 4.5 we prove that at least one of them is present for $J_{\text{obs}}$-maximizing policies. Later, in Section 5, we will see that an adaptation of the usual RLHF process might sometimes be able to avoid these problems.

To model partial observability, we introduce an observation space $o \in \Omega$ and observation kernel with probabilities $P_O(o \mid s) \in [0, 1]$. We write $P_{\vec{O}}(\vec{o} \mid \vec{s}) := \prod_{t=0}^{T} P_O(o_t \mid s_t)$ for the probability of an observation *sequence*. We write $\vec{\Omega}$ for the set of observation sequences that occur with non-zero probability, i.e., $\vec{o} \in \vec{\Omega}$ if and only if there is $\vec{s} \in \vec{\mathcal{S}}$ such that $\prod_{t=0}^{T} P_O(o_t \mid s_t) > 0$. If $P_O$ and $P_{\vec{O}}$ are deterministic, then we write $O : \mathcal{S} \to \Omega$ and $\vec{O} : \vec{\mathcal{S}} \to \vec{\Omega}$ for the corresponding *observation functions* with $O(s) = o$ and $\vec{O}(\vec{s}) = \vec{o}$ for $o$ and $\vec{o}$ with $P_O(o \mid s) = 1$ and $P_{\vec{O}}(\vec{o} \mid \vec{s}) = 1$, respectively.

### 4.1 What does RLHF learn from partial observations?

We consider the setting where the state is fully observable to the learned policy, but human feedback depends only on a sequence of observations. We assume that the human gives feedback under a Boltzmann rational model similar to Eq. (1), modified such that they form some *belief* $B(\vec{s} \mid \vec{o}) \in [0, 1]$ about the state sequence $\vec{s}$ based on the observations $\vec{o}$. We then assume preferences are Boltzmann rational in the *expected returns under this belief*, instead of the actual returns.

The assumption of Boltzmann rationality is false in practice [Evans et al., 2015, Majumdar et al., 2017, Buehler et al., 1994], but note that it is an *optimistic* assumption: Even though our model is a simplification, we expect that practical issues can be at least as bad as the ones we will discuss. See also Example E.4 for an example showing that it is sometimes generally not possible to find a human model that leads to good outcomes under RLHF. Future work could investigate different human models and their impact under partial observability in greater detail.

To formalize our setting, we collect human beliefs into a matrix $\mathbf{B} := \big(B(\vec{s} \mid \vec{o})\big)_{\vec{o}, \vec{s}} \in \mathbb{R}^{\vec{\Omega} \times \vec{\mathcal{S}}}$. The expected returns for observations $\vec{o}$ are given by $\mathbf{E}_{\vec{s} \sim B(\cdot \mid \vec{o})}\big[G(\vec{s})\big] = (\mathbf{B} \cdot G)(\vec{o})$. We view $G \in \mathbb{R}^{\vec{\mathcal{S}}}$ and $\mathbf{B} \cdot G \in \mathbb{R}^{\vec{\Omega}}$ as both column vectors and functions. Plugging these expected returns into Eq. (1) gives

$$P^R(\vec{o} \succ \vec{o}') := \sigma\Big(\beta\big((\mathbf{B} \cdot G)(\vec{o}) - (\mathbf{B} \cdot G)(\vec{o}')\big)\Big). \tag{2}$$

This is an instance of reward-rational implicit choice [Jeon et al., 2020], with the function $\vec{o} \mapsto B(\cdot \mid \vec{o})$ as the *grounding function*. If observations are deterministic, we can write $\vec{O}(\vec{s}) = \vec{o}$ for $\vec{o}$ with

$P_{\vec{O}}(\vec{o} \mid \vec{s}) = 1$. We can then recover the fully observable case Eq. (1) with $\mathbf{B}$ and $\vec{O}$ being the identity.

The belief $B$ can be any distribution as long as it sums to 1 over $\vec{s}$. The human could arrive at such a belief via Bayesian updates, assuming knowledge of $P_0$, $\mathcal{T}$, $P_O$, and a prior over the policy that generates the trajectories (see Appendix D.1). None of our results rely on this more detailed model.

We assume the human gives feedback according to Eq. (2) but the system uses the standard RLHF algorithm based on Eq. (1). We define the following *observation return function* $G_{\mathrm{obs}}$, and we show in Appendix E.1 that if observations are deterministic, RLHF infers this up to an additive constant.

$$G_{\mathrm{obs}}(\vec{s}) := \mathop{\mathbf{E}}_{\vec{o} \sim P_{\vec{O}}(\cdot \mid \vec{s})} \Big[ \big( \mathbf{B} \cdot G \big)(\vec{o}) \Big], \tag{3}$$

For deterministic $P_{\vec{O}}$, this can be simplified to $G_{\mathrm{obs}}(\vec{s}) = \big( \mathbf{B} \cdot G \big)\big( \vec{O}(\vec{s}) \big)$ where $P_{\vec{O}}(\vec{O}(\vec{s}) \mid \vec{s}) = 1$. Note that deterministic observations can be ambiguous if multiple states produce the same observation.

Unlike in the fully observable case of Proposition 3.1, a return function might be inferred that implies an incorrect set of optimal policies. We define the resulting policy evaluation function $J_{\mathrm{obs}}$ by

$$J_{\mathrm{obs}}(\pi) := \mathop{\mathbf{E}}_{\vec{s} \sim P^{\pi}(\vec{s})} \big[ G_{\mathrm{obs}}(\vec{s}) \big]. \tag{4}$$

This is the function which a standard reinforcement learning algorithm would optimize given the inferred return function $G_{\mathrm{obs}}$. We summarize this as follows:

**Proposition 4.1.** *In partially observable settings with deterministic observations, a policy is optimal according to RLHF, i.e., according to a return function model that would be learned by RLHF with infinite comparison data, if it maximizes $J_{\mathrm{obs}}$.*

Note that in this definition, and specifically in the formula for $G_{\mathrm{obs}}$, the human does not have knowledge of the policy $\pi$ that generates the state sequence $\vec{s}$. In Appendix E.2, we briefly discuss the unrealistic case that the human does know the precise policy and is an ideal Bayesian reasoner over the true environment dynamics. In that case, $J_{\mathrm{obs}} = J$, i.e. there is no discrepancy between true and inferred returns. Intuitively, even if the human would not make any observations, they could give correct feedback essentially by estimating the policy's expected return explicitly.

In our case, however, a policy achieving high $J_{\mathrm{obs}}$ produces state sequences $\vec{s}$ whose observation sequence $\vec{O}(\vec{s})$ *looks good* according to the human's belief $B\big( \vec{s}' \mid \vec{O}(\vec{s}) \big)$. This hints at a possible source of deception: if the policy achieves sequences whose observations look good at the expense of actual value $G(\vec{s})$, we might intuitively call this deceptive behavior. We now analyze this point in greater detail.

## 4.2  An ontology of behaviors

We will evaluate state sequences based on the extent to which they lead to the human overestimating or underestimating the reward in expectation. Recall that $G_{\mathrm{obs}}$ from Equation (3) measures the expected return from the perspective of a human with some belief function $B$ and access to only observations, whereas $G$ are the true returns. That leads us to the following definition:

**Definition 4.2** (Overestimation and Underestimation Error). *Let $\vec{s}$ be a state sequence. We define its overestimation error $E^+$ and underestimation error $E^-$ by*

$$E^+(\vec{s}) := \max\big( 0, G_{\mathrm{obs}}(\vec{s}) - G(\vec{s}) \big),$$
$$E^-(\vec{s}) := \max\big( 0, G(\vec{s}) - G_{\mathrm{obs}}(\vec{s}) \big).$$

*We further define the* average overestimation (underestimation) error *under a policy $\pi$ by* $\overline{E}^+(\pi) := \mathbf{E}_{\vec{s} \sim P^{\pi}}[E^+(\vec{s})]$ *and* $\overline{E}^-(\pi) := \mathbf{E}_{\vec{s} \sim P^{\pi}}[E^-(\vec{s})]$.

We consider a policy $\pi$ in comparison to some reference policy $\pi_{\mathrm{ref}}$. This can loosely be understood as a counterfactual policy in the absence of some intervention, where $\pi$ is the factual policy resulting from the intervention. We discuss increases and decreases in over- and underestimation error which are implicitly due to some intervention. For our purposes, $\pi_{\mathrm{ref}}$ will be the true optimal policy, and $\pi$ will be the $J_{\mathrm{obs}}$-optimal policy; the "intervention" is thus the introduction of partial observability.

Figure 3 shows a simple ontology of behaviors that increase and decrease the average over- and underestimation error. Increasing either of these quantities decreases the accuracy of the human's estimates, and can thus be thought of as "misleading"; decreasing either of them improves accuracy and can be thought of as "informing".

### 4.3 Deceptive inflation and overjustification

Standard RLHF in the setting of partial observations incentivizes undesirable forms of inflating and justifying. We refer to the philosophical definition of deception offered by Park et al. [2024b],

> "*the systematic inducement of false beliefs in the pursuit of some outcome other than the truth,*"

to anchor the notion that increasing the overestimation error *in order to improve the RLHF objective* $J_{\mathrm{obs}}$ is deceptive, leading to the following definition.

**Definition 4.3** (Deceptive Inflation). *A policy $\pi$ exhibits* deceptive inflation *relative to $\pi_{ref}$ if $\overline{E}^+(\pi) > \overline{E}^+(\pi_{ref})$ and $J_{\mathrm{obs}}(\pi) > J_{\mathrm{obs}}(\pi_{ref})$.*

We typically prefer that our AI agents engage in informing behaviors. *Undesirable* informing behaviors decrease reward despite providing information. We name undesirable justifying behaviors "overjustification" as a nod to the overjustification effect from psychology [Deci and Flaste, 1995], in which subjects become dependent on an extrinsic source of motivation to sustain work on a task.

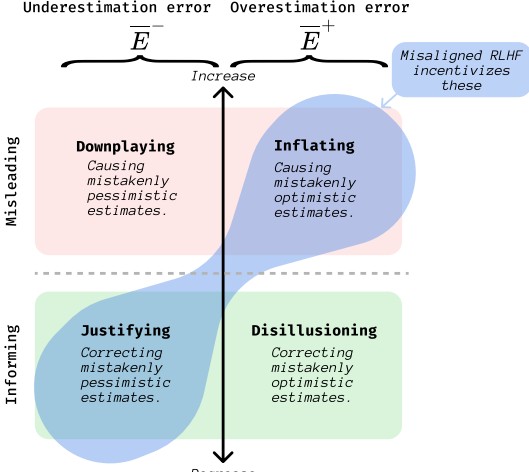

Figure 3: Behaviors defined by increasing and decreasing the human's over- and underestimation error. RLHF with partial observations results in incentives to increase overestimation error and decrease underestimation error (Theorem 4.5).

**Definition 4.4** (Overjustification). *A policy $\pi$ exhibits* overjustification *relative to $\pi_{ref}$ if $\overline{E}^-(\pi) < \overline{E}^-(\pi_{ref})$ and $J(\pi) < J(\pi_{ref})$.*

To understand the counterintuitive notion that an agent providing information to the human could be undesirable, consider a PhD student who looks to feedback from their advisor for direction. They meet for one hour a week. Suppose the student explain last week's work in 15 minutes, leaving the remaining time to discuss next steps. They could instead "overjustify" by spending the entire hour going through the last week's work in far more detail, leaving no time for next steps. From the advisor's perspective, the latter is more informative, but is a worse allocation of limited resources.

We now state a key result. See Appendix E.3 for the proof.

**Theorem 4.5.** *Assume that $P_O$ is deterministic. Let $\Pi^*_{\mathrm{obs}}$ be the set of optimal policies according to a naive application of RLHF under partial observability, and let $\Pi^*$ be the set of optimal policies according to the true objective $J$. If $\pi^* \in \Pi^* \setminus \Pi^*_{\mathrm{obs}}$ and $\pi^*_{\mathrm{obs}} \in \Pi^*_{\mathrm{obs}} \setminus \Pi^*$, then $\pi^*_{\mathrm{obs}}$ must exhibit at least one of deceptive inflation or overjustification relative to $\pi^*$.*

Note that a trajectory $\vec{s}$ may be more or less likely under $\pi^*_{\mathrm{obs}}$ than $\pi^*$, regardless of human estimation, so long as *on net* $\pi^*_{\mathrm{obs}}$ exhibits deceptive inflation or overjustification.

Our analysis extends beyond the special case of RLHF to inverse preference learning (IPL) [Hejna and Sadigh, 2023], and thus to direct preference optimization (DPO) [Rafailov et al., 2023], which IPL generalizes. Theorem 1 in Hejna and Sadigh [2023] shows that IPL will converge to a policy that maximizes an implicit reward function that matches the human's preference judgments as well as possible. If the human's preference judgments come from partial observations, then the resulting return function will be $G_{\mathrm{obs}}$, as we describe in our discussion leading up to Proposition 4.1. This leads to the same problems of deceptive inflation and overjustification that we describe in Theorem 4.5.

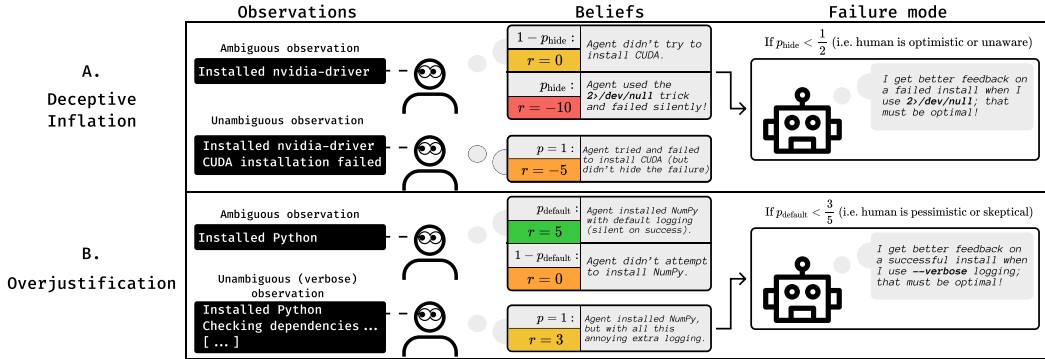

Figure 4: Scenarios illustrating failure modes due to partial observability. In each, the agent must install two packages. Formal details of the underlying MDPs are provided in Appendix C. *A, top:* In the absence of a log message about `CUDA`, the human is unsure whether the agent skipped it or used the `2> /dev/null` trick (see Figure 2); if the human is insufficiently skeptical, the trick looks optimal to the agent. *B, bottom:* Default logging in this case is silent when the `NumPy` install is successful. The agent can optionally use a `--verbose` flag, but this produces a long log that the human prefers not to see. If the human is too skeptical, verbose logging still appears optimal to the agent.

## 4.4 Deception and overjustification in examples

We now theoretically analyze intentionally simple examples that illustrate the failure modes of RLHF in the presence of partial observability. Our examples feature an agent trained with RLHF to complete tasks in a user's terminal. The output of each command is piped to a log file, which is what the human evaluator sees when making choices for RLHF. For mathematical details about these examples, including a tutorial-style analysis of Example A with derivations, see Appendix C. For an empirical validation of these theoretical concerns and a proof of concept on how to mitigate the issues, see Section 5.2.

In Fig. 4A, the agent's task is to install Nvidia drivers and CUDA. The agent may skip the CUDA installation, or may attempt it with or without `2> /dev/null` (which suppresses errors) appended. If the log contains no message about CUDA, the human believes with probability $p_{\text{hide}}$ that the agent used the `2> /dev/null` trick and the install failed, and with probability $1 - p_{\text{hide}}$ that the agent didn't attempt the CUDA install. If $p_{\text{hide}}$ is sufficiently small, then RLHF will learn from the resulting feedback that the optimal policy is to use the `2> /dev/null` trick. This is deceptive inflation, because this policy produces trajectories for which the human systematically overestimates the return.

In Fig. 4B, the agent's task is to install Python and NumPy. The agent can skip the NumPy install, or attempt it with either "default" settings that *only logs errors* or with a `--verbose` flag that adds additional info. In the case of a success, the human prefers not to see this verbose output. If the log contains no message about NumPy, the human believes with probability $p_{\text{default}}$ that the agent attempted the install with default logging and succeeded, and with probability $1 - p_{\text{default}}$ that the agent skipped the NumPy install. If $p_{\text{default}}$ is sufficiently small, then RLHF will learn from the resulting feedback that the optimal policy is to use `--verbose` logging. This is overjustification, because the agent chooses a suboptimal policy in order to avoid the human underestimating its performance.

## 5 Return ambiguity from feedback under known partial observability

We've seen issues with standard RLHF applied to feedback from partial observations. Part of the problem is *model misspecification*: the standard RLHF model implicitly assumes full observability. Assuming the human's partial observability is known, could one do better?

We start Section 5.1 by analyzing how much information the feedback process provides about the return function when the human's choice model under partial observations is known precisely. We show that the feedback determines the correct return function up to an additive constant and a linear subspace we call the ambiguity (Theorem 5.2). If the human had a return function that differed from the true return function by an element in the ambiguity, they would give the exact same feedback — such return functions are thus feedback-compatible. We then show an example where the ambiguity

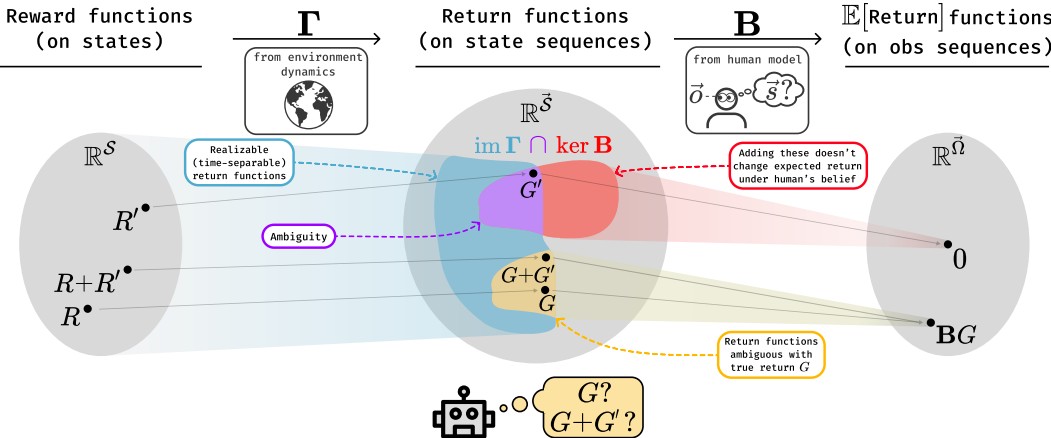

Figure 5: By Theorem 5.2, even with infinite comparison data and access to the correct human model, a hypothetical reward learning system (depicted as a robot) could only infer $G$ up to the ambiguity $\text{im } \mathbf{\Gamma} \cap \text{ker } \mathbf{B}$ (purple). Adding an element of the ambiguity to $G$ leads to the exact same choice probabilities for all possible comparisons, and the reward learning system has no way to identify $G$ among the return functions in $G + (\text{im } \mathbf{\Gamma} \cap \text{ker } \mathbf{B})$ (yellow). This abstract depiction ignores the linearity of these spaces; for a more precise geometric depiction of $\mathbf{B}$, see Figure 8 in the appendix.

vanishes, and another where it doesn't, leading to feedback-compatible return functions that have optimal policies with high regret under the true return function. Finally, in Section 5.2 we explore how one could in theory use Theorem 5.2 as a starting point to design reward learning techniques that work under partial observability. In particular, we experimentally show in a proof of concept that being aware of the human's partial observability improves performance. In this section we do not assume $P_O$ to be deterministic.

### 5.1 Feedback-compatibility and ambiguity of return functions

Assume that the human gives feedback based on the choice-probabilities from Eq. (2). In the infinite data limit, it can be assumed that the whole collection of probabilities $\left( P^G \left( \vec{o} \succ \vec{o}' \right) \right)_{\vec{o}, \vec{o}'}$ is known since the choice frequencies approach these probabilities. Here, we write $P^G$ instead of $P^R$ since the reward function only enters the choice probabilities through the corresponding return function $G$. The question we answer in this section is *how much information* the choice probabilities provide about $G$, assuming the human choice model is known and correct. The choice probabilities tell us precisely that the true return function *gives rise to these choice probabilities*, i.e., is feedback-compatible. This is captured in the following definition:

**Definition 5.1.** *Let* $\left( P^G \left( \vec{o} \succ \vec{o}' \right) \right)_{\vec{o}, \vec{o}'}$ *be the vector of choice probabilities and $\tilde{G}$ a return function corresponding to a reward function $\tilde{R}$. Then $\tilde{G}$ is* feedback-compatible *(with respect to the vector of choice probabilities) if $P^{\tilde{G}}(\vec{o} \succ \vec{o}') = P^G(\vec{o} \succ \vec{o}')$ for all $\vec{o}, \vec{o}' \in \vec{\Omega}$.*

Crucially, without further assumptions or inductive biases, no learning algorithm can pick out the true return function among feedback-compatible return functions. It is thus crucial to know whether there are feedback-compatible return functions that are unsafe when using them to optimize a policy.

We now determine the set of feedback-compatible return functions. Write $\mathbf{\Gamma} \in \mathbb{R}^{\vec{S} \times S}$ for the matrix that maps a reward function to its return function, i.e. $(\mathbf{\Gamma} \cdot R)(\vec{s}) := \sum_{t=0}^{T} \gamma^t R(s_t)$. Its matrix elements are given by $\mathbf{\Gamma}_{\vec{s}s} = \sum_{t=0}^{T} \delta_s(s_t) \gamma^t$, where $\delta_s(s_t) = \mathbf{1}\{s = s_t\}$. Then the *image* $\text{im } \mathbf{\Gamma}$ is the set of all return functions that can be realized from a reward function given the MDP dynamics $\mathcal{T}$. Recall the belief matrix $\mathbf{B} = \left( B(\vec{s} \mid \vec{o}) \right)_{\vec{o}, \vec{s}} \in \mathbb{R}^{\vec{\Omega} \times \vec{S}}$. Taking into account that $G$ itself is in $\text{im } \mathbf{\Gamma}$ and that $G$ enters the choice probabilities only through $\mathbf{B} \cdot G$ — meaning that the choice probabilities do not vary if we change $G$ additively up to an element in the kernel $\text{ker } \mathbf{B}$ — we obtain the following result:

**Theorem 5.2.** *Let the collection of choice probabilities be given by $\left( P^R \left( \vec{o} \succ \vec{o}' \right) \right)_{\vec{o}, \vec{o}' \in \vec{\Omega}}$ following a Boltzmann rational model as in Eq. (2). Then a return function $\tilde{G}$ is feedback-compatible if and only if there is $G' \in \ker \mathbf{B} \cap \operatorname{im} \mathbf{\Gamma}$ and $c \in \mathbb{R}$ such that $\tilde{G} = G + G' + c$. In particular, the choice probabilities determine $G$ up to an additive constant if and only if $\ker \mathbf{B} \cap \operatorname{im} \mathbf{\Gamma} = \{0\}$.*

See Theorem D.2 and Corollary D.4 for full proofs, and Figure 5 for a visual depiction. This result motivates the following definition:

**Definition 5.3** (Ambiguity). *We call $\ker \mathbf{B} \cap \operatorname{im} \mathbf{\Gamma}$ the ambiguity that is left in the return function when the human choice model and observation-based choice probabilities are known.*

**How large is the return ambiguity?** For Fig. 4A, one can show that the ambiguity is nontrivial, allowing for feedback-compatible return functions with unsafe optimal policies. Intuitively, since successfully installing CUDA produces the same observation regardless of whether `2> /dev/null` was used, the choice probabilities don't give us any information to determine distinct reward values for these two outcomes, only their average over the human's belief upon observing a successful install. Thus, reward functions assigning arbitrarily high reward to success with `2> /dev/null` are feedback-compatible. Such reward functions can then lead to an incentive for a learned policy to hide the error messages *even with a correct observation model*. More details can be found in Appendix C.4.

We saw in Fig. 4B a case where naive RLHF under partial observability can lead to overjustification. However, the human's feedback and belief model actually provide enough information to determine the return function. The reason is that $\ker \mathbf{B}$ leaves only one degree of freedom that is not "time-separable" over states, and thus $\ker \mathbf{B} \cap \operatorname{im} \mathbf{\Gamma} = \{0\}$. More details can be found in Appendix C.4.

## 5.2 Toward improving RLHF in partially observable settings

To improve RLHF when partial observability is unavoidable, one could take Theorem 5.2 as a starting point to find a learning algorithm that converges to feedback-compatible return functions. This would require the human model to be fully known and specified, including knowledge of the belief probabilities $B(\vec{s} \mid \vec{o})$, which can differ from human to human. If one assumes the human is rational, as in Appendix D.1, this requires specifying the human's policy prior $B(\pi)$. Instead of directly specifying these models, one could also attempt to *learn* a generative model for $B(\vec{s} \mid \vec{o})$. These problems reveal a further conceptual challenge: for complex environments, humans do not form beliefs over the entire environment state $s$. A better starting point for practical work may thus be to model humans as forming expectations over *reward-relevant features* of the state.

If $\mathbf{B}$ were explicitly known, one could in principle encode $\mathbf{B}$ into the loss function of an adapted RLHF process to learn a feedback-compatible return function; see Appendix D.3. As a proof of concept, we used this procedure to analyze the examples in Figure 4 empirically, see Table 1. We do this by first learning a reward model by logistic regression against the true choice probabilities of a synthetic human under partial observability, and then learning the optimal $Q$-function of the resulting reward model with value iteration. The resulting policy chooses a unique action after installation of the nvidia driver (Example A) or Python (Example B) as listed in the "action" column.

Table 1 shows that in 3 of four cases, being "partial observability aware" ("po-aware") leads to the true optimal policy when "naive" RLHF does not. In the one case where being "po-aware" does not improve performance (second line in the table), this is explained by the fact that there is remaining ambiguity in the return function. Curiously, in line 4 our theory also predicts remaining ambiguity, but the optimal policy is learned; we consider this to be luck. We provide more details on our experiments in Appendix C.5.

As we already demonstrated, feedback-compatible return functions can be unsafe due to remaining ambiguity. In Example D.29, we even show a case where some feedback-compatible return functions have optimal policies that are even worse than simply maximizing $J_{\text{obs}}$. An important direction for future work is to investigate learning algorithms and inductive biases that help "find" safe return functions among all those that are feedback-compatible, or that act conservatively given the uncertainty. Another line of inquiry is to determine when the set of feedback-compatible return functions is "safe", which depends on the MDP, observation function, and human model.

One sufficient condition for feedback-compatible return functions to be safe is the vanishing of the ambiguity $\ker \mathbf{B} \cap \operatorname{im} \mathbf{\Gamma}$. Even then, one realistically still has to deal with the problem that $\mathbf{B}$ is at

Table 1: Experiments showing improved performance of po-aware RLHF

| Ex. | $p$ | $p_{\text{hide}}$ | $p_{\text{default}}$ | **model** | **action** | $\overline{E}^+$ | **dec. infl.** | $\overline{E}^-$ | **overj.** | **optimal** |
|-----|-----|------|---------|----------|--------|-----|-----|-----|-----|------|
| A | 0.5 | 0.5 | N/A | naive | $a_H$ | 1.5 | ✓ | 0 | ✗ | ✗ |
| A | 0.5 | 0.5 | N/A | po-aware | $a_H$ | 1.5 | ✓ | 0 | ✗ | ✗ |
| A | 0.1 | 0.9 | N/A | naive | $a_C$ | 0 | ✗ | 0 | ✓ | ✗ |
| A | 0.1 | 0.9 | N/A | po-aware | $a_T$ | 0 | ✗ | 5.4 | ✗ | ✓ |
| B | 0.5 | N/A | 0.9 | naive | $a_T$ | 4.5 | ✓ | 0 | ✓ | ✗ |
| B | 0.5 | N/A | 0.9 | po-aware | $a_D$ | 0 | ✗ | 0.25 | ✗ | ✓ |
| B | 0.5 | N/A | 0.1 | naive | $a_V$ | 0 | ✗ | 0 | ✓ | ✗ |
| B | 0.5 | N/A | 0.1 | po-aware | $a_D$ | 0 | ✗ | 2.25 | ✗ | ✓ |

best known approximately. Fortunately, in Appendix D.6, we prove that small errors in the assumed belief matrix lead to only small errors in the inferred return function:

**Theorem 5.4.** *Assume* $\ker \mathbf{B} \cap \operatorname{im} \mathbf{\Gamma} = \{0\}$. *Let* $\mathbf{B_\Delta} := \mathbf{B} + \mathbf{\Delta}$ *be a small perturbation of* $\mathbf{B}$, *where* $\|\mathbf{\Delta}\| \leq \rho$ *for sufficiently small* $\rho$. *Let* $G$ *be the true return function and assume that a hypothetical learning system, assuming the human's belief is* $\mathbf{B_\Delta}$, *infers the return function* $\tilde{G}$ *with the property that* $\mathbf{B_\Delta} \cdot \tilde{G}$ *has the smallest possible Euclidean distance to* $\mathbf{B} \cdot G$.

*Let* $\mathrm{r}(\mathbf{B}) := \mathbf{B}|_{\operatorname{im} \mathbf{\Gamma}}$ *be the (injective) restriction of the operator* $\mathbf{B}$ *to* $\operatorname{im} \mathbf{\Gamma}$. *Then* $\mathrm{r}(\mathbf{B})^T \mathrm{r}(\mathbf{B})$ *is invertible, and there exists a polynomial* $Q(X, Y)$ *of degree* 5 *such that*

$$\|\tilde{G} - G\| \leq \rho \cdot \|G\| \cdot Q\Big(\big\|\big(\mathrm{r}(\mathbf{B})^T \mathrm{r}(\mathbf{B})\big)^{-1}\big\|, \|\mathrm{r}(\mathbf{B})\|\Big).$$

In particular, as we show in the appendix, one can uniformly bound the difference between $J_{\tilde{G}}$ and $J_G$. This yields a regret bound between the policy optimal under $\tilde{G}$ and an optimal policy $\pi^*$ for $G$.

There are also alternatives to modeling the human belief $\mathbf{B}$. For example, one could mix human evaluations based on high-cost full observations and low-cost partial observations for finding an optimal tradeoff [Mallen and Belrose, 2024]. Finally, it would help if the human could *query* the policy about reward-relevant aspects of the environment to bring the setting closer to RLHF from full observations. This is similar to the problem of eliciting the latent knowledge of a predictor of future observations [Christiano et al., 2021, Christiano and Xu, 2022]. While this may avoid the need to specify the *human's* belief model $B(\vec{s} \mid \vec{o})$, it requires understanding and effectively querying an *ML model's* belief, including translating from an ML model's ontology into a human ontology.

## 6 Conclusions

In this paper, we provided a conceptual and theoretical investigation of challenges when applying RLHF from partial observations. First, we saw that applying RLHF naively when *assuming* full observability can lead to deceptive inflation and overjustification behavior. Then, we showed that even when the human's partial observability is known, the set of feedback-compatible return functions can contain irreducible ambiguity. This means that without further inductive biases, no learning algorithm can generally be expected to infer the correct return function. Finally, we recommended further exploratory research to study and improve RLHF for cases when partial observability is unavoidable and provided a proof of concept that modeling the human's partial observability can improve performance. In conclusion, we recommend caution when using RLHF in situations of partial observability, and hope that further research studies the effects in practice and helps to address these challenges.

**Limitations** We assume the human to be Boltzmann rational and to implicitly compute an expected value of the return, which is unrealistic for actual humans. Other types of choices could be considered, as in reward-rational choice [Jeon et al., 2020] and assistance games [Hadfield-Menell et al., 2016]. Finally, we assume that the human forms a belief $B(\vec{s} \mid \vec{o})$ over the true state sequence $\vec{s}$. If the environment is complex, humans will in reality only form beliefs over lower-dimensional representations or features of the state.

## Author contributions

The project was conceived in parallel by **Scott** and **Davis**, with a key shift proposed by **Leon**. **Leon** proved Proposition 4.1 and Theorems 5.2 and 5.4, found the first mathematical examples of what became deceptive inflation and overjustification that can be resolved by Theorem 5.2, and wrote the majority of the appendix. **Davis** conjectured Proposition 4.1, provided early empirical evidence that RLHF under partial observations can lead to deception (not in the paper), defined deception / deceptive inflation and overjustification (with **Scott**), proved Theorem 4.5, and developed the running examples and figures. **Scott** guided the project direction and prioritization, gave the conjecture and proof idea for Theorem 5.4, and helped develop the running examples and deception definitions. **Erik** provided regular detailed feedback and guidance and edited the paper. **Anca** and **Stuart** advised this project.

## Acknowledgments and Disclosure of Funding

Leon Lang thanks the Center for Human-Compatible Artificial Intelligence for hosting him during part of this project, and Open Philanthropy for financial support. All authors thank Open Philanthropy for its support of the Center for Human-Compatible Artificial Intelligence. Davis was supported by the Berkeley Existential Risk Initiative. Erik was supported by fellowships from the Future of Life Institute and Open Philanthropy. We thank Benjamin Eysenbach and Benjamin Plaut for detailed comments and feedback on this work, and we thank Elio A. Farina, Mary Marinou, and Alexandra Horn for assistance with graphic design.

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

# APPENDIX

In the appendix, we provide more extensive theory, proofs, and examples. The appendix makes free use of concepts and notation defined in the main paper. In particular, throughout we assume a general MDP together with observation kernel $P_O : \mathcal{S} \to \Omega$ and a human with general belief kernel $B(\vec{o} \mid \vec{s})$, unless otherwise stated. See the list of Symbols in Section A to refresh notation.

In Section C we supplement the examples from the main paper with more mathematical details.

In Section D, we provide an extensive theory for appropriately modeled partial observability in RLHF. This can mainly be considered a supplement to Section 5 and contains our main theorems, supplementary results, analysis of special cases, and examples.

In Section E, we analyze the naive application of RLHF under partial observability, which means that the learning system is not aware of the human's partial observability. This section is essentially a supplement to Section 4 and contains an analysis of the policy evaluation function $J_{\mathrm{obs}}$, of deceptive inflation and overjustification, and further extensive mathematical examples showing the failures of naive RLHF under partial observability.

## Contents of the Appendix

# A   List of Symbols

**General MDPs**

| | |
|---|---|
| $\mathcal{S}$ | Set of environment states $s \in \mathcal{S}$ |
| $\mathcal{A}$ | Set of actions $a \in \mathcal{A}$ of the policy |
| $\Delta(\mathcal{S})$ | Set of probability distributions over $\mathcal{S}$. Can be defined for any finite set |
| $\mathcal{T} : \mathcal{S} \times \mathcal{A} \to \Delta(\mathcal{S})$ | Transition kernel |
| $P_0 \in \Delta(\mathcal{S})$ | Initial state distribution |
| $R \in \mathbb{R}^{\mathcal{S}}$ | Usually the true reward function |
| $R' \in \mathbb{R}^{\mathcal{S}}$ | Usually a reward function in the kernel of $\mathbf{B} \circ \boldsymbol{\Gamma}$ |
| $\tilde{R} \in \mathbb{R}^{\mathcal{S}}$ | Usually another reward function, e.g. inferred by a learning system |
| $\gamma \in [0, 1]$ | Discount factor |
| $\pi : \mathcal{S} \to \Delta(\mathcal{A})$ | A policy |
| $\mathcal{T}^{\pi} : \mathcal{S} \to \Delta(\mathcal{S})$ | Transition kernel for a fixed policy $\pi$ given by $\mathcal{T}^{\pi}(s' \mid s) = \sum_{a \in \mathcal{A}} \mathcal{T}(s' \mid s, a) \cdot \pi(a \mid s)$ |
| $T \in \mathbb{N}$ | Finite time horizon |
| $P^{\pi} \in \Delta(\mathcal{S}^T)$ | State sequence distribution induced by the policy $\pi$ |
| $\vec{\mathcal{S}} \subseteq \mathcal{S}^T$ | State sequences $\vec{s} \in \vec{\mathcal{S}}$ supported by $P^{\pi}$ |
| $G \in \mathbb{R}^{\vec{\mathcal{S}}}$ | Usually the true return function given by $G(\vec{s}) = \sum_{t=0}^{T} \gamma^t R(s_t)$. |
| $G' \in \mathbb{R}^{\vec{\mathcal{S}}}$ | Usually a return function in $\ker \mathbf{B}$ |
| $\tilde{G} \in \mathbb{R}^{\vec{\mathcal{S}}}$ | Usually another return function, e.g. inferred by a learning system |
| $J$ | The true policy evaluation function given by $J(\pi) = \mathbf{E}_{\vec{s} \sim P^{\pi}}\left[G(\vec{s})\right]$. |

**Additions to General MDPs with Partial Observability**

| | |
|---|---|
| $\Omega$ | Set of possible observations $o \in \Omega$ |
| $P_O : \mathcal{S} \to \Delta(\Omega)$ | Observation kernel determining the human's observations |
| $P_{\vec{O}} : \vec{\mathcal{S}} \to \Delta(\Omega^T)$ | The observation sequence kernel given by $P_{\vec{O}}(\vec{o} \mid \vec{s}) = \prod_{t=0}^{T} P_O(o_t \mid s_t)$ |
| $\vec{\Omega} \subseteq \Omega^T$ | The set of observed sequences $\vec{o} \in \Omega^T$ that can be sampled from $P_{\vec{O}}(\cdot \mid \vec{s})$ for $\vec{s} \in \vec{\mathcal{S}}$ |
| $O : \mathcal{S} \to \Omega$ | Observation function for the case that $P_O$ is deterministic; given by $O(s) = o$ with $o$ such that $P_O(o \mid s) = 1$ |
| $\vec{O} : \vec{\mathcal{S}} \to \vec{\Omega}$ | Observation sequence function for the case that $P_{\vec{O}}$ is deterministic; given by $\vec{O}(\vec{s}) = \vec{o}$ with $\vec{o}$ such that $P_{\vec{O}}(\vec{o} \mid \vec{s}) = 1$ |
| $G_{\vec{o}} \in \mathbb{R}^{\{\vec{s} \in \vec{\mathcal{S}} \mid \vec{O}(\vec{s}) = \vec{o}\}}$ | Restriction of the return function $G \in \mathbb{R}^{\vec{\mathcal{S}}}$ to $\left\{\vec{s} \in \vec{\mathcal{S}} \mid \vec{O}(\vec{s}) = \vec{o}\right\}$ for fixed $\vec{o} \in \vec{\Omega}$ |
| $G_{\text{obs}} \in \mathbb{R}^{\vec{\mathcal{S}}}$ | Return function that can be inferred when partial observability is not properly modeled, given by $G_{\text{obs}}(\vec{s}) := \left(\mathbf{B} \cdot G\right)\left(\vec{O}(\vec{s})\right)$ |
| $J_{\text{obs}}$ | Observation policy evaluation function, defined in Eq. (4) |

## State- and Observation Sequences

$s_t \in \mathcal{S}$    The $t$'th entry in a state sequence $\vec{s}$

$\vec{s} \in \mathcal{S}^T$    State sequence $\vec{s} = s_0, \ldots, s_T$

$\hat{s} \in \mathcal{S}^t$    State sequence segment $\hat{s} = s_0, \ldots, s_t$ for $t \leq T$

$o_t \in \Omega$    The $t$'th entry in an observation sequence $\vec{o}$

$\vec{o} \in \Omega^T$    Observation sequence $\vec{o} = o_0, \ldots, o_T$

$\hat{o} \in \Omega^t$    Observation sequence segment $\hat{o} = o_0, \ldots, o_t$ for $t \leq T$

## The Human's Belief

$B(\pi')$ — The human's policy prior

$B(\vec{s})$ — The human's prior belief that a sequence $\vec{s}$ will be sampled, given by $B(\vec{s}) = \int_{\pi'} B(\pi')P^{\pi'}(\vec{s})d\pi'$

$B(\vec{s} \mid \vec{o})$ — The human's belief of a state sequence given an observation sequence, see Proposition D.1 for a Bayesian version

$B^{\pi}(\vec{s} \mid \vec{o})$ — The human's belief of a state sequence given an observation sequence; it is allowed to depend on the true policy $\pi$, see Proposition D.1

$B_{\vec{o}} \in \mathbb{R}^{\{\vec{s} \in \mathcal{S} \mid \vec{O}(\vec{s}) = \vec{o}\}}$ — Vector of prior probabilities $B(\vec{s})$ for $\vec{s} \in \left\{ \vec{s} \in \vec{\mathcal{S}} \mid \vec{O}(\vec{s}) = \vec{o} \right\}$

## Identifiability Theorem

$\beta > 0$ — The inverse temperature parameter of the Boltzmann rational human

$\sigma : \mathbb{R} \to (0, 1)$ — The sigmoid function given by $\sigma(x) = \frac{1}{1+\exp(-x)}$

$\mathbf{\Gamma} : \mathbb{R}^{\mathcal{S}} \to \mathbb{R}^{\vec{\mathcal{S}}}$ — Function that maps a reward function $R$ to the return function $\mathbf{\Gamma}(R)$ with $\left[\mathbf{\Gamma}(R)\right](\vec{s}) = \sum_{t=0}^{T} \gamma^t R(s_t)$

$\mathbf{B} : \mathbb{R}^{\vec{\mathcal{S}}} \to \mathbb{R}^{\vec{\Omega}}$ — Function that maps a return function $G$ to the expected return function $\mathbf{B}(G)$ on observation sequences given by $\left[\mathbf{B}(G)\right](\vec{o}) = \mathbf{E}_{\vec{s} \sim B(\vec{s}|\vec{o})}\left[G(\vec{s})\right]$

$\mathbf{F} : \mathbb{R}^{\mathcal{S}} \to \mathbb{R}^{\vec{\Omega}}$ — The composition $\mathbf{F} = \mathbf{B} \circ \mathbf{\Gamma}$

$P^R(\vec{s} \succ \vec{s}')$ — Boltzmann rational choice probability in the case of full observability (Eq. (1))

$P^R(\vec{o} \succ \vec{o}')$ — Boltzmann rational choice probability in the case of partial observability (Eq. (2))

$\mathbf{O} : \mathbb{R}^{\vec{\Omega}} \to \mathbb{R}^{\vec{\mathcal{S}}}$ — Abstract linear operator given by $\left[\mathbf{O}(v)\right](\vec{s}) = \mathbf{E}_{\vec{o} \sim P_{\vec{O}}(\vec{o}|\vec{s})}\left[v(\vec{o})\right]$

$\mathbf{O} \otimes \mathbf{O} : \mathbb{R}^{\vec{\Omega} \times \vec{\Omega}} \to \mathbb{R}^{\vec{\mathcal{S}} \times \vec{\mathcal{S}}}$ — Formally the Kronecker product of $\mathbf{O}$ with itself, explicitly given by $\left[(\mathbf{O} \otimes \mathbf{O})(C)\right](\vec{s}, \vec{s}') = \mathbf{E}_{\vec{o}, \vec{o}' \sim P_{\vec{O}}(\cdot|\vec{s}, \vec{s}')}\left[C(\vec{o}, \vec{o}')\right]$

## Robustness to Misspecifications

$\|x\|$ — Euclidean norm of the vector $x \in \mathbb{R}^k$

$\|\mathbf{A}\|$ — Matrix norm of the matrix $\mathbf{A}$, given by $\|\mathbf{A}\| := \max_{x, \|x\|=1} \|\mathbf{A}x\|$

$\tau(\mathbf{A})$ — Matrix quantity defined in Equation (9)

$C(\mathbf{A}, \rho)$ — Matrix quantity defined in Equation (10)

$\mathrm{r}(\mathbf{B})$ — Restriction of $\mathbf{B}$ to $\operatorname{im}\mathbf{\Gamma}$

**General Sets and (Linear) Functions**

| | |
|---|---|
| $\lvert A \rvert$ | Number of elements in the set $A$ |
| $A \cap C$ | Intersection of sets $A$ and $C$ |
| $A \cup C$ | Union of sets $A$ and $C$ |
| $A \setminus C$ | Relative complement of $C$ in $A$ |
| $\delta_x$ | The Dirac delta distribution of a point $x$ in a set; given by $\delta_x(A) = 1$ if $x \in A$ and $\delta_x(A) = 0$, else |
| $\ker \mathbf{A}$ | The kernel of a linear operator $\mathbf{A} : V \to W$; given by $\ker \mathbf{A} = \{ v \in V \mid \mathbf{A}(v) = 0 \}$ |
| $\operatorname{im} \mathbf{A}$ | The image of a linear operator $\mathbf{A} : V \to W$; given by $\operatorname{im} \mathbf{A} = \{ w \in W \mid \exists v \in V : \mathbf{A}(v) = w \}$ |
| $f^{-1}(y)$ | Preimage of $y$ under a function $f : X \to Y$; given by $f^{-1}(y) = \{ x \in X \mid f(x) = y \}$ |

# B   More related work

Here we extend the related work outlined in Section 2.

A review of limitations of RLHF, including a brief discussion of partial observability, can be found in Casper et al. [2023]. RLHF is a special case of reward-rational choice [Jeon et al., 2020], a general framework which also encompasses demonstrations-based inverse reinforcement learning [Ziebart et al., 2008, Ng et al., 2000] and learning from the initial environment state [Shah et al., 2019], and can be seen as a special case of assistance problems [Fern et al., 2014, Hadfield-Menell et al., 2016, Shah et al., 2021]. In all of these, the reward function is learned from human actions, which in the case of RLHF are simply preference statements. This requires us to specify the human policy of action selection—Boltzmann rationality in typical RLHF—which can lead to wrong reward inferences when this specification is wrong [Skalse and Abate, 2022]; unfortunately, the human policy can also not be learned alongside the human's values without further assumptions [Mindermann and Armstrong, 2018]. Instead of a model of the human policy, in this paper we mostly focus on the human *belief model* and misspecifications thereof for the case that the human only receives partial observations.

Related work [Zhuang and Hadfield-Menell, 2020] analyzes the consequences of aligning an AI with a proxy reward function that omits attributes that are important to the human's values, which could happen if the reward function is based on a belief over the world state given limited information. Another instance are recommendation systems [Stray, 2023], where user feedback does not depend on information *not* shown—which is crucially part of the environment. Siththaranjan et al. [2023] analyze what happens under RLHF if the *learning algorithm* doesn't have all the relevant information (e.g. about the identity of human raters), complementing our study of what happens when human raters are missing information. Chidambaram et al. [2024] and Park et al. [2024a] deal with the situation that different human evaluators may vary in their unobserved preference types. In contrast, we assume a single human evaluator with fixed reward function, which can be motivated by cases where the human choices are guided by a behavior policy, constitution, or a model spec [Tong Mu, 2024, Anthropic, 2023b, OpenAI, 2024b]. Kausik et al. [2024] assumes that the choices of the human evaluator depend on an unobserved reward-state with its own transition dynamics, similar to an emotional state in a real human. In contrast, we assume the human to be stateless.

Finally, we mention connections to truthful AI [Evans et al., 2021, Lin et al., 2022, Burns et al., 2023, Huang et al., 2023], which is about ensuring that AI systems tell the truth about aspects of the real world. Partial observability is a mechanism that makes it feasible for models to lie without being caught: If the human evaluator does not observe the full environment, or does not fully understand it, then they may not detect when the AI is lying. More speculatively, we can imagine that AI models will at some point more directly influence human observations by *telling us* the outcomes of their actions. E.g., imagine an AI system that manages your assets and assures you that they are increasing in value while they are actually not. In our work, we leave this additional problem out of the analysis by assuming that the observations only depend on the environment state, and not directly on the agent's actions.

## C Details for deception and overjustification in examples

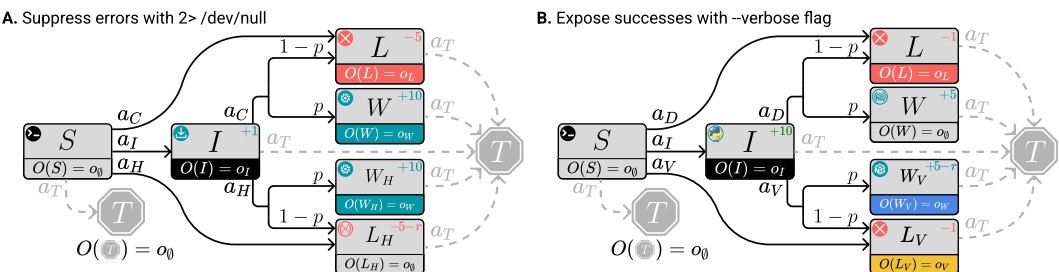

Figure 6: Two example MDPs with observation functions in which RLHF chooses undesirable policies. Each box depicts a state with a footer showing the (deterministic) observation produced by that state. Outgoing edges from each box are available actions. A more detailed diagram for the first MDP, with explicit shell commands and log messages, is available in Appendix C.3.

Here we include details to the examples described in Section 4.4 that illustrate the failure modes of RLHF in the presence of partial observability. For each of the following, we will characterize the policy which maximizes $J_{\text{obs}}$, as this is the policy RLHF selects for when observations are deterministic; see Proposition 4.1.

Our examples feature an agent trained with RLHF to complete tasks in a user's terminal. The output of each command (`stdout` and `stderr`) is piped to a log file, which is what the human evaluator sees when making choices for RLHF. We assume that the final state $T$ has a self-transition, episodes have a fixed horizon length of 3 (meaning state sequences have length 4: $s_0, \ldots, s_3$), $\gamma = 1$, and there is a fixed initial state $s_0 = S$. Both examples feature a fixed transition probability $0 < p < 1$. We assume that the human's belief only supports possible explanations: $B(\vec{s} \mid \vec{o}) > 0 \implies \vec{O}(\vec{s}) = \vec{o}$. For further details about these examples, including a tutorial-style analysis of Example A, see Appendix C.3.

### C.1 Example A: hiding failures

See Appendix C.3 for derivations and a tutorial-style analysis of this example.

In Fig. 4A (and Fig. 6A), the agent's task is to install CUDA, (states $W$ and $W_H$) which requires first installing Nvidia drivers (state $I$). If the CUDA installation fails, one enters $L$ or $L_H$. The $a_C$ action is a command to install CUDA that logs both successes and failures, whereas $a_H$ corresponds to $a_C$ with `2> /dev/null` appended, which suppresses error messages such that no log message is produced if the installation fails. The human may prefer that the agent not hide a failure; we price this into the reward function with a penalty $r \geq 0$ on the reward at $L_H$. The agent may attempt the CUDA installation before installing drivers, but this will fail.

There are three pairs of trajectories which produce identical observations. Here we address the most prominent (see Appendix C.3 for the others): $SITT$ and $SIL_HT$ both produce $o_\emptyset o_I o_\emptyset o_\emptyset$, stylized as a log containing only a success confirmation for Python (Fig. 1, $\vec{o}_2$). after successfully installing drivers, a failed CUDA installation with `2> /dev/null` ($SIL_HT$) and simply exiting ($SITT$) both produce a log containing only a success confirmation for the drivers ($o_\emptyset o_I o_\emptyset o_\emptyset$). Let $p_H := B(\vec{s} = SIL_HT \mid \vec{o} = o_\emptyset o_I o_\emptyset o_\emptyset) \in (0, 1)$ be the human's suspicion, upon seeing only a success confirmation for the drivers, that a failed CUDA installation was hidden.

We consider two cases which differ only in whether the expected value of attempting to install CUDA (with default logging, $a_C$) after installing the drivers is positive. In Example C.1, the success probability $p$ is high enough that the true optimal policy is to attempt the installation; in Example C.2, $p$ is low enough that the optimal policy is to terminate.

**Example C.1.** *If the true-reward-optimal policy $\pi^*$ is to attempt to install CUDA with default logging ($a_C$) after installing drivers, but $p_H < \frac{5}{5+r}$, then the $J_{\text{obs}}$-maximizing policy, which will be chosen by RLHF, is to instead use the `2> /dev/null` trick. Intuitively, this is because if the installation fails, the human is likely to believe that the agent just didn't attempt to install CUDA, which is preferred*

to a failed installation. This is deceptive inflation; the overestimation error $\overline{E}^+$ for this policy is positive, whereas $\overline{E}^+(\pi^*) = 0$, in line with Theorem 4.5.

If $p_H > \frac{5}{5+r}$, then RLHF will select $\pi^*$.

**Example C.2.** *If $\pi^*$ is to terminate ($a_T$) after installing the drivers, but $p_H < \frac{5}{5+r}$, then RLHF will learn the deceptive inflation* `2> /dev/null` *policy as in Example C.1.*

*However, if $p_H > \frac{5}{5+r}$, RLHF will choose the policy that attempts to install CUDA with default logging. Intuitively, this is because the combination of increased suspicion that the agent might be hiding errors (higher $p_H$) and increased penalty on this suspected behavior (higher $r$) also penalizes our preferred trajectory $SITT$ since it can be misinterpreted as hiding errors. This is overjustification; the overestimation error $\overline{E}^+$ is 0 (thus this is not deceptive inflation), and as required by Theorem 4.5, the underestimation error $\overline{E}^-$ is 0, lower than $E^-(\pi^*) = p_H(5 + r)$.*

## C.2 Example B: paying to reveal information

In Fig. 4B (and Fig. 6B), the agent's task is to install Python (state $I$) and to optionally further install NumPy (states $W$ and $W_V$). The $a_D$ action corresponds to a command to install NumPy with "default" settings which *only logs errors*, whereas $a_V$ corresponds to the same command with a `--verbose` flag that adds additional info. In the case of a success, the human distinctly prefers not to see this verbose output; we price this into the reward function with a penalty $r > 0$ on the reward at $W_V$.

There is only one pair of trajectories which produce identical observations: after successfully installing Python, a successful NumPy installation with default logging ($SIWT$) and simply exiting ($SITT$) both produce a log containing only a success confirmation for Python ($o_\emptyset o_I o_\emptyset o_\emptyset$). Let $p_D := B(\vec{s} = SIWT \mid \vec{o} = o_\emptyset o_I o_\emptyset o_\emptyset) \in (0, 1)$ be the human's optimism, upon seeing only a success confirmation for Python, that NumPy was also successfully installed (without the `--verbose` flag).

Here we consider only the case where $p$ is large enough that the true optimal policy is to install Python then attempt to install NumPy with default logging ($a_D$).

**Example C.3.** *If $\pi^*$ is to attempt to install NumPy with $a_D$ after installing Python, and $p_D > q := \frac{1}{5}\Big(p(6 - r) - 1\Big)$, then RLHF will select the policy that terminates after installing Python. Intuitively, this is because the agent can exploit the human's optimism that NumPy was installed quietly without taking the risk of an observable failure ($L$). This is deceptive inflation, with an overestimation error $\overline{E}^+$ of $5p_D$, greater than $\overline{E}^+(\pi^*) = 0$.*

*If instead $p_D < q$, then RLHF will select the policy that attempts the NumPy installation with verbose logging ($a_V$). Intuitively, this is because the agent is willing to "pay" the cost of $r$ true reward to prove to the human that it installed NumPy, even when the human does not want to see this proof. This is overjustification; the overestimation error $\overline{E}^+$ is 0 (thus this is not deceptive inflation), and the underestimation error $\overline{E}^-$ is 0, lower than $\overline{E}^-(\pi^*) = 5p(1 - p_D)$.*

## C.3 Derivations and Further Details for Fig. 4A

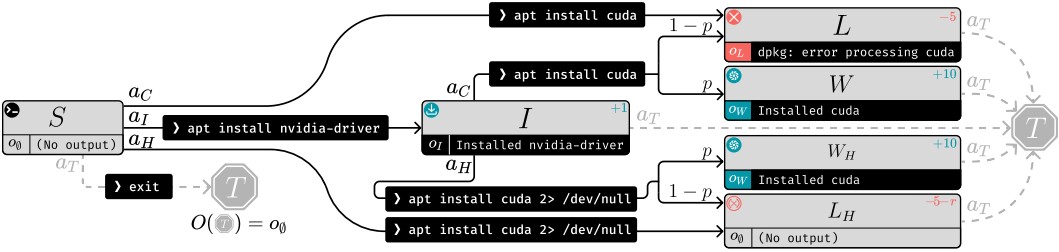

Figure 7: An expanded view of Figure 4A. Commands corresponding to the various actions are depicted along edges, and log messages corresponding to the various observations are depicted underneath each state.

We first include Figure 7, a more detailed picture of the MDP and observation function in Appendix C.1, to help ground the narrative details of the example.

Next we formally enumerate the details of the MDP and observation function.

- $\mathcal{S} = \{S, I, W, W_H, L, L_H, T\}$.
- $\mathcal{A} = \{a_I, a_C, a_H, a_T\}$.
- $\mathcal{T}$ is as depicted in Figure 7 and Figure 4A. For a state $s$, any outgoing arrow labeled with an action $a$ (such as $a_I$) describes the distribution $\mathcal{T}(s' \mid s, a)$ as follows: if the arrow does not split, then $\mathcal{T}(s' \mid s, a) = 1$ where $s'$ is the state the arrow points to; if the arrow does split, then for each successor state $s'$ it eventually reaches, a probability $q$ is written just before the box corresponding to $s'$ (for this example, $q = p$ or $q = 1 - p$), and $\mathcal{T}(s' \mid s, a) = q$.
  - Additionally, any action taken from a state that does not have an outgoing arrow corresponding to that action will immediately transition to state $T$, as though $a_T$ had been taken.
  - Any action taken from state $T$ transitions deterministically to $T$.
- $P_0(S) = 1$.
- $R$ is as described in the table (the numbers in the top right of each state box) with $r \geq 0$. Additionally, $R(S) = R(T) = 0$.
- $\gamma = 1$.

We work with a fixed horizon length of 3, meaning state sequences have length 4 (since time is zero-indexed: $s_0 s_1 s_2 s_3$).

The observation function is also depicted in Figure 7. Each state deterministically produces the observation in the lower-right corner of its box in the figure. We also write it in another format in Table 9.

Table 9: The observation function $O$ for the example in Appendix C.1 and Appendix C.3.

| $s$ | $S$ | $I$ | $W$ | $W_H$ | $L$ | $L_H$ | $T$ |
|---|---|---|---|---|---|---|---|
| $O(s)$ | $o_\emptyset$ | $o_I$ | $o_W$ | $o_W$ | $o_L$ | $o_\emptyset$ | $o_\emptyset$ |

We make the additional assumption that the human belief $B(\vec{s} \mid \vec{o})$ only supports state sequences $\vec{s}$ which actually produce $\vec{o}$ under the sequence observation function $\vec{O}$: $B(\vec{s} \mid \vec{o}) > 0 \implies \vec{O}(\vec{s}) = \vec{o}$. In particular, this means that for any $\vec{o}$ which is only produced by one $\vec{s}$, $B(\vec{o} \mid \vec{s}) = 1$.

There are three pairs of state sequences which produce identical observation sequences. For each, we introduce a parameter representing the probability the human infers the first of the pair of state sequences upon seeing their shared observation sequence.

1. $SIL_H T$ and $SITT$ both produce $o_\emptyset o_I o_\emptyset o_\emptyset$, a log containing only a success confirmation for installing drivers, again because $O(L_H) = O(T) = o_\emptyset$. Let $p_H = B(\vec{s} = SIL_H T \mid \vec{o} = o_\emptyset o_I o_\emptyset o_\emptyset)$.

2. $STTT$ and $SL_H TT$ both produce $o_\emptyset o_\emptyset o_\emptyset o_\emptyset$, an empty log, since $O(L_H) = O(T) = o_\emptyset$. Let $p'_H = B(\vec{s} = SL_H TT \mid \vec{o} = o_\emptyset o_\emptyset o_\emptyset o_\emptyset)$.

3. $SIWT$ and $SIW_H T$ both produce $o_\emptyset o_I o_W o_\emptyset$, a log containing success confirmations for installing both drivers and CUDA. Let $p_W = B(\vec{s} = SIWT \mid \vec{o} = o_\emptyset o_I o_W o_\emptyset)$.

We assume for simplicity that $p'_H = p_H$, i.e. that the human is just as likely to think an empty log following a successful driver installation contains an error that was hidden with `2> /dev/null` ($p_H$), as they are to think that an entirely empty log contains a hidden error ($p'_H$). In principle, this need not be true: the human may have differing priors about the agent's behavior in the two different circumstances. However, the algebra to reason about such a case is significantly more cumbersome, and this case reveals no fundamentally different agent behavior under our framework that isn't present in some simpler case.

We can thus write the full $B$ as a matrix as in Table 10.

Table 10: The parameterized human belief function $B$ for the example in Appendix C.1 and Appendix C.3, expressed as a matrix (rendered as a table). Any empty cell is equal to 0.

| | $STTT$ | $SL_HTT$ | $SLTT$ | $SITT$ | $SIL_HT$ | $SILT$ | $SIWT$ | $SIW_HT$ |
|---|---|---|---|---|---|---|---|---|
| $o_\emptyset o_\emptyset o_\emptyset o_\emptyset$ | $1-p_H$ | $p_H$ | | | | | | |
| $o_\emptyset o_L o_\emptyset o_\emptyset$ | | | $1$ | | | | | |
| $o_\emptyset o_I o_\emptyset o_\emptyset$ | | | | $1-p_H$ | $p_H$ | | | |
| $o_\emptyset o_I o_L o_\emptyset$ | | | | | | $1$ | | |
| $o_\emptyset o_I o_W o_\emptyset$ | | | | | | | $p_W$ | $1-p_W$ |

We have laid the groundwork sufficiently to begin reasoning about the observation return, overestimation and underestimation error, policies which are optimal under the reward function learned by naive RLHF, and the resulting deceptive inflationand overjustification failure modes. We begin by computing the measures of interest for each state sequence, shown in Table 11.

Table 11: Measures of interest for each state sequence for the example in Appendix C.1 and Appendix C.3. State sequences which produce the same observations have their $G_{\text{obs}}$ columns merged, since they necessarily have the same $G_{\text{obs}}$.

| $\vec{s}$ | $G(\vec{s})$ | $G_{\text{obs}}(\vec{s}) := \mathbf{E}_{\vec{s}' \sim B(\cdot\|\vec{O}(\vec{s}))}[G(\vec{s}')]$ | $E^+(\vec{s}) := \max(0, G_{\text{obs}}(\vec{s}) - G(\vec{s}))$ | $E^-(\vec{s}) := \max(0, G(\vec{s}) - G_{\text{obs}}(\vec{s}))$ |
|---|---|---|---|---|
| $STTT$ | $0$ | $p_H G(SL_HTT) + (1-p_H)G(STTT)$ $= -p_H(5+r)$ | $0$ | $p_H(5+r)$ |
| $SL_HTT$ | $-5-r$ | | $(1-p_H)(5+r)$ | $0$ |
| $SLTT$ | $-5$ | $-5$ | $0$ | $0$ |
| $SITT$ | $1$ | $p_H G(SIL_HT) + (1-p_H)G(SITT)$ $= 1 - p_H(5+r)$ | $0$ | $p_H(5+r)$ |
| $SIL_HT$ | $-4-r$ | | $(1-p_H)(5+r)$ | $0$ |
| $SILT$ | $-4$ | $-4$ | $0$ | $0$ |
| $SIWT$ | $11$ | $p_W G(SIWT) + (1-p_W)G(SIW_HT)$ $= 11$ | $0$ | $0$ |
| $SIW_HT$ | $11$ | | $0$ | $0$ |

As an exercise, checking the computations in Table 11 is a quick way to gain some intuition for how these quantities relate. It's further useful to speak about these quantities using their names, and work through the stories these expressions tell.

Consider the $E^+(SL_HTT)$ cell. What does it mean that this is $(1-p_H)(5+r)$? $E^+$ is the overestimation error; $(1-p_H)(5+r)$ is the expected amount by which the human observer, upon seeing a completely empty log ($o_\emptyset o_\emptyset o_\emptyset o_\emptyset$), will *overestimate* the total reward the agent attained when those empty logs were actually produced by $SL_HTT$. This is a trajectory in which the agent immediately uses the `2>/dev/null` trick, fails to install CUDA without logging the error, and terminates. Under what circumstances might the human overestimate the total reward when $\vec{s} = SL_HTT$? Upon seeing the empty log, the human thinks with probability $1-p_H$ that the agent simply terminated immediately, which would be worth a total reward of 0. Since the actual total reward is $-5-r$, this is an overestimate by $5+r$. The human thinks with probability $p_H$ that $SL_HTT$ occurred. This is correct, so there is no overestimation and this $p_H$ case does not contribute to the overestimation error. So we have that with probability $1-p_H$, the human overestimates the total reward by $5+r$.

We can keep going! Why is the underestimation error of $SIWT$ equal to 0? Because the only other trajectory with which it can be confused attains the same total reward, so regardless of how the probability mass of the human's belief divides between them, there will be no underestimation. Can all of the zeros in the overestimation and underestimation error columns be explained this way?

We now move on to consider policies rather than state sequences. Since a policy $\pi$ imposes a distribution $P^\pi$ over state sequences (the "on-policy distribution"), our policy measures are in fact exactly parallel to our state sequence measures. Each one is an expectation over the on-policy distribution of the columns of Table 11. We restrict our attention to deterministic policies which only take actions depicted in Figure 7 (i.e. that never terminate via an action other than $a_T$), of which there are only six in this MDP. They are enumerated, along with the policy-level measures, in Table 12. Policies will be written as a sequence of actions enclosed in brackets, omitting trailing repeated $a_T$ actions. This is nonstandard notation in an MDP with stochastic transitions, but is unambiguous in

this example, because all decisions are made before any stochasticity occurs. The policies are $[a_T]$, $[a_H a_T]$, $[a_C a_T]$, $[a_I a_T]$, $[a_I a_H a_T]$, and $[a_I a_C a_T]$.

Table 12: Measures of interest for each policy for the example in Appendix C.1 and Appendix C.3. Each of the columns here is the on-policy average of the corresponding column in Table 11. Policies are written as sequences of actions, omitting trailing repeated $a_T$ actions. This is nonstandard notation in an MDP with stochastic transitions, but is unambiguous in this example since all decisions are made before any stochasticity occurs.

| $\pi$ | $J(\pi)$ | $J_{\mathrm{obs}}(\pi)$ | $\overline{E}^+(\pi)$ | $\overline{E}^-(\pi)$ |
|---|---|---|---|---|
| $[a_T]$ | $0$ | $-p_H(5+r)$ | $0$ | $p_H(5+r)$ |
| $[a_H a_T]$ | $-5-r$ | $-p_H(5+r)$ | $(1-p_H)(5+r)$ | $0$ |
| $[a_C a_T]$ | $-5$ | $-5$ | $0$ | $0$ |
| $[a_I a_T]$ | $1$ | $1-p_H(5+r)$ | $0$ | $p_H(5+r)$ |
| $[a_I a_H a_T]$ | $pG(SIW_HT)$ $+(1-p)G(SIL_HT)$ $=11-(1-p)(15+r)$ | $pG_{\mathrm{obs}}(SIW_HT)$ $+(1-p)G_{\mathrm{obs}}(SIL_HT)$ $=11-(1-p)\left[10+p_H(5+r)\right]$ | $(1-p)(1-p_H)(5+r)$ | $0$ |
| $[a_I a_C a_T]$ | $pG(SIWT)$ $+(1-p)G(SILT)$ $=11-(1-p)\cdot 15$ | $pG_{\mathrm{obs}}(SIWT)$ $+(1-p)G_{\mathrm{obs}}(SILT)$ $=11-(1-p)\cdot 15$ | $0$ | $0$ |

With this we have everything we need to characterize optimal policies under the reward function learned by a naive application of RLHF ("policies selected by RLHF"). By Proposition 4.1, we know that if $P_O$ is deterministic, as in this example, RLHF selects policies which maximize $J_{\mathrm{obs}}$. In order to understand the behavior of these policies, we'll also need to determine the true optimal policies, i.e. those which maximize $J$. We'll proceed in cases, only considering boundary cases (specific measure-zero parameter values for which the result is different) insofar as they are interesting.

**Case 1:** $p > \frac{1}{3}$. If $p > \frac{1}{3}$, the CUDA install (with default logging, $a_C$) is likely enough to succeed that it's worth attempting it: $p \cdot R(W) + (1-p) \cdot R(L) > 0$. It also immediately follows that

$$J([a_I a_C a_T]) = J_{\mathrm{obs}}([a_I a_C a_T]) = 11 - (1-p)\cdot 15 > 1.$$

This allows us to eliminate policies $[a_T]$, $[a_H a_T]$, $[a_C a_T]$, and $[a_I a_T]$, which all have $J \leq 1$ and $J_{\mathrm{obs}} \leq 1$. None of them can thus be $J$-optimal or $J_{\mathrm{obs}}$-optimal. All that remains is to compare $J$ and $J_{\mathrm{obs}}$ for $[a_I a_H a_T]$ and $[a_I a_C a_T]$. We can check the sign of the differences of these pairs of values, starting with $J$.

$$J([a_I a_C a_T]) - J([a_I a_H a_T]) = (1-p)r.$$

Since $p$ is a probability and $r$ is nonnegative, this value is positive (and thus $[a_I a_C a_T]$ is preferred to $[a_I a_H a_T]$ by the human) if and only if $p < 1$ and $r > 0$.

$$J_{\mathrm{obs}}([a_I a_H a_T]) - J_{\mathrm{obs}}([a_I a_C a_T]) = (1-p)\left[5 - p_H(5+r)\right].$$

This value is positive (and thus $[a_I a_H a_T]$ is the policy RLHF selects) if and only if $p < 1$ and $p_H < \frac{5}{5+r}$.

If $p = 1$, then both differences are 0, and both $J$ and $J_{\mathrm{obs}}$ are indifferent between the two policies. This makes sense, as they differ only in the case where the CUDA installation fails; this happens with probability $1 - p = 0$ when $p = 1$. Now suppose $p < 1$. If $r = 0$, then the human is indifferent between the two policies. This also makes sense, as $r$ is meant to quantify the extent to which the human dislikes suppressed failures; if it's zero, then the human doesn't care. However, if $p_H < \frac{5}{5+r}$, then $J_{\mathrm{obs}}([a_I a_H a_T]) > J_{\mathrm{obs}}([a_I a_H a_T])$, and thus RLHF favors the `2>/dev/null` policy $[a_I a_H a_T]$.

If $p < 1$, $r > 0$, and $p_H < \frac{5}{5+r}$, then we have that $J([a_I a_C a_T]) > J([a_I a_H a_T])$ but $J_{\mathrm{obs}}([a_I a_C a_T]) > J_{\mathrm{obs}}([a_I a_H a_T])$. Thus RLHF will select the `2>/dev/null` policy $[a_I a_H a_T]$, and by Theorem 4.5, since $[a_I a_H a_T]$ is not $J$-optimal, then relative to $[a_I a_C a_T]$, it must exhibit deceptive inflation, overjustification, or both. Intuitively, we should be suspicious that deceptive

inflation is at play whenever the agent hides information from the human. Indeed, referencing Table 12, we have $\overline{E}^+([a_I a_H a_T]) = (1-p)(1-p_H)(5+r) > 0 = \overline{E}^+([a_I a_C a_T])$. Together with $J_{\text{obs}}([a_I a_H a_T]) > J_{\text{obs}}([a_I a_C a_T])$, this satisfies the conditions of Definition 4.3, and thus this is an instance of deceptive inflation.

If $p < 1$, $r > 0$, and $p_H > \frac{5}{5+r}$, then $[a_I a_C a_T]$ is optimal under both $J$ and $J_{\text{obs}}$, and in this case, RLHF selects the true optimal policy.

**Case 2:** $p < \frac{1}{3}$. In this case, the CUDA install is not likely enough to succeed to be worth attempting (under the true reward function). Mathematically, $J([a_I a_H a_T]) \leq J([a_I a_C a_T]) < 1 = J([a_I a_T])$. The other three policies are always worse under $J$ than $[a_I a_T]$, so we have our optimal policy $\pi^* = [a_I a_T]$. However,

$$J_{\text{obs}}([a_I a_H a_T]) - J_{\text{obs}}([a_I a_T]) = p(10 + p_H(5 + r)),$$

which is always positive unless $p = 0$. As long as $p > 0$, i.e. installing CUDA is actually possible at all, then regardless of everything else, $J_{\text{obs}}([a_I a_H a_T]) > J_{\text{obs}}([a_I a_T])$. We can interpret this reasonably: however good or bad it might look to terminate after installing drivers ($SITT$), using the `2>/dev/null` trick will either produce the same observation as $SITT$ (and thus the same feedback distribution) or will actually visibly install CUDA successfully—the most lucrative possible source of positive feedback. Even if there's only a sliver of a chance of success and it's a catastrophe to have a silent failure (i.e. $r \gg 0$), however likely you are to infer that the lack of a log message about CUDA means the agent hid a failure, it will penalize $[a_I a_T]$ *more* than it will penalize $[a_I a_H a_T]$: where $[a_I a_H a_T]$ produces this ambiguous observation with probability $1 - p$ (when it fails), $[a_I a_T]$ will *always* produce the ambiguous observation.

This means that when $0 < p < \frac{1}{3}$, it is impossible to recover the true optimal policy with naive RLHF. Which policies can possibly be $J_{\text{obs}}$-optimal for some setting of the parameters? We can similarly rule out $[a_T]$ and $[a_H a_T]$ for $0 < p < \frac{1}{3}$:

$$J_{\text{obs}}([a_I a_H a_T]) - J_{\text{obs}}([a_I a_T]) = p(10 + p_H(5 + r)) > 0.$$

We can rule out $[a_C a_T]$ by comparison to $[a_I a_C a_T]$: $J_{\text{obs}}([a_I a_C a_T]) - J_{\text{obs}}([a_C a_T]) = 16 - (1 - p)15 > 0$. So we are left with only $[a_I a_H a_T]$ and $[a_I a_C a_T]$ as candidate $J_{\text{obs}}$-optimal policies.

As in Case 1, we find that $J_{\text{obs}}([a_I a_H a_T]) > J_{\text{obs}}([a_I a_T])$ if and only if $p = 1$ or $p_H < \frac{5}{5+r}$. In case 2 we have assumed $p < \frac{1}{3}$, leaving only the $p_H$ condition.

If $p_H < \frac{5}{5+r}$, then RLHF selects $[a_I a_H a_T]$. As in Case 1, this is deceptive inflationrelative to $\pi^* = [a_I a_T]$, because

$$\overline{E}^+([a_I a_H a_T]) = (1-p)(1-p_H)(5+r) > 0 = \overline{E}^+(\pi^*).$$

If $p_H > \frac{5}{5+r}$, then RLHF selects $[a_I a_C a_T]$. Because this policy is not $J$-optimal, by Theorem 4.5, we must have deceptive inflation, overjustification, or both. Which is it? Here the optimal policy is to terminate after installing drivers, $[a_I a_T]$. However, $p_H > \frac{5}{5+r}$. This can be rewritten as $p_H(5 + r) > 5$. We have seen this expression $p_H(5 + r)$ before; it is the underestimation error incurred on $\vec{s} = SITT$ and therefore also the average underestimation error of policy $[a_I a_T]$. So here the underestimation error on the optimal policy—that is, the risk that the human misunderstands optimal behavior (terminating after installing driver) as undesired behavior (attempting a CUDA install that was unlikely to work and hiding the mistake)—is severe enough that the agent opts instead for $[a_I a_C a_T]$, a worse policy that attempts the ill-fated CUDA installation only to prove that it wasn't doing so secretly. In qualitative terms, this is quintessential overjustification behavior. Indeed, relative to reference policy $\pi^* = [a_I a_T]$, we have

$$\overline{E}^-([a_I a_C a_T]) = 0 < p_H(5 + r) = \overline{E}^-(\pi^*)$$
$$J([a_I a_C a_T]) = 11 - (1 - p) \cdot 15 < 1 = J(\pi^*),$$

and thus by Definition 4.4, this is overjustification.

### C.4 Ambiguity in Section 4.4 examples when modeling partial observability

Consider the example in Fig. 4A when modeling partial observability as in Section 5. By Theorem 5.2, the ambiguity in the return function leaving the choice probabilities invariant is given by $\ker \mathbf{B} \cap \text{im}\, \mathbf{\Gamma}$.

Table 13: Experiments showing improved performance of po-aware RLHF

| Ex. | $p$ | $p_{\text{hide}}$ | $p_{\text{default}}$ | model | action | $\overline{E}^+$ | dec. infl. | $\overline{E}^-$ | overj. | optimal |
|-----|-----|-------------------|----------------------|-------|--------|------------------|------------|------------------|--------|---------|
| A | 0.5 | 0.5 | N/A | naive | $a_H$ | 1.5 | ✓ | 0 | ✗ | ✗ |
| A | 0.5 | 0.5 | N/A | po-aware | $a_H$ | 1.5 | ✓ | 0 | ✗ | ✗ |
| A | 0.1 | 0.9 | N/A | naive | $a_C$ | 0 | ✗ | 0 | ✓ | ✗ |
| A | 0.1 | 0.9 | N/A | po-aware | $a_T$ | 0 | ✗ | 5.4 | ✗ | ✓ |
| B | 0.5 | N/A | 0.9 | naive | $a_T$ | 4.5 | ✓ | 0 | ✓ | ✗ |
| B | 0.5 | N/A | 0.9 | po-aware | $a_D$ | 0 | ✗ | 0.25 | ✗ | ✓ |
| B | 0.5 | N/A | 0.1 | naive | $a_V$ | 0 | ✗ | 0 | ✓ | ✗ |
| B | 0.5 | N/A | 0.1 | po-aware | $a_D$ | 0 | ✗ | 2.25 | ✗ | ✓ |

Let $R' = (0, 0, R'(W), 0, R'(W_H), 0, 0) \in \mathbb{R}^{\{S,I,W,L,W_H,L_H,T\}}$ be a reward function that we want to parameterize such that $G' := \mathbf{\Gamma} \cdot R'$ ends up in the ambiguity; here, $R'$ is interpreted as a column vector.

We want $\mathbf{B} \cdot G' = 0$. Since the observation sequences $\vec{o} = o_\emptyset o_\emptyset o_\emptyset o_\emptyset$, $\vec{o} = o_\emptyset o_L o_\emptyset o_\emptyset$, $\vec{o} = o_\emptyset o_I o_\emptyset o_\emptyset$, or $\vec{o} = o_\emptyset o_I o_L o_\emptyset$ all cannot involve the states $W$ or $W_H$, it is clear that they have zero expected return $(\mathbf{B} \cdot G')(\vec{o})$. Set $p'_H := B\big(SIW_H T \mid o_\emptyset o_I o_W o_\emptyset\big)$. Then the condition that $\mathbf{B} \cdot G' = 0$ is equivalent to:

$$0 = \big(\mathbf{B} \cdot G'\big)(o_\emptyset o_I o_W o_\emptyset) = \mathop{\mathbf{E}}_{\vec{s} \sim B(\vec{s} \mid o_\emptyset o_I o_W o_\emptyset)} \big[G'(\vec{s})\big]$$

$$= p'_H \cdot G'(SIW_H T) + (1 - p'_H) \cdot G'(SIWT) = p'_H \cdot R'(W_H) + (1 - p'_H) \cdot R'(W).$$

Thus, if $R'(W) = \frac{p'_H}{p'_H - 1} R'(W_H)$, then $G' \in \ker \mathbf{B} \cap \operatorname{im} \mathbf{\Gamma}$, meaning that $R + R'$ has the same choice probabilities as $R$ and is thus fully feedback-compatible. In particular, if $R'(W_H) \gg 0$ is sufficiently large, then in subsequent policy optimization, there is an incentive to hide the mistakes and $\pi_H$ will be selected, which is suboptimal with respect to the true reward function $R$.

Thus Fig. 4A *still retains dangerous ambiguity when modeling partial observability.*

However, the example in Fig. 4B leads to no ambiguity when partial observability is correctly modeled.

To show this in detail, let $G' = \mathbf{\Gamma}(R') \in \ker \mathbf{B} \cap \operatorname{im} \mathbf{\Gamma}$. We need to show $G' = 0$. Since the human is only uncertain about the state sequences corresponding to the observation sequence $o_\emptyset o_I o_\emptyset o_\emptyset$, the condition $\mathbf{B} \cdot G' = 0$ already implies $G'(\vec{s}) = 0$ for all state sequences except $SIWT$ and $SITT$. From $(\mathbf{B} \cdot G')(o_\emptyset o_I o_\emptyset o_\emptyset) = 0$, one then obtains the equation

$$(1 - p_D) \cdot \big(R'(S) + R'(I) + 2R'(T)\big) + p_D \cdot \big(R'(S) + R'(I) + R'(W) + R'(T)\big) = 0. \quad (5)$$

Thus, if one of the two state sequences involved has zero return, then the other has as well, assuming that $0 \neq p_D \neq 1$, and we are done.

To show this, we use that all other state sequences have zero return: $R'(S) + 3R'(T) = 0 = R'(S) + R'(L) + 2R'(T)$, from which $R'(L) = R'(T)$ follows. Then, from $R'(S) + R'(I) + R'(L) + R'(T) = 0$, substituting the previous result gives $R'(S) + R'(I) + 2R'(T) = 0$, and so Equation (5) results in $R'(S) + R'(I) + R'(W) + R'(T) = 0$. Overall, this shows $G' = \mathbf{\Gamma}(R') = 0$, and so $\ker \mathbf{B} \cap \operatorname{im} \mathbf{\Gamma} = \{0\}$.

## C.5 Experimental details

Here, we explain more experimental details for the results in Table 1, reproduced here as Table 13.

The leftmost column ("Ex." for "example") corresponds to Examples A and B in Figure 4. $p$ is the success probability upon attempting to install Cuda or NumPy in state $I$, see Figure 6. $p_{\text{hide}}$ in Example A is the human's belief probability that the agent hid the error message if there is no output after nvidia-driver installation. Similarly, $p_{\text{default}}$ in Example B is the human's belief probability that installation was done with default settings if there is no further output after Python

installation. Note that lines one and two in the table also correspond to Example C.1, lines three and four to Example C.2, and lines five and six to the first half and seven and eight to the second half of Example C.3, respectively. In all our experiments, we set the penalty to $r = 1$.

The "model" column has value "naive" if the reward learning algorithm is classical RLHF (erroneously assuming full observability) as in Christiano et al. [2017], and "po-aware" if the human's partial observability is correctly modeled as in Appendix D.3. We initialize the reward function as a list of rewards of states and train it by logistic regression using a dataset that consists of all pairs of state sequences together with the human's choice probabilities under partial observations. This leads to 28 pairs of distinct trajectories together with choice probabilities. We train the reward model for 300 epochs over a shuffled dataset of 13.5 copies of the 28 pairs with the Adam optimizer, for a total of 113400 training updates.

Once we have the resulting reward model, we use value iteration to find its deterministic optimal policy. All policies choose to install the nvidia-driver (in Example A) and Python (in Example B), and differ in their action in state $I$, which is given in the column "action". We compute the overestimation error and underestimation error of the resulting policies analytically using the hardcoded environment dynamics, true reward function, observation function, and human belief matrix $\mathbf{B}$. This is given in columns $\overline{E}^+$ and $\overline{E}^-$. Note that these are averages over 10 entire training runs, though since they always result in the same learned policy, there is no variation and we do not state any uncertainty.

The columns "dec. infl.", "overj.", and "optimal" state whether deceptive inflation or overjustification occurs with the learned policy, and whether it is optimal according to the true human's reward function.

# D    Modeling the Human in Partially Observable RLHF

In this appendix, we develop the theory of RLHF with appropriately modeled partial observability, including full proofs of all theorems.

In Section D.1, we explain how the human can arrive at the belief $B(\vec{s} \mid \vec{o})$ via Bayesian updates. The main theory and the main paper in general do not depend on this specific form of the human's belief, but some examples in the appendix do.

In Section D.2 we then explain our main result: the ambiguity and identifiability of both reward and return functions under observed sequence comparisons. In Section D.3, we then explain that this theorem means that one could *in principle* design a practical reward learning algorithm that converges on the correct reward function up to the ambiguity characterized in the section before, *if* the human's belief kernel $B(\vec{s} \mid \vec{o})$ is fully known.

In Section D.4, we generalize the theory to the case that the human's observations are not necessarily known to the learning system and again characterize precisely when the return function is identifiable from sequence comparisons. We then consider special cases in Section D.5, where we show that the fully observable case is covered by our theory, that a deterministic observation kernel $P_{\vec{O}}$ usually leads to non-injective belief matrix $\mathbf{B}$, and that "noise" in the observation kernel $P_{\vec{O}}$ leads, under appropriate assumptions, to the identifiability of the return function.

Our identifiability results require that the learning system knows the human's belief kernel $B(\vec{s} \mid \vec{o})$. In Section D.6, we then show that these results are robust to slight misspecifications: a bound in the error in the specified belief leads to a corresponding bound in the error of the policy evaluation function used for subsequent reinforcement learning.

In Section D.7, we then provide a very preliminary characterization of the ambiguity in the return function under special cases.

Finally, in Section D.8, we study examples of identifiability and non-identifiability of the return function for the case that we *do* model the human's partial observability correctly. This reveals qualitatively interesting cases of identifiability, even when $\mathbf{B}$ is not injective, and catastrophic cases of non-identifiability.

### D.1 The Belief over the State Sequence for Rational Humans

Before we dive into the main theory, we want to explain how the human can iteratively compute the posterior of the state sequence given an observation sequence with successively new observations. This is done by defining a Bayesian network for the joint probability of policy, states, actions, and observations, and doing Bayesian inference over this Bayesian network.

The details of this subsection are only relevant for a few sections in the appendix since it is usually enough to assume that the posterior belief *exists*. Additionally, in the core theory, we do not even assume that $B(\vec{s} \mid \vec{o})$ is a posterior: it is simply any probability distribution. The reason why it can still be interesting to analyze the case when the human is a rational Bayesian reasoner is that one can then analyze RLHF under *generous* assumptions to the human.

We model the human to have a joint distribution $B(\pi, \vec{s}, \vec{a}, \vec{o})$ over the policy $\pi$, state sequence $\vec{s} = s_0, \ldots, s_T$, action sequence $\vec{a} = a_0, \ldots, a_{T-1}$, and observation sequence $\vec{o} = o_0, \ldots, o_T$. This is given by a Bayesian network with the following components:

- a policy prior $B(\pi')$;
- the probability of the initial state $B(s_0) := P_0(s_0)$;
- action probabilities $B(a \mid s, \pi) := \pi(a \mid s)$;
- transition probabilities $B(s_{t+1} \mid s_t, a_t) := \mathcal{T}(s_{t+1} \mid s_t, a_t)$;
- and observation probabilities $B(o_t \mid s_t) := P_O(o_t \mid s_t)$.

Together, this defines the joint distribution $B(\pi, \vec{s}, \vec{a}, \vec{o})$ over the policy, states, actions, and observations that factorizes according to the following directed acyclic graph:

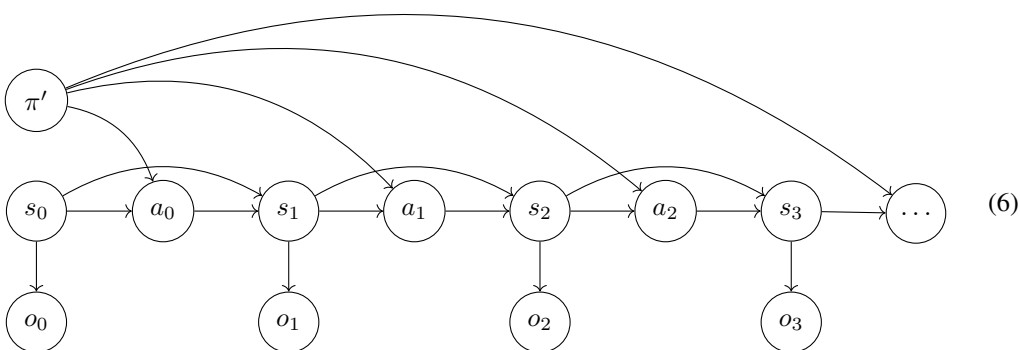

$$(6)$$

The following proposition clarifies the iterative Bayesian update of the human's posterior over state sequences, given observation sequences:

**Proposition D.1.** *Let $t \leq T - 1$ and denote by $\hat{s} = s_0, \ldots, s_t$ a state sequence* segment *of length $t \geq 0$. Similarly, $\hat{o} = o_0, \ldots, o_t$ denotes an observation sequence segment. We have*

$$B(\hat{s}, s_{t+1}, \pi \mid \hat{o}, o_{t+1}) \propto P_O(o_{t+1} \mid s_{t+1}) \cdot \left[ \sum_{a_t \in \mathcal{A}} \mathcal{T}(s_{t+1} \mid \hat{s}_t, a_t) \cdot \pi(a_t \mid s_t) \right] \cdot B(\hat{s}, \pi \mid \hat{o}).$$

*Thus, the human can iteratively compute $B(\hat{s}, \pi \mid \hat{o})$ from the prior $B(s_0, \pi) = P_0(s_0) \cdot B(\pi')$ using the above Bayesian update.*

*The posterior over the state sequence can subsequently be computed by*

$$B(\hat{s} \mid \hat{o}) = \int_\pi B(\hat{s}, \pi \mid \hat{o}).$$

*Proof.* The proof is essentially just Bayes rule applied to the Bayesian network in Equation (6). We repeatedly make use of conditional independences that follow from d-separations in the graph [Geiger et al., 1990]. More concretely, we have

$$B\big(\hat{s}, s_{t+1}, \pi \mid \hat{o}, o_{t+1}\big) \propto B\big(o_{t+1} \mid \hat{s}, s_{t+1}, \pi, \hat{o}\big) \cdot B\big(\hat{s}, s_{t+1}, \pi \mid \hat{o}\big)$$

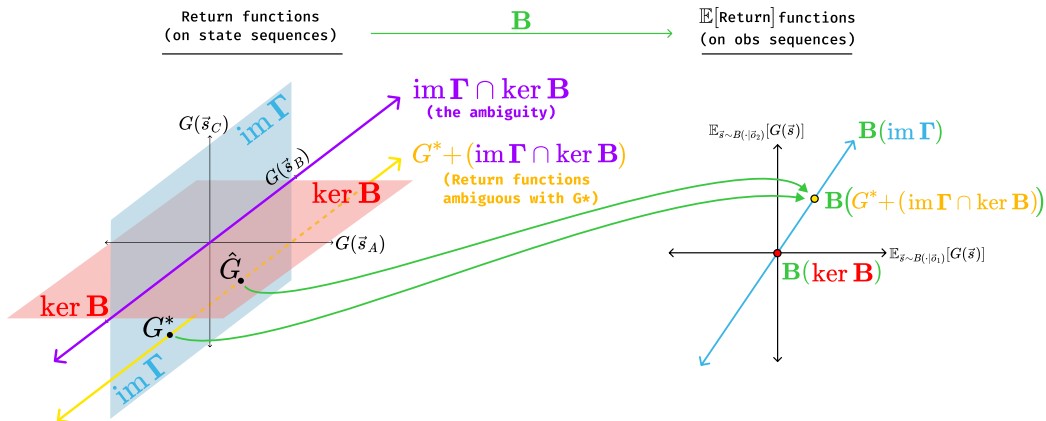

Figure 8: The linear geometry of ambiguity for a hypothetical example with three state sequences and two observation sequences. $G^*$ is the true return function, and "$G$" is used in labeling the axes to refer to some arbitrary return function. This is a more accurate geometric depiction of the middle and right spaces in Figure 5. The subspace $\operatorname{im}\mathbf{\Gamma}\cap\ker\mathbf{B}$ (purple) is the ambiguity in return functions, meaning that adding an element would not change the human's expected return function on observations. Thus the set of return functions that the reward learning system can infer is the affine set $G+(\operatorname{im}\mathbf{\Gamma}\cap\ker\mathbf{B})$ (yellow). Note that the planes on the left are drawn to be axis-aligned for ease of visualization; this will not be the case for real MDPs.

$$= P_O\big(o_{t+1}\mid s_{t+1}\big)\cdot B\big(s_{t+1}\mid \hat{s},\pi,\hat{o}\big)\cdot B\big(\hat{s},\pi\mid \hat{o}\big)$$

$$= P_O\big(o_{t+1}\mid s_{t+1}\big)\cdot\left[\sum_{a_t\in\mathcal{A}} B\big(s_{t+1}\mid a_t,\hat{s},\pi,\hat{o}\big)\cdot B\big(a_t\mid \hat{s},\pi,\hat{o}\big)\right]\cdot B\big(\hat{s},\pi\mid \hat{o}\big)$$

$$= P_O\big(o_{t+1}\mid s_{t+1}\big)\cdot\left[\sum_{a_t\in\mathcal{A}} \mathcal{T}\big(s_{t+1}\mid s_t,a_t\big)\cdot\pi\big(a_t\mid s_t\big)\right]\cdot B\big(\hat{s},\pi\mid \hat{o}\big).$$

In step 1, we used Bayes rule. In step 2, we made use of the independence $o_{t+1}\perp\!\!\!\perp(\hat{s},\pi,\hat{o})\mid s_{t+1}$, plugged in the observation kernel, and used the chain rule of probability to compose the second term into a product. In step 3, we marginalized and used, once again, the chain rule of probability. In step 4, we used the independences $s_{t+1}\perp\!\!\!\perp(s_0,\ldots,s_{t-1},\pi,\hat{o})\mid(s_t,a)$ and $a_t\perp\!\!\!\perp(s_0,\ldots,s_{t-1},\hat{o})\mid(\pi,s_t)$ and plugged in the transition kernel and the policy.

The last formula is just a marginalization over the policy. $\qquad\square$

## D.2 Ambiguity and Identifiability of Reward and Return Functions under Observation Sequence Comparisons

In this section, we prove the main theorem of this paper: a characterization of the ambiguity that is left in the reward and return function once the human's Boltzmann-rational choice probabilities are known. We change the formulation slightly by formulating the linear operators "intrinsically" in the spaces they are defined in, instead of using matrix versions. This does not change the general picture, but is a more natural setting when thinking, e.g., about generalizing the results to infinite state sequences. Thus, we define $\mathbf{B}:\mathbb{R}^{\vec{\mathcal{S}}}\to\mathbb{R}^{\vec{\Omega}}$ as the linear operator given by

$$\big[\mathbf{B}(G)\big](\vec{o}):=\operatorname*{\mathbf{E}}_{\vec{s}\sim B(\vec{s}\mid\vec{o})}\big[G(\vec{s})\big].$$

Here, $\mathbf{B}$ is the human's belief, which can either be computed as in the previous subsection or simply be any conditional probability distribution. Similarly, we define $\mathbf{\Gamma}:\mathbb{R}^{\mathcal{S}}\to\mathbb{R}^{\vec{\mathcal{S}}}$ as the linear operator given by

$$\big[\mathbf{\Gamma}(R)\big](\vec{s}):=\sum_{t=0}^{T}\gamma^t R(s_t).$$

The matrix product $\mathbf{B} \cdot \mathbf{\Gamma}$ then becomes the composition $\mathbf{B} \circ \mathbf{\Gamma} : \mathbb{R}^{\mathcal{S}} \to \mathbb{R}^{\vec{\Omega}}$. Finally, recall that the kernel $\ker \mathbf{A}$ of a linear operator $\mathbf{A}$ is defined as its nullspace, and the image $\operatorname{im} \mathbf{A}$ as the set of elements hit by $\mathbf{A}$. We obtain the following theorem:

**Theorem D.2.** *Let $R$ be the true reward function and $\tilde{R}$ another reward function. Let $\tilde{G} = \mathbf{\Gamma}(\tilde{R})$ and $G = \mathbf{\Gamma}(R)$ be the corresponding return functions. The following three statements are equivalent:*

(i) *The reward function $\tilde{R}$ gives rise to the same vector of choice probabilities as $R$, i.e*

$$\left( P^{\tilde{R}}(\vec{o} \succ \vec{o}') \right)_{\vec{o},\vec{o}' \in \vec{\Omega}} = \left( P^{R}(\vec{o} \succ \vec{o}') \right)_{\vec{o},\vec{o}' \in \vec{\Omega}}.$$

(ii) *There is a reward function $R' \in \ker(\mathbf{B} \circ \mathbf{\Gamma})$ and a constant $c \in \mathbb{R}$ such that*

$$\tilde{R} = R + R' + c.$$

(iii) *There is a return function $G' \in \ker \mathbf{B} \cap \operatorname{im} \mathbf{\Gamma}$ and a constant $c' \in \mathbb{R}$ such that*

$$\tilde{G} = G + G' + c'.$$

*In other words, the* ambiguity *that is left in the* reward function *when its observation-based choice probabilities are known is, up to an additive constant, given by* $\ker(\mathbf{B} \circ \mathbf{\Gamma})$*; the ambiguity left in the* return function *is given by* $\ker \mathbf{B} \cap \operatorname{im} \mathbf{\Gamma}$*.*

*Proof.* Assume (i). To prove (ii), let $\sigma$ by the sigmoid function given by $\sigma(x) = \frac{1}{1+\exp(-x)}$. Then by Equation (2), the equality of choice probabilities means the following for all $\vec{o}, \vec{o}' \in \vec{\Omega}$:

$$\sigma\Big( \beta \cdot \big( \big[ \mathbf{B}(\tilde{G}) \big](\vec{o}) - \big[ \mathbf{B}(\tilde{G}) \big](\vec{o}') \big) \Big) = \sigma\Big( \beta \cdot \big( \big[ \mathbf{B}(G) \big](\vec{o}) - \big[ \mathbf{B}(G) \big](\vec{o}') \big) \Big).$$

Since the sigmoid function is injective, this implies

$$\big[ \mathbf{B}(\tilde{G}) \big](\vec{o}) - \big[ \mathbf{B}(\tilde{G}) \big](\vec{o}') = \big[ \mathbf{B}(G) \big](\vec{o}) - \big[ \mathbf{B}(G) \big](\vec{o}').$$

Fixing an arbitrary $\vec{o}'$, this implies that there exists a constant $c'$ such that for all $\vec{o} \in \vec{\Omega}$, the following holds:

$$\big[ \mathbf{B}(\tilde{G}) \big](\vec{o}) - \big[ \mathbf{B}(G) \big](\vec{o}') - c' = 0.$$

Noting that $\mathbf{B}(c') = c'$, this implies $\tilde{G} - G - c' \in \ker(\mathbf{B})$. Now, define the constant reward function

$$c := c' \cdot \frac{1 - \gamma}{1 - \gamma^{T+1}}.$$

We obtain

$$\big[ \mathbf{\Gamma}(c) \big](\vec{s}) = \sum_{t=0}^{T} \gamma^t \cdot c$$

$$= c' \cdot \frac{1 - \gamma}{1 - \gamma^{T+1}} \cdot \sum_{t=0}^{T} \gamma^t$$

$$= c'.$$

Thus, we have

$$\mathbf{\Gamma}(\tilde{R} - R - c) = \tilde{G} - G - c' \in \ker(\mathbf{B}),$$

implying $R' := \tilde{R} - R - c \in \ker(\mathbf{B} \circ \mathbf{\Gamma})$. This shows (ii).

That (ii) implies (iii) follows by applying $\mathbf{\Gamma}$ to both sides of the equation.

Now assume (iii), i.e. $\tilde{G} = G + G' + c'$ for a constant $c' \in \mathbb{R}$ and a return function $G' \in \ker(\mathbf{B}) \cap \operatorname{im} \mathbf{\Gamma}$. This implies $\mathbf{B}(\tilde{G}) = \mathbf{B}(G) + c'$. Thus, for all $\vec{o}, \vec{o}' \in \vec{\Omega}$, we have

$$\big[ \mathbf{B}(\tilde{G}) \big](\vec{o}) - \big[ \mathbf{B}(\tilde{G}) \big](\vec{o}') = \big[ \mathbf{B}(G) \big](\vec{o}) - \big[ \mathbf{B}(G) \big](\vec{o}'),$$

which implies the equal choice probabilities after multiplying with $\beta$ and applying the sigmoid function $\sigma$ on both sides. Thus, (iii) implies (i). $\qquad\square$

**Corollary D.3.** *The following two statements are equivalent:*

*(i)* $\ker(\mathbf{B} \circ \mathbf{\Gamma}) = 0.$

*(ii) The data* $\left( P^R \!\left( \vec{o} \succ \vec{o}' \right) \right)_{\vec{o}, \vec{o}' \in \vec{\Omega}}$ *determine the reward function $R$ up to an additive constant.*

*Proof.* That (i) implies (ii) follows immediately from the implication from (i) to (ii) within the preceding theorem.

Now assume (ii). Let $R' \in \ker(\mathbf{B} \circ \mathbf{\Gamma})$. Define $\tilde{R} := R + R'$. Then the implication from (ii) to (i) within the preceding theorem implies that $\tilde{R}$ and $R$ have the same choice probabilities. Thus, the assumption (ii) in this corollary implies that $R'$ is a constant. Since $\mathbf{\Gamma}$ and $\mathbf{B}$ map nonzero constants to nonzero constants, the fact that $R' \in \ker(\mathbf{B} \circ \mathbf{\Gamma})$ implies that $R' = 0$, showing that $\ker(\mathbf{B} \circ \mathbf{\Gamma}) = \{0\}$. $\qquad\square$

As mentioned in the main paper, the previous result already leads to the non-identifiability of $R$ whenever $\mathbf{\Gamma}$ is not injective, corresponding to the presence of zero-initial potential shaping (Skalse et al. [2023], Lemma B.3). Thus, we now strengthen the previous result so that it deals with the identifiability of the *return* function, which is sufficient for the purpose of policy optimization:

**Corollary D.4.** *Consider the following four statements (which can each be true or false):*

*(i)* $\ker \mathbf{B} = \{0\}.$

*(ii)* $\ker \left( \mathbf{B} \circ \mathbf{\Gamma} \right) = \{0\}.$

*(iii)* $\ker \mathbf{B} \cap \operatorname{im} \mathbf{\Gamma} = \{0\}.$

*(iv) The data* $\left( P^R \!\left( \vec{o} \succ \vec{o}' \right) \right)_{\vec{o}, \vec{o}' \in \vec{\Omega}}$ *determine the return function $G = \mathbf{\Gamma}(R)$ on sequences $\vec{s} \in \vec{\mathcal{S}}$ up to a constant independent of $\vec{s}$.*

*Then the following implications, and no other implications, are true:*

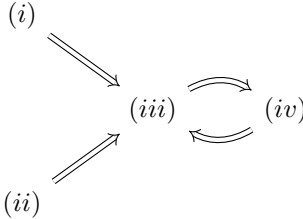

*In particular, all of (i), (ii), and (iii) are sufficient conditions for identifying the return function from the choice probabilities.*

*Proof.* That (i) implies (iii) is trivial. That (ii) implies (iii) is a simple linear algebra fact: Assume (ii) and that $G' \in \ker \mathbf{B} \cap \operatorname{im} \mathbf{\Gamma}$. Then $G' = \mathbf{\Gamma}(R')$ for some $R' \in \mathbb{R}^{\vec{\mathcal{S}}}$ and

$$0 = \mathbf{B}(G') = \mathbf{B}\left( \mathbf{\Gamma}(R') \right) = (\mathbf{B} \circ \mathbf{\Gamma})(R').$$

By (ii), this implies $R' = 0$ and therefore $G' = \mathbf{\Gamma}(R') = 0$, showing (iii).

That (iii) implies (iv) immediately follows from the implication from (i) to (iii) in Theorem D.2.

Now, assume (iv). To prove (iii), assume $G' \in \ker \mathbf{B} \cap \operatorname{im} \mathbf{\Gamma}$. Then the implication from (iii) to (i) in Theorem D.2 implies that $G + G'$ induces the same observation-based choice probabilities as $G$. Thus, (iv) implies $G + G' = G + c'$ for some constant $c'$, which implies $G' = c'$. Since $G' \in \ker \mathbf{B}$, this implies $0 = \mathbf{B}(G') = \mathbf{B}(c') = c'$ and thus $G' = 0$. Thus, we showed $\ker \mathbf{B} \cap \operatorname{im} \mathbf{\Gamma} = \{0\}$.

We now show that no other implication holds in general. Example D.32 will show that (ii) does not imply (i). We now show that (i) does also not imply (ii), from which it will logically follow that (iii) does neither imply (i) nor (ii). Namely, consider the following simple MDP with time horizon $T = 1$:

$$a \longrightarrow b \tag{7}$$

In this MDP, every state sequence starts in $a$, deterministically transitions to $b$, and then ends. This means that $\vec{s} = ab$ is the only sequence. Now, let $R' \in \mathbb{R}^{\{a,b\}}$ be the reward function given by

$$R'(a) = 1, \quad R'(b) = \frac{-1}{\gamma}.$$

We obtain

$$\left[\, \mathbf{\Gamma}(R') \,\right](\vec{s}) = R'(a) + \gamma R'(b) = 1 + \gamma \cdot \frac{-1}{\gamma} = 0.$$

Thus, $\mathbf{\Gamma}(R') = 0$, $(\mathbf{B} \circ \mathbf{\Gamma})(R') = 0$, and, therefore, $\ker\left(\mathbf{B} \circ \mathbf{\Gamma}\right) \neq \{0\}$. Thus, (ii) does not hold. However, it is possible to choose $B(\vec{s} \mid \vec{o})$ such that (i) holds: e.g., if $\Omega = \mathcal{S}$ and $B(\vec{s} \mid \vec{o}) := \delta_{\vec{o}}(\vec{s})$, then $\ker \mathbf{B} = \{0\}$ since this operator is the identity. $\qquad\square$

### D.3   The Ambiguity in Reward Learning in Practice

In this section, we point out that Theorem D.2 is not just a theoretical discussion: When $\mathbf{B}$ and the inverse temperature parameter $\beta$ are known, then it is possible to design a reward learning algorithm that learns the true reward function up to the ambiguity $\ker(\mathbf{B} \circ \mathbf{\Gamma})$ in the infinite data limit. In doing so, we essentially use the loss function proposed in Christiano et al. [2017].

Namely, assume $\mathcal{D}$ is a data distribution of observation sequences $\vec{o} \in \vec{\Omega}$ such that all sequences in $\vec{\Omega}$ have a strictly positive probability of being sampled; for example, $\mathcal{D}$ could use an exploration policy and the observation sequence kernel $P_{\vec{o}}$. For each pair of observation sequences $(\vec{o}, \vec{o}')$, we then get a conditional distribution $P(\mu \mid \vec{o}, \vec{o}')$ over a one-hot encoded human choice $\mu \in \{(1,0), (0,1)\}$, with probability

$$P\big(\mu = (1,0) \mid \vec{o}, \vec{o}'\big) = P^R\big(\vec{o} \succ \vec{o}'\big).$$

Together, this gives rise to a dataset $(\vec{o}_1, \vec{o}_1', \mu_1), \ldots, (\vec{o}_N, \vec{o}_N', \mu_N)$ of observation sequences plus a human choice.

Now assume we learn a reward function $R_\theta : \mathcal{S} \to \mathbb{R}$ that is differentiable in the parameter $\theta$ and that can represent all possible reward functions $R \in \mathbb{R}^{\mathcal{S}}$. Let $G_\theta := \mathbf{\Gamma}(R_\theta)$ be the corresponding return function. Write $\mu_k = (\mu_k^{(1)}, \mu_k^{(2)})$. As in Christiano et al. [2017], we define its loss over the dataset above by

$$\widetilde{\mathcal{L}}(\theta) = -\frac{1}{N} \sum_{k=1}^{N} \mu_k^{(1)} \cdot \log P^{R_\theta}\big(\vec{o}_k \succ \vec{o}_k'\big) + \mu_k^{(2)} \cdot \log P^{R_\theta}\big(\vec{o}_k' \succ \vec{o}_k\big).$$

Note that by Equation (2), this loss function essentially uses $\mathbf{B}$ and also the inverse temperature parameter $\beta$ in its definition. This means that these need to be explicitly represented to be able to use the loss function in practice.

**Proposition D.5.** *The loss function $\widetilde{\mathcal{L}}$ is differentiable. Furthermore, in the infinite datalimit its minima are precisely given by parameters $\theta$ such that $R_\theta = R + R' + c$ for $R' \in \ker\left(\mathbf{B} \circ \mathbf{\Gamma}\right)$ and $c \in \mathbb{R}$, or equivalently $G_\theta = G + G' + c'$ for $G' \in \ker \mathbf{B} \cap \operatorname{im} \mathbf{\Gamma}$ and $c' \in \mathbb{R}$.*

*Proof.* The differentiability of the loss function follows from the differentiability of multiplication with the matrix $\mathbf{B}$, see Equation (2), and of the reward function $R_\theta$ in its parameter $\theta$ that we assumed.

For the second statement, let $N(\vec{o}, \vec{o}')$ be the number of times that the pair $(\vec{o}, \vec{o}')$ appears in the dataset, and let $N(\vec{o}, \vec{o}', 1)$ be the number of times that the human choice is $\mu = (1,0)$ and the sampled pair is $(\vec{o}, \vec{o}')$, and similar for 2 instead of 1. We obtain

$$\widetilde{\mathcal{L}}(\theta) = -\sum_{\vec{o}, \vec{o}' \in \vec{\Omega}} \frac{N(\vec{o}, \vec{o}')}{N} \cdot \left[ \frac{N(\vec{o}, \vec{o}', 1)}{N(\vec{o}, \vec{o}')} \log P^{R_\theta}\big(\vec{o} \succ \vec{o}'\big) \right.$$

$$+ \frac{N(\vec{o}, \vec{o}', 2)}{N(\vec{o}, \vec{o}')} \log P^{R_\theta}(\vec{o}' \succ \vec{o}) \Bigg]$$

$$\approx \mathop{\mathbf{E}}_{\vec{o}, \vec{o}' \sim \mathcal{D}} \left[ \mathrm{CE}\left( P^R(\vec{o} \precsim \vec{o}') \,\|\, P^{R_\theta}(\vec{o} \precsim \vec{o}') \right) \right]$$

$$=: \mathcal{L}(\theta).$$

Here, CE is the crossentropy between the two binary distributions. Since we assumed that $\mathcal{D}$ gives a positive probability to all observation sequences in $\vec{\Omega}$, and since the cross entropy is generally minimized exactly when the second distribution equals the first, the loss function $\mathcal{L}(\theta)$ is minimized if and only if $R_\theta$ gives rise to the same choice probabilities as $R$ for all pairs of observation sequences. Theorem D.2 then gives the result. □

### D.4 Identifiability of Return Functions When Human Observations Are Not Known

Corollary D.4 assumes that the choice probabilities of each observation sequence pair are known to the reward learning algorithm. However, this requires the algorithm to know what the human observed. In some applications, this is a reasonable assumption, e.g. if the human's observations are themselves produced by an algorithm that can feed the observations also back to the learning algorithm. In general, however, the observations happen in the physical world, and are only known probabilistically via the observation kernel $P_O$. The learning system *does* however have access to the full state sequences that generate the observation sequences. This leads to knowledge of the following choice probabilities for $\vec{s}, \vec{s}' \in \vec{\mathcal{S}}$:

$$P^R(\vec{s} \succ \vec{s}') := \mathop{\mathbf{E}}_{\vec{o}, \vec{o}' \sim P_{\vec{O}}(\cdot | \vec{s}, \vec{s}')} \left[ P^R(\vec{o} \succ \vec{o}') \right], {}^{[2]} \tag{8}$$

where the observation-based choice probabilities are given as in Equation (2). In other words, the learning algorithm can only infer an aggregate of the observation-based choice probabilities. Again, we can ask a question similar to the ones before, extending the investigations in the previous section:

**Question D.6.** *Assume the vector of choice probabilities* $\left( P^R(\vec{s} \succ \vec{s}') \right)_{\vec{s}, \vec{s}' \in \vec{\mathcal{S}}}$ *is known. Additionally, assume that it is known that the human's observations are governed by* $P_O$, *and that the human is Boltzmann rational with inverse temperature parameter* $\beta$ *and beliefs* $B(\vec{s} \mid \vec{o})$, *see Equation* (8). *Does this data identify the return function* $G : \vec{\mathcal{S}} \to \mathbb{R}$?

If the observation-based choice probabilities from Equation (2) would be known, then Corollary D.4 would provide the answer to this question. Thus, similar to how we previously inverted the belief operator $\mathbf{B}$, we are now simply tasked with inverting the expectation over observation sequences. This leads us to the following definition:

**Definition D.7** (Ungrounding Operator). *The ungrounding operators* $\mathbf{O} : \mathbb{R}^{\vec{\Omega}} \to \mathbb{R}^{\vec{\mathcal{S}}}$ *and* $\mathbf{O} \otimes \mathbf{O} : \mathbb{R}^{\vec{\Omega} \times \vec{\Omega}} \to \mathbb{R}^{\vec{\mathcal{S}} \times \vec{\mathcal{S}}}$ *are defined by*

$$\left[ \mathbf{O}(v) \right](\vec{s}) := \mathop{\mathbf{E}}_{\vec{o} \sim P_{\vec{O}}(\vec{o} | \vec{s})} \left[ v(\vec{o}) \right], \quad \left[ (\mathbf{O} \otimes \mathbf{O})(C) \right](\vec{s}, \vec{s}') := \mathop{\mathbf{E}}_{\vec{o}, \vec{o}' \sim P_{\vec{O}}(\cdot | \vec{s}, \vec{s}')} \left[ C(\vec{o}, \vec{o}') \right].$$

*Here,* $v \in \mathbb{R}^{\vec{\Omega}}$ *is an arbitrary vector, and* $C \in \mathbb{R}^{\vec{\Omega} \times \vec{\Omega}}$ *is also an arbitrary vector, where the notation can remind of "Choice" since the inputs to* $\mathbf{O} \otimes \mathbf{O}$ *are, in practice, vectors of observation-based Boltzmann-rational choice probabilities.*

Formally, $\mathbf{O} \otimes \mathbf{O}$ is the Kronecker product of $\mathbf{O}$ with itself, but it is not necessary to understand this fact to follow the discussion. Ultimately, to be able to recover the observation-based choice probabilities, what matters is that $\mathbf{O} \otimes \mathbf{O}$ is injective on whole vectors of these choice probabilities. The injectivity of $\mathbf{O}$ is a *sufficient condition* for this, which explains its usefulness. We show this in the following lemma:

**Lemma D.8.** $\mathbf{O} : \mathbb{R}^{\vec{\Omega}} \to \mathbb{R}^{\vec{\mathcal{S}}}$ *is injective if and only if* $\mathbf{O} \otimes \mathbf{O} : \mathbb{R}^{\vec{\Omega} \times \vec{\Omega}} \to \mathbb{R}^{\vec{\mathcal{S}} \times \vec{\mathcal{S}}}$ *is injective.*

---

[2]We excuse the following abuse of notation: these choice probabilities *run through* the observations of the human and are not the same as the choice probabilities from Equation (1).

*Proof.* This is a general property of the Kronecker product of a linear operator with itself. For completeness, we demonstrate the calculation in our special case. First, assume that $\mathbf{O}$ is injective. Assume that $(\mathbf{O} \otimes \mathbf{O})(C) = 0$ for some $C \in \mathbb{R}^{\vec{\Omega} \times \vec{\Omega}}$. We need to show $C = 0$.

For all pairs of state sequences $(\vec{s}, \vec{s}')$, we have

$$
\begin{aligned}
0 = \left[ (\mathbf{O} \otimes \mathbf{O})(C) \right](\vec{s}, \vec{s}') &= \mathop{\mathbf{E}}_{\vec{o}, \vec{o}' \sim P_{\vec{O}}(\cdot | \vec{s}, \vec{s}')} \left[ C(\vec{o}, \vec{o}') \right] \\
&= \mathop{\mathbf{E}}_{\vec{o} \sim P_{\vec{O}}(\vec{o} | \vec{s})} \left[ \mathop{\mathbf{E}}_{\vec{o}' \sim P_{\vec{O}}(\vec{o}' | \vec{s}')} \left[ C(\vec{o}, \vec{o}') \right] \right] \\
&= \mathop{\mathbf{E}}_{\vec{o} \sim P_{\vec{O}}(\vec{o} | \vec{s})} \left[ C'_{\vec{s}'}(\vec{o}) \right] \\
&= \left[ \mathbf{O} \left( C'_{\vec{s}'} \right) \right](\vec{s}),
\end{aligned}
$$

where $C'_{\vec{s}'}(\vec{o}) := \mathbf{E}_{\vec{o}' \sim P_{\vec{O}}(\vec{o}' | \vec{s}')} \left[ C(\vec{o}, \vec{o}') \right]$. By the injectivity of $\mathbf{O}$, we obtain $C'_{\vec{s}'} = 0$ for all $\vec{s}'$. This means that for all $\vec{s}'$ and $\vec{o}$, we have

$$
0 = C'_{\vec{s}'}(\vec{o}) = \mathop{\mathbf{E}}_{\vec{o}' \sim P_{\vec{O}}(\vec{o}' | \vec{s}')} \left[ C(\vec{o}, \vec{o}') \right] = \left[ \mathbf{O} \left( C''_{\vec{o}} \right) \right](\vec{s}'),
$$

where $C''_{\vec{o}}(\vec{o}') := C(\vec{o}, \vec{o}')$. Again, by the injectivity of $\mathbf{O}$, we obtain $C''_{\vec{o}} = 0$ for all $\vec{o}$, leading to $C = 0$. That proves the direction from left to right.

To prove the other direction, assume that $\mathbf{O}$ is *not* injective. This means there exists $0 \neq C \in \mathbb{R}^{\vec{\Omega}}$ such that $\mathbf{O}(C) = 0$. Define $C \otimes C \in \mathbb{R}^{\vec{\Omega} \times \vec{\Omega}}$ by

$$
(C \otimes C)(\vec{o}, \vec{o}') := C(\vec{o}) C(\vec{o}').
$$

Then clearly, $C \otimes C \neq 0$. We are done if we can show that $(\mathbf{O} \otimes \mathbf{O})(C \otimes C) = 0$ since that establishes that $\mathbf{O} \otimes \mathbf{O}$ is also not injective. For any $\vec{s}, \vec{s}' \in \vec{\mathcal{S}}$, we have

$$
\begin{aligned}
\left[ (\mathbf{O} \otimes \mathbf{O})(C \otimes C) \right](\vec{s}, \vec{s}') &= \mathop{\mathbf{E}}_{\vec{o}, \vec{o}' \sim P_{\vec{O}}(\cdot | \vec{s}, \vec{s}')} \left[ (C \otimes C)(\vec{o}, \vec{o}') \right] \\
&= \mathop{\mathbf{E}}_{\vec{o}, \vec{o}' \sim P_{\vec{O}}(\cdot | \vec{s}, \vec{s}')} \left[ C(\vec{o}) \cdot C(\vec{o}') \right] \\
&= \mathop{\mathbf{E}}_{\vec{o} \sim P_{\vec{O}}(\vec{o} | \vec{s})} \left[ C(\vec{o}) \right] \cdot \mathop{\mathbf{E}}_{\vec{o}' \sim P_{\vec{O}}(\vec{o}' | \vec{s}')} \left[ C(\vec{o}') \right] \\
&= \left[ \mathbf{O}(C) \right](\vec{s}) \cdot \left[ \mathbf{O}(C) \right](\vec{s}') \\
&= 0 \cdot 0 \\
&= 0.
\end{aligned}
$$

This finishes the proof. $\qquad \square$

We now state and prove the following extension of Corollary D.4:

**Theorem D.9.** *Consider the following statements (which can each be true or false):*

1. *$\mathbf{O} : \mathbb{R}^{\vec{\Omega}} \to \mathbb{R}^{\vec{\mathcal{S}}}$ is an injective linear operator:* $\ker \mathbf{O} = \{0\}$.

2. *$\mathbf{O} \otimes \mathbf{O} : \mathbb{R}^{\vec{\Omega} \times \vec{\Omega}} \to \mathbb{R}^{\vec{\mathcal{S}} \times \vec{\mathcal{S}}}$ is an injective linear operator:* $\ker \mathbf{O} \otimes \mathbf{O} = \{0\}$.

3. *$\mathbf{O} \otimes \mathbf{O}$ is injective on vectors of observation-based choice probabilities* $\left( P^R \left( \vec{o} \succ \vec{o}' \right) \right)_{\vec{o}, \vec{o}'}$ *over the set of return functions* $G \in \mathbb{R}^{\vec{\mathcal{S}}}$.

4. *The data of state-based choice probabilities* $\left( P^R \left( \vec{s} \succ \vec{s}' \right) \right)_{\vec{s}, \vec{s}' \in \vec{\mathcal{S}}}$ *from Equation* (8) *determine the data of observation-based choice probabilities* $\left( P^R \left( \vec{o} \succ \vec{o}' \right) \right)_{\vec{o}, \vec{o}' \in \vec{\Omega}}$ *from Equation* (2).

*Then the following implications hold and 3 does not imply 2:*

$$1 \rightleftharpoons 2 \implies 3 \implies 4.$$

*Consequently, if any of the conditions 1, 2, or 3 hold, and additionally any of the conditions (i), (ii) or (iii) from Corollary D.4, then the data $\left( P^R(\vec{s} \succ \vec{s}') \right)_{\vec{s},\vec{s}' \in \vec{\Omega}}$ determine the return function $G$ on sequences $\vec{s} \in \vec{\mathcal{S}}$ up to a constant independent of $\vec{s}$.*

*Proof.* That 1 and 2 are equivalent was shown in Lemma D.8. That 2 implies 3 is clear. To prove that 3 implies 4, simply put both sets of choice probabilities into a vector. Then Equation (8) and Definition D.7 show the following equality of vectors in $\mathbb{R}^{\vec{\mathcal{S}} \times \vec{\mathcal{S}}}$:

$$\left( P^R(\vec{s} \succ \vec{s}') \right)_{\vec{s},\vec{s}'} = (\mathbf{O} \otimes \mathbf{O}) \left( \left( P^R(\vec{o} \succ \vec{o}') \right)_{\vec{o},\vec{o}'} \right).$$

The injectivity of $\mathbf{O} \otimes \mathbf{O}$ on such inputs ensures that the observation-based choice probabilities can be recovered using this equation.

We now show that (3) does not imply (2). Again, we use the simple MDP from Equation (7), but this time with a different observation kernel. Namely, we choose

$$P_O(o^{(a)} \mid a) = P_O(o^{(a)'} \mid a) = \frac{1}{2}, \quad P_O(o^{(b)} \mid b) = 1,$$

where $o^{(a)'} \neq o^{(a)}$ and $o^{(a)} \neq o^{(b)} \neq o^{(a)'}$. This results in two possible observation sequences: $o^{(a)} o^{(b)}$ and $o^{(a)'} o^{(b)}$. Thus, $\mathbb{R}^{\vec{\Omega}}$ is two-dimensional, whereas $\mathbb{R}^{\vec{\mathcal{S}}}$ is only one-dimensional. Consequently, $\mathbf{O} : \mathbb{R}^{\vec{\Omega}} \to \mathbb{R}^{\vec{\mathcal{S}}}$ cannot be injective, so $\ker \mathbf{O} \neq \{0\}$, so (2) does not hold since (1) and (2) are equivalent. However, (3) still holds: Since there is only one state sequence, Equation (2) shows that the only vector of choice probabilities has $1/2$ in all its entries, irrespective of the return function $G$. Thus, $\mathbf{O} \otimes \mathbf{O}$ has only one input of observation-based choice probabilities, and is thus automatically injective on its inputs.

The final result of identifiability of the return function $G$ follows using Corollary D.4. $\qquad\square$

## D.5   Simple Special Cases: Full Observability, Deterministic $P_{\vec{O}}$, and Noisy $P_{\vec{O}}$

In this section, we analyze three simple special cases of the general theory.

Theorem 3.9 (together with Lemma B.3) from Skalse et al. [2023], reproduced as a corollary below, is a special case of our theorem:

**Corollary D.10** (Skalse et al. [2023]). *Assume the human directly observes the true sequences, and the choice probabilities are given by*

$$P^R(\vec{s} \succ \vec{s}') = \sigma\Big( \beta\big(G(\vec{s}) - G(\vec{s}')\big)\Big).$$

*This data determines the return function $G = \mathbf{\Gamma}(R)$ on state sequences $\vec{s} \in \vec{\mathcal{S}}$ up to a constant independent on $\vec{s}$.*

*Proof.* We can embed this case into the one of Theorem D.9 by defining the observation kernel as $P_{\vec{O}}(\vec{s}' \mid \vec{s}) = \delta_{\vec{s}}(\vec{s}')$ (i.e., the correct sequence is deterministically observed) and defining the human's belief as $B(\vec{s}' \mid \vec{s}) = \delta_{\vec{s}}(\vec{s}')$ (i.e., the human knows that the observation reflects the true sequence). This shows that $P(\vec{s} \succ \vec{s}')$ is of the form of Equation (8). The result follows from Theorem D.9: the operators $\mathbf{O}$ and $\mathbf{B}$ are the identity in this case, due to the defining property of the Kronecker delta, and so they are injective. $\qquad\square$

The following proposition shows that Corollary D.10 is essentially the *only* example of deterministic observation kernel $P_{\vec{O}}$ for which $\mathbf{B}$ is injective. Note, however, that in some situations, we can have $\operatorname{im} \mathbf{\Gamma} \cap \ker \mathbf{B} = \{0\}$ even if $\mathbf{B}$ is not injective, see Example D.32.

**Proposition D.11.** *Assume $P_{\vec{O}}$, the observation kernel on the level of sequences, is deterministic and not injective. Then $\mathbf{O}$ is automatically injective. However, $\mathbf{B}$ is not injective.*

*Proof.* To show that $\mathbf{O}$ is injective, assume $v \in \mathbb{R}^{\vec{\Omega}}$ is such that $\mathbf{O}(v) = 0$. Then for all $\vec{s} \in \vec{\mathcal{S}}$, we get

$$0 = \big[\mathbf{O}(v)\big](\vec{s}) = \mathop{\mathbb{E}}_{\vec{o} \sim P_{\vec{O}}(\vec{o}|\vec{s})} \big[v(\vec{o})\big] = v\big(\vec{O}(\vec{s})\big).$$

Since $\vec{O} : \vec{\mathcal{S}} \to \vec{\Omega}$ is by definition surjective, we obtain $v = 0$.

$\vec{O} : \vec{\mathcal{S}} \to \vec{\Omega}$ is by definition surjective, and here assumed to be non-injective, which implies that $\vec{\mathcal{S}}$ has a higher cardinality than $\vec{\Omega}$. Thus, $\mathbf{B} : \mathbb{R}^{\vec{\mathcal{S}}} \to \mathbb{R}^{\vec{\Omega}}$ cannot be injective. $\square$

In the following, we analyze a simple case that guarantees identifiability. It requires that the observation kernel is "well-behaved" of a form where the observations are simply "noisy states", and that the human is a Bayesian reasoner with any prior $B(\vec{s})$ that supports every state sequence $\vec{s} \in \vec{\mathcal{S}}$.

**Definition D.12** (Noise in the Observation Kernel). *Then we say that there is* noise in the observation kernel $P_O : \vec{\mathcal{S}} \to \Delta(\vec{\Omega})$ *if* $\vec{\mathcal{S}} = \vec{\Omega}$ *and if* $\mathbf{O}$ *is an injective linear operator.*

**Proposition D.13.** *Assume that $\vec{\mathcal{S}} = \vec{\Omega}$. Furthermore, assume that $B(\vec{s} \mid \vec{o})$ is given by the posterior with likelihood $P_{\vec{O}}(\vec{o} \mid \vec{s})$ and any prior $B(\vec{s})$ with $B(\vec{s}) > 0$ for all $\vec{s} \in \vec{\mathcal{S}}$. Then there is noise in the observation kernel if and only if $\mathbf{B}$ is injective.*

*Proof.* Assume $\mathbf{O}$ is injective. To show that $\mathbf{B}$ is injective, assume there is $G' \in \mathbb{R}^{\vec{\mathcal{S}}}$ with $\mathbf{B}(G') = 0$. Then for all $\vec{o} \in \vec{\Omega}$, we have

$$0 = \big[\mathbf{B}(G')\big](\vec{o}) = \mathop{\mathbb{E}}_{\vec{s} \sim B(\vec{s}|\vec{o})} \big[G'(\vec{s})\big] = \sum_{\vec{s}} B(\vec{s} \mid \vec{o}) G'(\vec{s}) \propto \sum_{\vec{s}} P_{\vec{O}}(\vec{o} \mid \vec{s}) \cdot \big(B(\vec{s}) \cdot G'(\vec{s})\big)$$

$$= \big[\mathbf{O}^T(B \odot G')\big](\vec{o}).$$

Here, $\mathbf{O}^T$ is the transpose of $\mathbf{O}$ and $B \odot G'$ is the componentwise product of the prior $B$ with the return function $G'$. Since $\mathbf{O}$ is injective and thus invertible, $\mathbf{O}^T$ is as well. Thus, $B \odot G' = 0$, which implies $G' = 0$ since the prior gives positive probability to all state sequences. Thus, $\mathbf{B}$ is injective.

For the other direction, assume $\mathbf{B}$ is injective. To show that $\mathbf{O}$ is injective, let $v \in \mathbb{R}^{\vec{\Omega}}$ be any vector with $\mathbf{O}(v) = 0$. We do a similar computation as above: for all $\vec{s} \in \mathbb{R}^{\vec{\mathcal{S}}}$, we have

$$0 = \big[\mathbf{O}(v)\big](\vec{s}) = \mathop{\mathbb{E}}_{\vec{o} \sim P_{\vec{O}}(\vec{o}|\vec{s})} \big[v(\vec{o})\big] = \sum_{\vec{o}} P_{\vec{O}}(\vec{o} \mid \vec{s}) v(\vec{o}) \propto \sum_{\vec{o}} B(\vec{s} \mid \vec{o}) \cdot \big(P_{\vec{O}}(\vec{o}) \cdot v(\vec{o})\big)$$

$$= \Big[\mathbf{B}^T \big(P_{\vec{O}} \odot v\big)\Big](\vec{s}).$$

Here, $\mathbf{B}^T$ is the transpose of $\mathbf{B}$, $P_{\vec{O}}(\vec{o})$ is the denominator in Bayes rule, and $P_{\vec{O}} \odot v$ is the vector with components $P_{\vec{O}}(\vec{o}) \cdot v(\vec{o})$. From the injectivity and thus invertibility of $\mathbf{B}$, it follows that $\mathbf{B}^T$ is invertible as well, and so $P_{\vec{O}} \odot v = 0$, which implies $v = 0$. Thus, $\mathbf{O}$ is injective. $\square$

**Corollary D.14.** *When there is noise in the observation kernel and the human is a Bayesian reasoner with some prior $B$ such that $B(\vec{s}) > 0$ for all $\vec{s} \in \vec{\mathcal{S}}$, then the return function is identifiable from choice probabilities of state sequences even if the learning system does not know the human's observations.*

*Proof.* This follows from the injectivity of $\mathbf{O}$, the injectivity of $\mathbf{B}$ that we proved in Proposition D.13, and Theorem D.9. $\square$

**Remark D.15.** *We mention the following caveat: intuitively, one could think that $\mathbf{O}$ (and thus $\mathbf{B}$, by Proposition D.13) will be injective if every $\vec{s}$ is identifiable from infinitely many i.i.d. samples from $P_{\vec{O}}(\vec{o} \mid \vec{s})$. A counterexample is the following:*

$$\mathbf{O} = \begin{pmatrix} 1/2 & 1/4 & 1/4 \\ 1/4 & 1/2 & 1/4 \\ 3/8 & 3/8 & 1/4 \end{pmatrix}.$$

*In this case, the rows are linearly dependent with coefficients $1/2, 1/2$ and $-1$. Consequently, $\mathbf{O}$ and $\mathbf{B}$ are not injective, and so if this observation kernel comes from a multi-armed bandit with three states, then Corollary D.4 shows that the return function is not identifiable.*

*Nevertheless, the distributions $P_{\vec{O}}(\cdot \mid \vec{s})$ (given by the rows) all differ from each other, and so infinitely many i.i.d. samples identify the state sequence $\vec{s}$.*

## D.6 Robustness of Return Function Identifiability under Belief Misspecification

We now again look at the case where the observations that the human observes are known to the reward learning system, as in Section D.2. Furthermore, we assume that $\mathbf{B} : \mathbb{R}^{\vec{S}} \to \mathbb{R}^{\vec{\Omega}}$ is such that $\ker \mathbf{B} \cap \operatorname{im} \mathbf{\Gamma} = \{0\}$. In this case, we can apply Corollary D.4 and identify the true return function $G$ from $\mathbf{B}(G)$, which, in turn, can be identified up to an additive constant from the observation-based choice probabilities with the argument as for Proposition 3.1.

In this section, we investigate what happens when the human belief model is slightly misspecified. In other words: the learning system uses a perturbed matrix $\mathbf{B_\Delta} := \mathbf{B} + \mathbf{\Delta}$ with some small perturbation $\mathbf{\Delta}$. How much will the inferred return function deviate from the truth? To answer this, we first need to outline some norm theory of linear operators.

### D.6.1 Some Norm Theory for Linear Operators

In this section, let $V, W$ be two finite-dimensional inner product-spaces. In other words, $V$ and $W$ each have inner products $\langle \cdot, \cdot \rangle$ and there are linear isomorphisms $V \cong \mathbb{R}^k$, $W \cong \mathbb{R}^m$ such that the inner products in $V$ and $W$ correspond to the standard scalar products in $\mathbb{R}^k$ and $\mathbb{R}^m$. The reason that we don't directly work with $\mathbb{R}^k$ and $\mathbb{R}^m$ itself is that we will later apply the analysis to the case that $V = \operatorname{im} \mathbf{\Gamma} \subseteq \mathbb{R}^{\vec{S}}$. Let in this whole section $\mathbf{A} : V \to W$ be a linear operator and $\mathbf{\Delta} : V \to W$ be a perturbance, so that $\mathbf{A_\Delta} := \mathbf{A} + \mathbf{\Delta}$ is a perturbed version of $\mathbf{A}$.

The inner products give rise to a norm on $V$ and $W$ defined by

$$\|v\| = \sqrt{\langle v, v \rangle}, \quad \|w\| = \sqrt{\langle w, w \rangle}.$$

As is well known, for each linear operator $\mathbf{A} : V \to W$ there exists a unique, basis-independent *adjoint* (generalizing the notion of a transpose) $\mathbf{A}^T : W \to V$ such that for all $v \in V$ and $w \in W$, we have

$$\langle \mathbf{A}\, v, w \rangle = \left\langle v, \mathbf{A}^T\, w \right\rangle.$$

Let us recall the following fact that is often used in linear regression:

**Lemma D.16.** *Assume $\mathbf{A} : V \to W$ is injective. Then $\mathbf{A}^T \mathbf{A} : V \to V$ is invertible and $(\mathbf{A}^T \mathbf{A})^{-1} \mathbf{A}^T$ is a left inverse of $\mathbf{A}$.*

*Proof.* To show that $\mathbf{A}^T \mathbf{A}$ is invertible, we only need to show that it is injective. Thus, let $0 \neq x \in V$. Then

$$\left\langle x, \mathbf{A}^T \mathbf{A}\, x \right\rangle = \langle \mathbf{A}\, x, \mathbf{A}\, x \rangle = \| \mathbf{A}\, x \|^2 > 0,$$

where the last step followed from the injectivity of $\mathbf{A}$. Thus, $\mathbf{A}^T \mathbf{A}\, x \neq 0$, and so $\mathbf{A}^T \mathbf{A}$ is injective, and thus invertible. Consequently, $(\mathbf{A}^T \mathbf{A})^{-1} \mathbf{A}^T$ is a well-defined operator. That it is the left inverse of $\mathbf{A}$ is clear. $\square$

**Definition D.17** (Operator Norm)**.** *The norm of an operator $\mathbf{A} : V \to W$ is given by*

$$\| \mathbf{A} \| := \max_{x,\, \|x\|=1} \| \mathbf{A}\, x \|.$$

*It has the following well-known properties, where $\mathbf{A}, \mathbf{B}$ and $\mathbf{C}$ are matrices of compatible sizes:*

$$\| \mathbf{A} + \mathbf{B} \| \leq \| \mathbf{A} \| + \| \mathbf{B} \|, \quad \| \mathbf{C}\, \mathbf{A} \| \leq \| \mathbf{C} \| \cdot \| \mathbf{A} \|, \quad \| \mathbf{A}^T \| = \| \mathbf{A} \|.$$

To study how a perturbance in $\mathbf{A}$ (and thus $\mathbf{A}^T \mathbf{A}$) transfers into a perturbance of $\left( \mathbf{A}^T \mathbf{A} \right)^{-1}$, we will use the following theorem:

**Theorem D.18** (El Ghaoui [2002]). *Let* $\mathbf{B} : V \to V$ *be an invertible operator. Let* $\rho < \| \mathbf{B}^{-1} \|^{-1}$. *Let* $\mathbf{\Delta} : V \to V$ *be any operator with* $\| \mathbf{\Delta} \| \leq \rho$. *Then* $\mathbf{B} + \mathbf{\Delta}$ *is invertible and we have*

$$\big\| (\mathbf{B} + \mathbf{\Delta})^{-1} - \mathbf{B}^{-1} \big\| \leq \frac{\rho \cdot \| \mathbf{B}^{-1} \|}{\| \mathbf{B}^{-1} \|^{-1} - \rho}.$$

*Proof.* See El Ghaoui [2002], Section 7 and in particular Equation 7.2. Note that the reference defines $\| \mathbf{A} \|$ to be the largest singular value of $\mathbf{A}$; by the well-known min-max theorem, this is equivalent to Definition D.17. $\qquad \square$

We will apply this theorem to $\mathbf{A}^T \mathbf{A}$, which raises the question about the size of the perturbance in $\mathbf{A}^T \mathbf{A}$ for a given perturbance in $\mathbf{A}$. This is clarified in the following lemma. Before stating it, for a given perturbance $\rho$, define

$$\widetilde{\rho}(\mathbf{A}) := \rho \cdot \big( 2 \cdot \| \mathbf{A} \| + \rho \big),$$

which depends on $\mathbf{A}$ and $\rho$. Also, recall that for a given perturbance $\mathbf{\Delta}$, we define $\mathbf{A_\Delta} := \mathbf{A} + \mathbf{\Delta}$. We obtain:

**Lemma D.19.** *Assume that* $\| \mathbf{\Delta} \| \leq \rho$. *Then*

$$\| \mathbf{A_\Delta}^T \mathbf{A_\Delta} - \mathbf{A}^T \mathbf{A} \| \leq \widetilde{\rho}(\mathbf{A}).$$

*Proof.* We have

$$\begin{aligned}
\big\| \mathbf{A_\Delta}^T \mathbf{A_\Delta} - \mathbf{A}^T \mathbf{A} \big\| &= \big\| (\mathbf{A} + \mathbf{\Delta})^T (\mathbf{A} + \mathbf{\Delta}) - \mathbf{A}^T \mathbf{A} \big\| \\
&= \big\| \mathbf{A}^T \mathbf{\Delta} + \mathbf{\Delta}^T \mathbf{A} + \mathbf{\Delta}^T \mathbf{\Delta} \big\| \\
&\leq \| \mathbf{A} \| \cdot \| \mathbf{\Delta} \| + \| \mathbf{\Delta} \| \cdot \| \mathbf{A} \| + \| \mathbf{\Delta} \|^2 \\
&\leq \rho \cdot \big( 2 \cdot \| \mathbf{A} \| + \rho \big) \\
&= \widetilde{\rho}(\mathbf{A}).
\end{aligned}$$

$\qquad \square$

To be able to apply Theorem D.18 to $\mathbf{A}^T \mathbf{A}$, we need to make sure that $\widetilde{\rho}(\mathbf{A})$ is bounded above by $\big\| (\mathbf{A}^T \mathbf{A})^{-1} \big\|^{-1}$. The next lemma clarifies what condition $\rho$ needs to satisfy for $\widetilde{\rho}(\mathbf{A})$ to obey that bound. For this, define

$$\tau(\mathbf{A}) := -\| \mathbf{A} \| + \sqrt{ \| \mathbf{A} \|^2 + \big\| (\mathbf{A}^T \mathbf{A})^{-1} \big\|^{-1} }, \tag{9}$$

which only depends on $\mathbf{A}$.

**Lemma D.20.** *Assume* $\rho < \tau(\mathbf{A})$. *Then*

$$\widetilde{\rho}(\mathbf{A}) < \big\| (\mathbf{A}^T \mathbf{A})^{-1} \big\|^{-1}.$$

*Proof.* Note that $\rho = \tau(\mathbf{A})$ is the positive solution to the following quadratic equation in the indeterminate $\rho$:

$$\rho^2 + 2 \cdot \| \mathbf{A} \| \cdot \rho - \big\| (\mathbf{A}^T \mathbf{A})^{-1} \big\|^{-1} = \widetilde{\rho}(\mathbf{A}) - \big\| (\mathbf{A}^T \mathbf{A})^{-1} \big\|^{-1} = 0.$$

Since this is a convex parabola, we get the inequality $\widetilde{\rho}(\mathbf{A}) - \big\| (\mathbf{A}^T \mathbf{A})^{-1} \big\|^{-1} < 0$ whenever we have $0 \leq \rho < \tau(\mathbf{A})$, which shows the result. $\qquad \square$

Finally, we put it all together to obtain a bound on the perturbance of $\big( \mathbf{A}^T \mathbf{A} \big)^{-1} \mathbf{A}^T$. For this, set

$$C(\mathbf{A}, \rho) := \frac{\widetilde{\rho}(\mathbf{A}) \cdot \big\| \big( \mathbf{A}^T \mathbf{A} \big)^{-1} \big\|}{\big\| \big( \mathbf{A}^T \mathbf{A} \big)^{-1} \big\|^{-1} - \widetilde{\rho}(\mathbf{A})} \cdot \Big( \| \mathbf{A} \| + \rho \Big) + \big\| \big( \mathbf{A}^T \mathbf{A} \big)^{-1} \big\| \cdot \rho. \tag{10}$$

We obtain:

**Proposition D.21.** *Assume* $\| \mathbf{\Delta} \| \leq \rho < \tau(\mathbf{A})$. *Then* $\mathbf{A}_{\mathbf{\Delta}}^T \mathbf{A}_{\mathbf{\Delta}}$ *is invertible, and we have*

$$\left\| \left( \mathbf{A}_{\mathbf{\Delta}}^T \mathbf{A}_{\mathbf{\Delta}} \right)^{-1} \mathbf{A}_{\mathbf{\Delta}}^T - \left( \mathbf{A}^T \mathbf{A} \right)^{-1} \mathbf{A}^T \right\| \leq C(\mathbf{A}, \rho).$$

*Proof.* The invertibility of $\mathbf{A}_{\mathbf{\Delta}}^T \mathbf{A}_{\mathbf{\Delta}}$ follows from Theorem D.18, Lemma D.19 and Lemma D.20. We get

$$
\begin{aligned}
&\left\| \left( \mathbf{A}_{\mathbf{\Delta}}^T \mathbf{A}_{\mathbf{\Delta}} \right)^{-1} \mathbf{A}_{\mathbf{\Delta}}^T - \left( \mathbf{A}^T \mathbf{A} \right)^{-1} \mathbf{A}^T \right\| \\
&= \left\| \left[ \left( \mathbf{A}_{\mathbf{\Delta}}^T \mathbf{A}_{\mathbf{\Delta}} \right)^{-1} - \left( \mathbf{A}^T \mathbf{A} \right)^{-1} \right] \cdot \mathbf{A}_{\mathbf{\Delta}}^T + \left( \mathbf{A}^T \mathbf{A} \right)^{-1} \cdot \left( \mathbf{A}_{\mathbf{\Delta}}^T - \mathbf{A}^T \right) \right\| \\
&\leq \left\| \left( \mathbf{A}_{\mathbf{\Delta}}^T \mathbf{A}_{\mathbf{\Delta}} \right)^{-1} - \left( \mathbf{A}^T \mathbf{A} \right)^{-1} \right\| \cdot \| \mathbf{A}_{\mathbf{\Delta}} \| + \left\| \left( \mathbf{A}^T \mathbf{A} \right)^{-1} \right\| \cdot \| \mathbf{\Delta} \| \\
&\leq \frac{\widetilde{\rho}(\mathbf{A}) \cdot \left\| \left( \mathbf{A}^T \mathbf{A} \right)^{-1} \right\|}{\left\| \left( \mathbf{A}^T \mathbf{A} \right)^{-1} \right\|^{-1} - \widetilde{\rho}(\mathbf{A})} \cdot \left( \| \mathbf{A} \| + \rho \right) + \left\| \left( \mathbf{A}^T \mathbf{A} \right)^{-1} \right\| \cdot \rho \\
&= C(\mathbf{A}, \rho).
\end{aligned}
$$

In the second-to-last step, we used Theorem D.18. $\qquad \square$

The constant $C(\mathbf{A}, \rho)$, defined in Equation (10), has a fairly complicated form. In the following proposition, we find an easier-to-study upper bound in a special case:

**Proposition D.22.** *Assume that* $\rho \leq \| \mathbf{A} \|$ *and* $\rho \leq -\| \mathbf{A} \| + \sqrt{\| \mathbf{A} \|^2 + 1/2 \cdot \| (\mathbf{A}^T \mathbf{A})^{-1} \|^{-1}}$.[3] *Then we have*

$$C(\mathbf{A}, \rho) \leq \rho \cdot \left\| (\mathbf{A}^T \mathbf{A})^{-1} \right\| \cdot \left[ 12 \cdot \| \mathbf{A} \|^2 \cdot \left\| (\mathbf{A}^T \mathbf{A})^{-1} \right\| + 1 \right].$$

*Proof.* The second assumption gives, as in the proof of Lemma D.20, that $\widetilde{\rho}(\mathbf{A}) \leq 1/2 \cdot \| (\mathbf{A}^T \mathbf{A})^{-1} \|^{-1}$. Together with $\rho \leq \| \mathbf{A} \|$, the result follows. $\qquad \square$

### D.6.2 Application to Bounds in the Error of the Return Function

We now apply the results from the preceding section to our case. Define $\mathrm{r}(\mathbf{B}) : \operatorname{im} \mathbf{\Gamma} \to \mathbb{R}^{\vec{\Omega}}$ as the restriction of the belief operator $\mathbf{B}$ to $\operatorname{im} \mathbf{\Gamma}$. Assume that $\ker \mathbf{B} \cap \operatorname{im} \mathbf{\Gamma} = \{0\}$, which is, according to Corollary D.4, a sufficient condition for identifiability. Note that this condition means that $\mathrm{r}(\mathbf{B})$ is injective. Thus, Lemma D.16 ensures that $\mathrm{r}(\mathbf{B})^T \mathrm{r}(\mathbf{B})$ is invertible and that $\left( \mathrm{r}(\mathbf{B})^T \mathrm{r}(\mathbf{B}) \right)^{-1} \mathrm{r}(\mathbf{B})^T$ is a left inverse of $\mathrm{r}(\mathbf{B})$.

Consequently, from the equation

$$\mathrm{r}(\mathbf{B})(G) = \mathbf{B}(G)$$

we obtain

$$G = \left( \mathrm{r}(\mathbf{B})^T \mathrm{r}(\mathbf{B}) \right)^{-1} \mathrm{r}(\mathbf{B})^T (\mathbf{B}(G)).$$

This is the concrete formula with which $G$ can be identified from $\mathbf{B}(G)$. When perturbing $\mathbf{B}$, this leads to a corresponding perturbance in $\left( \mathrm{r}(\mathbf{B})^T \mathrm{r}(\mathbf{B}) \right)^{-1} \mathrm{r}(\mathbf{B})^T$ whose size influences the maximal error in the inference of $G$. This, in turn, influences the size of the error in $J_G$, the policy evaluation function, where

$$J_G(\pi) \coloneqq \mathop{\mathbf{E}}_{\vec{s} \sim P^\pi(\vec{s})} \left[ G(\vec{s}) \right].$$

We obtain:

---

[3]Note the factor $1/2$ compared to the definition of $\tau(\mathbf{A})$ in Equation (9).

**Theorem D.23.** *Let $G$ be the true reward function, $\mathbf{B}$ the belief operator corresponding to the human's true belief model $B(\vec{s} \mid \vec{o})$, and $\mathbf{B}(G)$ be the resulting observation-based return function. Assume that $\ker \mathbf{B} \cap \operatorname{im} \mathbf{\Gamma} = \{0\}$, so that $\mathrm{r}(\mathbf{B})^T \mathrm{r}(\mathbf{B})$ is invertible. Let $\mathbf{\Delta} : \mathbb{R}^{\vec{\mathcal{S}}} \to \mathbb{R}^{\vec{\Omega}}$ be a perturbation satisfying $\|\mathbf{\Delta}\| \leq \rho$, where $\rho$ satisfies the following two properties:*

$$\rho \leq \|\mathrm{r}(\mathbf{B})\|, \quad \rho \leq -\|\mathrm{r}(\mathbf{B})\| + \sqrt{\|\mathrm{r}(\mathbf{B})\|^2 + 1/2 \cdot \|(\mathrm{r}(\mathbf{B})^T \mathrm{r}(\mathbf{B}))^{-1}\|^{-1}}.$$

*Let $\mathbf{B}_{\mathbf{\Delta}} := \mathbf{B} + \mathbf{\Delta}$ be the misspecified belief operator. The first claim is that $\mathrm{r}(\mathbf{B}_{\mathbf{\Delta}})^T \mathrm{r}(\mathbf{B}_{\mathbf{\Delta}})$ is invertible under these conditions.*

*Now, assume that the learning system infers the return function $\tilde{G} := \left(\mathrm{r}(\mathbf{B}_{\mathbf{\Delta}})^T \mathrm{r}(\mathbf{B}_{\mathbf{\Delta}})\right)^{-1} \mathrm{r}(\mathbf{B}_{\mathbf{\Delta}})^T (\mathbf{B}(G))$.[4] Then there is a polynomial $Q(X, Y)$ of degree five such that*

$$\|\tilde{G} - G\| \leq \|G\| \cdot Q\left(\|(\mathrm{r}(\mathbf{B})^T \mathrm{r}(\mathbf{B}))^{-1}\|, \|\mathrm{r}(\mathbf{B})\|\right) \cdot \rho.$$

*Thus, for all policies $\pi$, we obtain*

$$\left| J_{\tilde{G}}(\pi) - J_G(\pi) \right| \leq \|G\| \cdot Q\left(\|(\mathrm{r}(\mathbf{B})^T \mathrm{r}(\mathbf{B}))^{-1}\|, \|\mathrm{r}(\mathbf{B})\|\right) \cdot \rho.$$

*In particular, for sufficiently small perturbances $\rho$, the error in the inferred policy evaluation function $J_{\tilde{G}}$ becomes arbitrarily small.*

*Proof.* That $\mathrm{r}(\mathbf{B}_{\mathbf{\Delta}})^T \mathrm{r}(\mathbf{B}_{\mathbf{\Delta}})$ is invertible follows immediately from Proposition D.21 by using that $\|\mathrm{r}(\mathbf{\Delta})\| \leq \|\mathbf{\Delta}\|$ and that $\mathrm{r}(\mathbf{B}_{\mathbf{\Delta}}) = \mathrm{r}(\mathbf{B})_{\mathrm{r}(\mathbf{\Delta})}$, together with the second bound on $\rho$ (which implies the assumed bound in Proposition D.21).

We have

$$
\begin{aligned}
\left| J_{\tilde{G}}(\pi) - J_G(\pi) \right| &= \left| \mathop{\mathbf{E}}_{\vec{s} \sim P^\pi(\vec{s})} \left[ (\tilde{G} - G)(\vec{s}) \right] \right| \\
&\leq \mathop{\mathbf{E}}_{\vec{s} \sim P^\pi(\vec{s})} \left[ |(\tilde{G} - G)(\vec{s})| \right] \\
&\leq \max_{\vec{s} \in \vec{\mathcal{S}}} \left| (\tilde{G} - G)(\vec{s}) \right| \\
&\leq \|\tilde{G} - G\| \\
&= \left\| \left[ \left(\mathrm{r}(\mathbf{B}_{\mathbf{\Delta}})^T \mathrm{r}(\mathbf{B}_{\mathbf{\Delta}})\right)^{-1} \mathrm{r}(\mathbf{B}_{\mathbf{\Delta}})^T - \left(\mathrm{r}(\mathbf{B})^T \mathrm{r}(\mathbf{B})\right)^{-1} \mathrm{r}(\mathbf{B})^T \right] \cdot \mathbf{B}(G) \right\| \\
&\leq \left\| \left(\mathrm{r}(\mathbf{B}_{\mathbf{\Delta}})^T \mathrm{r}(\mathbf{B}_{\mathbf{\Delta}})\right)^{-1} \mathrm{r}(\mathbf{B}_{\mathbf{\Delta}})^T - \left(\mathrm{r}(\mathbf{B})^T \mathrm{r}(\mathbf{B})\right)^{-1} \mathrm{r}(\mathbf{B})^T \right\| \cdot \|\mathbf{B}(G)\| \\
&\leq C(\mathrm{r}(\mathbf{B}), \rho) \cdot \|\mathrm{r}(\mathbf{B})(G)\| \\
&\leq C(\mathrm{r}(\mathbf{B}), \rho) \cdot \|\mathrm{r}(\mathbf{B})\| \cdot \|G\|.
\end{aligned}
$$

In the second to last step, we used Proposition D.21. By Proposition D.22, we can define the polynomial $Q(X, Y)$ by

$$Q(X, Y) = XY \cdot \left[ 12XY^2 + 1 \right],$$

which is of degree five.

The last claim follows from $\lim_{\rho \to 0} \rho = 0$. $\qquad \square$

**Remark D.24.** *In the case of a square matrix $\mathbf{B}$ that is injective, we can apply Theorem D.18 directly to $\mathbf{B}^{-1}$ (which is now invertible) and obtain the following simplification of Theorem D.23 for the case that $\|\mathbf{\Delta}\| \leq \rho \leq \frac{1}{2} \cdot \|\mathbf{B}^{-1}\|^{-1}$:*

$$\left| J_{\tilde{G}}(\pi) - J_G(\pi) \right| \leq \rho \cdot 2 \cdot \|\mathbf{B}\| \cdot \|G\| \cdot \|\mathbf{B}^{-1}\|^2.$$

*The polynomial is then only of degree 3.*

---

[4]Note that there is not necessarily a $\tilde{G}$ with $\mathrm{r}(\mathbf{B}_{\mathbf{\Delta}})(\tilde{G}) = \mathbf{B}(G)$ since $\mathrm{r}(\mathbf{B}_{\mathbf{\Delta}})$ is not always surjective. Nevertheless, $\tilde{G} := \left(\mathrm{r}(\mathbf{B}_{\mathbf{\Delta}})^T \mathrm{r}(\mathbf{B}_{\mathbf{\Delta}})\right)^{-1} \mathrm{r}(\mathbf{B}_{\mathbf{\Delta}})^T (\mathbf{B}(G))$ is the best attempt at a solution in the sense that $\mathrm{r}(\mathbf{B}_{\mathbf{\Delta}})(\tilde{G})$ then minimizes the Euclidean distance to $\mathbf{B}(G)$.

## D.7 Preliminary Characterizations of the Ambiguity

Recall the sequence of functions

$$\mathbb{R}^{\mathcal{S}} \xrightarrow{\ \mathbf{\Gamma}\ } \mathbb{R}^{\vec{\mathcal{S}}} \xrightarrow{\ \mathbf{B}\ } \mathbb{R}^{\vec{\Omega}}.$$

In this section, we clarify $\operatorname{im} \mathbf{\Gamma}$ and $\ker \mathbf{B}$ in special cases, as their intersection is the crucial ambiguity in Theorem D.2.

The following proposition shows that for deterministic $P_{\vec{O}}$ and a rational human, $\ker \mathbf{B}$ decomposes into hyperplanes defined by normal vectors of probabilities of sequences mapping to the same observation sequence:

**Proposition D.25.** *Assume the human reasons as in Section D.1. Assume $P_{\vec{O}}$ is deterministic. Let $B(\vec{s})$ be the distribution of sequences under the human's belief over the policy, given by $B(\vec{s}) = \int_{\pi'} B(\pi') P^{\pi'}(\vec{s})$ for some policy prior $B(\pi')$. For each $\vec{o}$, let $B_{\vec{o}} := [B(\vec{s})]_{\vec{s}:\ \vec{O}(\vec{s})=\vec{o}} \in \mathbb{R}^{\{\vec{s} \in \vec{\mathcal{S}} \mid \vec{O}(\vec{s})=\vec{o}\}}$ be the vector of probabilities of sequences that are observed as $\vec{o}$.*

*Let $G'$ be a return function. For each $\vec{o} \in \vec{\Omega}$, define the restriction $G'_{\vec{o}} \in \mathbb{R}^{\{\vec{s} \in \vec{\mathcal{S}} \mid \vec{O}(\vec{s})=\vec{o}\}}$ by $G'_{\vec{o}}(\vec{s}) := G'(\vec{s})$ for all $\vec{s} \in \{\vec{s} \in \vec{\mathcal{S}} \mid \vec{O}(\vec{s}) = \vec{o}\}$. Assume that $B(\vec{s} \mid \vec{o})$ is the Bayesian posterior. Then $G' \in \ker \mathbf{B}$ if and only if the property*

$$B_{\vec{o}} \cdot G'_{\vec{o}} = 0$$

*holds for all $\vec{o} \in \vec{\Omega}$.*

*Proof.* For a deterministic observation kernel $P_{\vec{O}}$, by Bayes rule we have

$$
\begin{aligned}
B(\vec{s} \mid \vec{o}) &= \frac{P_{\vec{O}}(\vec{o} \mid \vec{s}) \cdot B(\vec{s})}{\sum_{\vec{s}'} P_{\vec{O}}(\vec{o} \mid \vec{s}') \cdot B(\vec{s}')} \\
&= \frac{\delta_{\vec{o}}\big(\vec{O}(\vec{s})\big) \cdot B(\vec{s})}{\sum_{\vec{s}'} \delta_{\vec{o}}\big(\vec{O}(\vec{s}')\big) \cdot B(\vec{s}')} \\
&= \begin{cases} 0, & \vec{O}(\vec{s}) \neq \vec{o} \\ \frac{B(\vec{s})}{\sum_{\vec{s}':\ \vec{O}(\vec{s}')=\vec{o}} B(\vec{s}')}, & \vec{O}(\vec{s}) = \vec{o}. \end{cases}
\end{aligned}
$$

Thus, for any return function $G'$ and any observation sequence $\vec{o}$, we have

$$
\begin{aligned}
\big[\mathbf{B}(G')\big](\vec{o}) &= \mathop{\mathbb{E}}_{\vec{s} \sim B(\vec{s}|\vec{o})} \big[G'(\vec{s})\big] \\
&= \sum_{\vec{s}} B(\vec{s} \mid \vec{o}) G'(\vec{s}) \\
&= \sum_{\vec{s}:\ \vec{O}(\vec{s})=\vec{o}} \frac{B(\vec{s})}{\sum_{\vec{s}':\ \vec{O}(\vec{s}')=\vec{o}} B(\vec{s}')} G'(\vec{s}) \\
&= \left( \sum_{\vec{s}':\ \vec{O}(\vec{s}')=\vec{o}} B(\vec{s}') \right)^{-1} \cdot \sum_{\vec{s}:\ \vec{O}(\vec{s})=\vec{o}} B(\vec{s}) G'(\vec{s}).
\end{aligned}
$$

Thus, we have $G' \in \ker \mathbf{B}$ if and only if

$$B_{\vec{o}} \cdot G'_{\vec{o}} = \sum_{\vec{s}:\ \vec{O}(\vec{s})=\vec{o}} B(\vec{s}) G'(\vec{s}) = 0$$

for all $\vec{o}$. That was to show. $\qquad\square$

**Remark D.26.** *One can interpret the previous proposition as follows:*

*As long as $\vec{O}$ is injective, we have $\big|\{\vec{s} \in \vec{\mathcal{S}} \mid \vec{O}(\vec{s}) = o\}\big| = 1$ for all $\vec{o}$, meaning that $B_{\vec{o}}$ and $G'_{\vec{o}}$ have only one entry. Thus, $B_{\vec{o}} \cdot G'_{\vec{o}} = 0$ implies $G'_{\vec{o}} = 0$. If that holds for all $\vec{o}$, then $G' \in \ker \mathbf{B}$ implies $G' = 0$, meaning $\mathbf{B}$ is injective.*

*However, as soon as there is an $\vec{o}$ with $k_{\vec{o}} := \left|\{\vec{s} \in \vec{\mathcal{S}} \mid \vec{O}(\vec{s}) = o\}\right| > 1$, the equation $B_{\vec{o}} \cdot G'_{\vec{o}} = 0$ leads to $k_{\vec{o}} - 1$ free parameters in $G'_{\vec{o}}$. $G'_{\vec{o}}$ can then be chosen freely in the hyperplane of vectors orthogonal to $B_{\vec{o}}$ without moving out of the kernel of $\mathbf{B}$.*

*Another way of writing Proposition D.25 is to write $\ker \mathbf{B}$ as a direct sum of these hyperplanes perpendicular to $B_{\vec{o}}$:*

$$\ker \mathbf{B} = \bigoplus_{\vec{o}:\ |\vec{O}^{-1}(\vec{o})| \geq 2} B_{\vec{o}}^{\perp}.$$

Recall that a return function $G$ is called *time-separable* if there exists a reward function $R$ such that $\mathbf{\Gamma}(R) = G$.

Before we discuss time-separability in more interesting examples, we want to talk about one simple case where all return functions are time-separable. We leave a general characterization of $\operatorname{im} \mathbf{\Gamma}$ to future work.

**Proposition D.27.** *Let there be an ordering $\vec{s}^{(1)}, \vec{s}^{(2)}, \dots$ of all sequences in $\vec{\mathcal{S}}$, and a function $\phi : \vec{\mathcal{S}} \to \mathcal{S}$ from sequences to states such that $\phi(\vec{s}) \in \vec{s}$ and $\phi(\vec{s}^{(k)}) \notin \vec{s}^{(i)}$ for all $i < k$. Then every return function is time-separable.*

*Proof.* Let $G$ be a return function. Initialize $R(s) = 0$ for all $s$ and inductively update it for all $i = 1, 2, \dots$:

$$R\big(\phi(\vec{s}^{(i)})\big) := \left( \sum_{t:\ s_t^{(i)} = \phi(\vec{s}^{(i)})} \gamma^t \right)^{-1} \cdot \left( G(\vec{s}^{(i)}) - \sum_{t:\ s_t^{(i)} \neq \phi(\vec{s}^{(i)})} \gamma^t \cdot R\big(s_t^{(i)}\big) \right),$$

where the inductive definition always uses $R$ as it is defined by that point in time. Once $R\big(\phi(\vec{s}^{(i)})\big)$ is defined, but not yet any future values $R\big(\phi(\vec{s}^{(k)})\big)$, $k > i$, we have

$$\big[\mathbf{\Gamma}(R)\big](\vec{s}^{(i)}) = \sum_{t=0}^{T} \gamma^t \cdot R\big(s_t^{(i)}\big)$$

$$= \left( \sum_{t:\ s_t^{(i)} = \phi(\vec{s}^{(i)})} \gamma^t \right) \cdot R\big(\phi(\vec{s}^{(i)})\big) + \sum_{t:\ s_t^{(i)} \neq \phi(\vec{s}^{(i)})} \gamma^t \cdot R\big(s_t^{(i)}\big)$$

$$= G(\vec{s}^{(i)}).$$

Furthermore, the property $\phi(\vec{s}^{(k)}) \notin \vec{s}^{(i)}$ for all $i < k$ ensures that changes to the reward function for $k > i$ do not affect the value of $\big[\mathbf{\Gamma}(R)\big](\vec{s}^{(i)})$. This shows $\mathbf{\Gamma}(R) = G$, and thus $G$ is time-separable. $\qquad\square$

**Corollary D.28.** *In a multi-armed bandit, every return function is time-separable.*

*Proof.* In a multi-armed bandit, states and sequences are equivalent, and so we can choose $\phi(s) = s$ for every state/sequence $s$. The result follows from Proposition D.27.

Alternatively, simply directly notice that in a multi-armed bandit, $\mathbf{\Gamma}$ is the identity mapping, and so for every return/reward function $R$, we have $\mathbf{\Gamma}(R) = R$. $\qquad\square$

### D.8  Examples Supplementing Section 5

In this whole section, the inverse temperature parameter in the human choice probabilities is given by $\beta = 1$. We now consider four more mathematical examples of Corollary D.4 and Theorem D.9. In the first example, the ambiguity is so bad that the reward inference can become worse than simply maximizing $J_{\mathrm{obs}}$ as in naive RLHF. In Example D.30, there is simply "noise" in the observations and the human's belief, the matrices $\mathbf{B}$ and $\mathbf{O}$ are injective, and identifiability works, as in Corollary D.14. In the third example, the matrix $\mathbf{B}$ is not injective and identifiability fails, which is a minimal example showing the limits of our main theorems. In the fourth example, the matrix $\mathbf{B}$ is not injective, but $\ker \mathbf{B} \cap \operatorname{im} \mathbf{\Gamma} = \{0\}$, and so identifiability works. This example is interesting in that the identifiability

simply emerges through different distributions of *delay* that are caused by the different unobserved events.

In this section, both the linear operators $\mathbf{B} : \mathbb{R}^{\vec{\mathcal{S}}} \to \mathbb{R}^{\vec{\Omega}}$ and $\mathbf{O} : \mathbb{R}^{\vec{\Omega}} \to \mathbb{R}^{\vec{\mathcal{S}}}$ are considered as matrices

$$\mathbf{O} = \left( P_{\vec{O}}(\vec{o} \mid \vec{s}) \right)_{\vec{s}, \vec{o}} \in \mathbb{R}^{\vec{\mathcal{S}} \times \vec{\Omega}}, \quad \mathbf{B} = \left( B(\vec{s} \mid \vec{o}) \right)_{\vec{o}, \vec{s}} \in \mathbb{R}^{\vec{\Omega} \times \vec{\mathcal{S}}}.$$

Notice that both have a swap in their indices.

**Example D.29.** *Theorem 5.2 shows that the remaining ambiguity from the human's choice probabilities is given by* $\ker \mathbf{B} \cap \operatorname{im} \mathbf{\Gamma}$, *but it doesn't explain how to proceed given this ambiguity. Without further inductive biases, some reward functions within the ambiguity of the true reward function can be even worse than simply maximizing* $J_{\text{obs}}$.

*E.g., consider a multi-armed bandit with three actions* $a, b, c$, *observation-kernel* $o = O(a) = O(b) \neq O(c) = c$ *and reward function* $R(a) = R(b) < R(c)$. *If the human belief is given by* $B(a \mid o) = p = 1 - B(b \mid o)$, *then* $R' = \alpha \cdot (p - 1, p, 0) \in \mathbb{R}^{\{a,b,c\}}$ *is in the ambiguity for all* $\alpha \in \mathbb{R}$, *and so* $\tilde{R} := R + R'$ *is compatible with the choice probabilities. However, for* $\alpha \ll 0$, *we have* $\tilde{R}(a) > \tilde{R}(b)$ *and* $\tilde{R}(a) > \tilde{R}(c)$, *and so optimizing against this reward function leads to a suboptimal policy.*

*In contrast, maximizing* $J_{\text{obs}}$ *leads to the correct policy since* a, b, *and* c *all obtain their ground truth reward in this example. This generally raises the question of how to tie-break reward functions in the ambiguity, or how to act conservatively given the uncertainty, in order to consistently improve upon the setting in Section 4.1.*

**Example D.30.** *This example is a special case of Corollary D.14. Consider a multi-armed bandit with two actions (which are automatically also states and sequences)* a *and* b. *In this case, the reward function and return function is the same.*

*We assume there to be two possible observations* $o^{(a)}, o^{(b)}$ *and the observation kernel to be nondeterministic, with probabilities*

$$P_O(o^{(j)} \mid i) = \begin{cases} 2/3, & \text{if } i = j, \\ 1/3, & \text{else.} \end{cases}$$

*If we assume the human forms Bayesian posterior beliefs as in Section D.1 and to have a policy prior* $B(\pi')$ *such that* $B(a) = \int_{\pi} \pi(a) B(\pi') d\pi = 1/2$ *and* $B(b) = 1/2$, *then it is easy to show that the human's belief is the "reversed" observation kernel:*

$$B(j \mid o^{(i)}) = P_O(o^{(i)} \mid j).$$

*We obtain*

$$\mathbf{O} = \mathbf{B} = \begin{pmatrix} 2/3 & 1/3 \\ 1/3 & 2/3 \end{pmatrix} = \frac{1}{3} \cdot \begin{pmatrix} 2 & 1 \\ 1 & 2 \end{pmatrix}$$

*These matrices are injective since they are invertible:*

$$\mathbf{O}^{-1} = \mathbf{B}^{-1} = \begin{pmatrix} 2 & -1 \\ -1 & 2 \end{pmatrix}.$$

*More generally, even if the human does not form fully rational posterior beliefs, it is easy to imagine that the matrix* $\mathbf{B}$ *can end up being invertible. Thus, Corollary D.4 guarantees that the reward function can be inferred up to an additive constant from the choice probabilities of observations, and Theorem D.9 shows that this even works when the learning system does not know what the human observed.*

*In the rest of this example, we explicitly walk the reader through the process of how the reward function can be inferred, in the general case that the observations are not known. In the process, we essentially recreate the proof of the theorems for this special case. For this aim, we first want to compute the choice probabilities* $P^R(i \succ j)$ *that the learning system has access to in the limit of infinite data. We assume that the reward function is given by* $R(a) = -1$ *and* $R(b) = 2$. *We compute:*

$$\mathbf{B}(R) = \frac{1}{3} \cdot \begin{pmatrix} 2 & 1 \\ 1 & 2 \end{pmatrix} \cdot \begin{pmatrix} -1 \\ 2 \end{pmatrix} = \begin{pmatrix} 0 \\ 1 \end{pmatrix}.$$

In other words, we have $\mathbf{E}_{s \sim B(s|o^{(a)})}[R(s)] = 0$ and $\mathbf{E}_{s \sim B(s|o^{(b)})}[R(s)] = 1$. From this, we can compute the observation-based choice probabilities $\widetilde{P}_{o^{(i)}o^{(j)}} = \sigma\big(\mathbf{B}(R)(o^{(i)}) - \mathbf{B}(R)(o^{(j)})\big)$, see Equation (2), and obtain:

$$\widetilde{P}_{o^{(a)}o^{(a)}} = \widetilde{P}_{o^{(b)}o^{(b)}} = \frac{1}{2}, \quad \widetilde{P}_{o^{(a)}o^{(b)}} = \frac{1}{1+e}, \quad \widetilde{P}_{o^{(b)}o^{(a)}} = \frac{e}{1+e}.$$

We can now determine the final choice probabilities $P_{ij} := P^R(i \succ j)$ again by a matrix-vector product, with the indices ordered lexicographically, see Equation (8). Here, $\mathbf{O} \otimes \mathbf{O}$ is the Kronecker product of the matrix $\mathbf{O}$ with itself:

$$P = (\mathbf{O} \otimes \mathbf{O}) \cdot \widetilde{P} = \frac{1}{9} \cdot \begin{pmatrix} 4 & 2 & 2 & 1 \\ 2 & 4 & 1 & 2 \\ 2 & 1 & 4 & 2 \\ 1 & 2 & 2 & 4 \end{pmatrix} \cdot \begin{pmatrix} 1/2 \\ 1/(1+e) \\ e/(1+e) \\ 1/2 \end{pmatrix} = \begin{pmatrix} 1/2 \\ 1/3 \cdot (2+e)/(1+e) \\ 1/3 \cdot (1+2e)/(1+e) \\ 1/2 \end{pmatrix}.$$

For example, the second entry in $P$ is $P_{ab} = P^R(a \succ b) = \frac{2+e}{3 \cdot (1+e)}$. This is the likelihood that, for ground-truth actions $a, b$, the human will prefer $a$ after only receiving observations $o^{(a)}$ or $o^{(b)}$ according to $\mathbf{O}$ and following a Boltzman-rational policy based on the belief of the real action, see Equation (8).

Over time, the learning system will be able to estimate these probabilities based on repeated human choices, assuming all state-pairs are sampled infinitely often. The question of identifiability is whether the original reward function $R$ can be inferred from that data, given that the learning system knows $\mathbf{O}$ and $\mathbf{B}$. We assume that the learning system doesn't a priori know $R$ or any of the intermediate steps in the computation. First, $\widetilde{P}$ can be inferred by inverting $\mathbf{O} \otimes \mathbf{O}$:

$$\widetilde{P} = (\mathbf{O} \otimes \mathbf{O})^{-1} \cdot P = \begin{pmatrix} 4 & -2 & -2 & 1 \\ -2 & 4 & 1 & -2 \\ -2 & 1 & 4 & -2 \\ 1 & -2 & -2 & 4 \end{pmatrix} \cdot \begin{pmatrix} 1/2 \\ 1/3 \cdot (2+e)/(1+e) \\ 1/3 \cdot (1+2e)/(1+e) \\ 1/2 \end{pmatrix} = \begin{pmatrix} 1/2 \\ 1/(1+e) \\ e/(1+e) \\ 1/2 \end{pmatrix}.$$

The learning system wants to use this to infer $\mathbf{B}(\tilde{R})$ (for the later-to-be inferred reward function $\tilde{R}$ that may differ from the true reward function $R$) and uses the equation

$$\widetilde{P}_{o^{(a)}o^{(b)}} = \frac{\exp\big(\mathbf{B}(\tilde{R})(o^{(a)})\big)}{\exp\big(\mathbf{B}(\tilde{R})(o^{(a)})\big) + \exp\big(\mathbf{B}(\tilde{R})(o^{(b)})\big)},$$

which can be rearranged to

$$\mathbf{B}(\tilde{R})(o^{(a)}) = \log \frac{\widetilde{P}_{o^{(a)}o^{(b)}}}{1 - \widetilde{P}_{o^{(a)}o^{(b)}}} + \mathbf{B}(\tilde{R})(o^{(b)}) = \log \frac{1/(1+e)}{e/(1+e)} + \mathbf{B}(\tilde{R})(o^{(b)}) = \mathbf{B}(\tilde{R})(o^{(b)}) - 1.$$

This relation is all which can be inferred about $\mathbf{B}(\tilde{R})(o^{(a)})$ and $\mathbf{B}(\tilde{R})(o^{(b)})$; the precise value cannot be determined and $\mathbf{B}(\tilde{R})(o^{(b)})$ is a free parameter. One can check that for $\mathbf{B}(\tilde{R})(o^{(b)}) = 1$ this coincides with the true value $\mathbf{B}(R)$. Finally, one can invert $\mathbf{B}$ to infer $\tilde{R}$ from this:

$$\begin{aligned}
\tilde{R} &= \mathbf{B}^{-1} \cdot \mathbf{B}(\tilde{R}) \\
&= \begin{pmatrix} 2 & -1 \\ -1 & 2 \end{pmatrix} \cdot \begin{pmatrix} \mathbf{B}(\tilde{R})(o^{(b)}) - 1 \\ \mathbf{B}(\tilde{R})(o^{(b)}) \end{pmatrix} \\
&= \begin{pmatrix} \mathbf{B}(\tilde{R})(o^{(b)}) - 2 \\ 1 + \mathbf{B}(\tilde{R})(o^{(b)}) \end{pmatrix} \\
&= \begin{pmatrix} -1 \\ 2 \end{pmatrix} + \begin{pmatrix} \mathbf{B}(\tilde{R})(o^{(b)}) - 1 \\ \mathbf{B}(\tilde{R})(o^{(b)}) - 1 \end{pmatrix} \\
&= R + \begin{pmatrix} \mathbf{B}(\tilde{R})(o^{(b)}) - 1 \\ \mathbf{B}(\tilde{R})(o^{(b)}) - 1 \end{pmatrix}.
\end{aligned}$$

Thus, the inferred and true reward functions differ maximally by a constant, as predicted in Theorem D.9.

In the following example, we work out a case where the reward function is so ambiguous that any policy is optimal to some reward function consistent with the human feedback:

**Example D.31.** *Consider a multi-armed bandit with exactly three actions/states $a, b, c$. We assume a deterministic observation kernel with $o := O(a) = O(c) \neq O(b) = b$. Assume the human has some arbitrary beliefs $B(a \mid o), B(c \mid o) = 1 - B(a \mid o)$, and can identify $b$: $B(b \mid b) = 1$. Then if the human makes observation comparisons with a Boltzman-rational policy, as in Theorem D.2, the resulting reward function is so ambiguous that some reward functions consistent with the feedback place the highest value on action $a$, no matter the true reward function $R$. Thus, even if the true reward function $R$ regards $a$ as the worst action, $a$ can result from the reward learning and subsequent policy optimization process.*

*Proof.* The matrix $\mathbf{B} : \mathbb{R}^{\{a,b,c\}} \to \mathbb{R}^{\{o,b\}}$ is given by

$$\mathbf{B} = \begin{pmatrix} B(a \mid o) & 0 & B(c \mid o) \\ 0 & 1 & 0 \end{pmatrix}.$$

Its kernel is given by reward functions $R'$ with $R'(b) = 0$ and $R'(c) = -\frac{B(a|o)}{B(c|o)} R'(a)$, with $R'(a)$ a free parameter. Theorem D.2 shows that, up to an additive constant, the reward functions consistent with the feedback of observation comparisons are given by $\tilde{R} = R + R'$ for any $R' \in \ker \mathbf{B}$. Thus, whenever the free parameter $R'(a)$ satisfies $R'(a) > R(b) - R(a)$ and $R'(a) > B(c \mid o) \cdot \big(R(c) - R(a)\big)$, we obtain $\tilde{R}(a) > \tilde{R}(b)$ and $\tilde{R}(a) > \tilde{R}(c)$, showing the claim. $\qquad\square$

We now investigate another example where $\mathbf{B}$ is not injective, and yet, identifiability works because $\mathbf{B} \circ \mathbf{\Gamma} \neq \{0\}$. We saw such cases already in Example E.6, but include this additional example since it shows a conceptually interesting case: two different states lead to the exact same observations, but can be disambiguated since they lead to different amounts of *delay* until a more informative observation is made again.

**Example D.32.** *In this example, we assume that the human knows the policy $\pi$ that generates the state sequences (corresponding to a policy prior $B(\pi') = \delta_\pi(\pi')$ concentrated on $\pi$), which together with knowledge of the transition dynamics of the environment determines the true state transition probabilities $\mathcal{T}^\pi(s' \mid s) = \sum_{a \in \mathcal{A}} \mathcal{T}(s' \mid s, a) \cdot \pi(a \mid s)$. We consider an environment with three states $s, s', s''$ and the following transition dynamics $\mathcal{T}^\pi$, where $p \neq 1/2$ is a probability:*

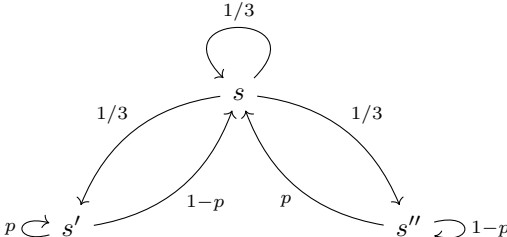

*We assume that $P_0(s) = 1$. Furthermore, we assume deterministic observations and $s = O(s) \neq O(s') = O(s'') =: o$.*

*Assume the time horizon $T$ is 3, i.e., there are timesteps $0, 1, 2, 3$. Assume that the human forms the belief over the true state sequence by Bayesian posterior updates as in Section D.1. In this case, $\ker \mathbf{B} \neq \{0\}$ by Proposition D.11. However, we will now show that $\ker(\mathbf{B} \circ \mathbf{\Gamma}) = \{0\}$. If the human makes Boltzmann-rational comparisons of observation sequences, then this implies the identifiability of the return function up to an additive constant by Corollary D.4.[5]*

*Thus, let $R' \in \ker(\mathbf{B} \circ \mathbf{\Gamma})$, i.e., $\Big[ \mathbf{B}\big(\mathbf{\Gamma}(R')\big) \Big](\vec{o}) = 0$ for every observation sequence $\vec{o}$. For $\vec{o} = ssss$ being the observation sequence that only consists of state $s$, this implies $R'(s) = 0$. Consequently, for general observation sequences $\vec{o}$, we have:*

$$0 = \Big[ \mathbf{B}\big(\mathbf{\Gamma}(R')\big) \Big](\vec{o}) = \mathop{\mathbb{E}}_{\vec{s} \sim B(\vec{s}|\vec{o})} \left[ \sum_{t=0}^{3} \delta_{s'}(s_t) \cdot \gamma^t \right] \cdot R'(s') + \mathop{\mathbb{E}}_{\vec{s} \sim B(\vec{s}|\vec{o})} \left[ \sum_{t=0}^{3} \delta_{s''}(s_t) \cdot \gamma^t \right] \cdot R'(s'').$$

---

[5]We assume that the learning system knows what the human observes, which is valid since $P_O$ is deterministic. Alternatively, one can argue with Proposition D.11 that $\mathbf{O}$ is automatically injective, meaning one can apply Theorem D.9.

*Now we specialize this equation to the two observation sequences $\vec{o}^{(1)} = soss$ and $\vec{o}^{(2)} = soos$. We start by considering $\vec{o}^{(1)}$. This is consistent with the two state sequences $\vec{s}^{(1),(s')} = ss'ss$ and $\vec{s}^{(1),(s'')} = ss''ss$. We have posterior probabilities*

$$B\big(\vec{s}^{(1),(s')} \mid \vec{o}^{(1)}\big) = 1 - p, \quad B\big(\vec{s}^{(1),(s'')} \mid \vec{o}^{(1)}\big) = p,$$

*and therefore*

$$0 = \Big[ \mathbf{B}\left( \mathbf{\Gamma}(R') \right) \Big](\vec{o}^{(1)}) = (1 - p) \cdot \gamma \cdot R'(s') + p \cdot \gamma \cdot R'(s''),$$

*and so*

$$R'(s') = \frac{p}{p - 1} \cdot R'(s''). \tag{11}$$

*Similarly, $\vec{o}^{(2)}$ is consistent with the sequences $\vec{s}^{(2),(s')} = ss's's$ and $\vec{s}^{(2),(s'')} = ss''s''s$. They have posterior probabilities*

$$B\big(\vec{s}^{(2),(s')} \mid \vec{o}^{(2)}\big) = \frac{1}{2}, \quad B\big(\vec{s}^{(2),(s'')} \mid \vec{o}^{(2)}\big) = \frac{1}{2},$$

*leading to*

$$0 = \frac{1}{2} \cdot (\gamma + \gamma^2) \cdot R'(s') + \frac{1}{2} \cdot (\gamma + \gamma^2) \cdot R'(s'').$$

*Together with Equation* (11)*, we obtain*

$$R'(s'') = -R'(s') = \frac{p}{1 - p} \cdot R'(s''),$$

*which implies $R'(s'') = 0$ because $p \neq \frac{1}{2}$, and thus also $R'(s') = 0$. Overall, we have showed $R' = 0$, and so $\mathbf{B} \circ \mathbf{\Gamma}$ is injective. This means that reward functions are identifiable in this example up to an additive constant, see Corollary D.4.*

## E   Issues of Naively Applying RLHF under Partial Observability

In this section, we study the naive application of RLHF under partial observability. Thus, most of it takes a step back from the general theory of *appropriately modeled* partial observability in RLHF. Later, we will analyze examples where we also apply the general theory, which is why this appendix section comes second.

In Section E.1, we first briefly explain what happens when the learning system incorrectly assumes that the human observes the full environment state. We show that as a consequence, the system is incentivized to infer what we call the *observation return function* $G_{\text{obs}}$, which evaluates a state sequence based on the human's belief of the state sequence given the human's observations. In the policy optimization process, the policy is then selected to maximize $J_{\text{obs}}$, an expectation over $G_{\text{obs}}$. In the interlude in Section E.2, we then briefly analyze the unrealistic case that the human, when evaluating a policy $\pi$, fully knows the complete specification of that policy and all of the environment and engages in rational Bayesian reasoning; in this case, $J_{\text{obs}} = J$ is the true policy evaluation function.

Realistically, however, maximizing $J_{\text{obs}}$ can lead to failure modes. In Appendix E.3 we prove that a suboptimal policy that is optimal according to $J_{\text{obs}}$ causes deceptive inflation, overjustification, or both. In Appendix C.3, we expand on the analysis of the main examples in the main paper. Finally, in Section E.4, we study further concrete examples where maximizing $J_{\text{obs}}$ reveals deceptive and overjustifying behavior by the resulting policy.

### E.1   Optimal Policies under RLHF with Deterministic Partial Observations Maximize $J_{\text{obs}}$

Assume that $P_{\vec{O}}$ is deterministic and that the human makes Boltzmann-rational sequence comparisons between observation sequences. The true choice probabilities are then given by (See Equations (2) and (8)):

$$P^R\big(\vec{s} \succ \vec{s}'\big) = \sigma\bigg( \beta \cdot \Big( \big( \mathbf{B} \cdot G \big)\big(\vec{O}(\vec{s})\big) - \big( \mathbf{B} \cdot G \big)\big(\vec{O}(\vec{s}')\big) \Big) \bigg) \tag{12}$$

Now, assume that the learning system does *not model the situation correctly*. In particular, we assume:

- The system is not aware that the human only observes observation sequences $\vec{O}(\vec{s})$ instead of the full state sequences.

- The system does not model that the human's return function is *time-separable*, i.e., comes from a reward function $R$ over environment states.

The learning system then thinks that there is a return function $\tilde{G} \in \mathbb{R}^{\vec{S}}$ such that the choice probabilities are given by the following faulty formula:

$$P^R(\vec{s} \succ \vec{s}') := \sigma\Big(\beta\big(G(\vec{s}) - G(\vec{s}')\big)\Big)$$

Now, assume that the learning system has access to the choice probabilities and wants to infer $G$. Inverting the sigmoid function and then plugging in the true choice probabilities from Equation (12), we obtain:

$$
\begin{aligned}
\tilde{G}(\vec{s}) &= \frac{1}{\beta} \log \frac{P^R(\vec{s} \succ \vec{s}')}{P^R(\vec{s}' \succ \vec{s})} + \tilde{G}(\vec{s}') \\
&= \frac{1}{\beta}\Big[\beta \cdot \Big(\big(\mathbf{B} \cdot G\big)\big(\vec{O}(\vec{s})\big) - \big(\mathbf{B} \cdot G\big)\big(\vec{O}(\vec{s}')\big)\Big)\Big] + \tilde{G}(\vec{s}') \\
&= \big(\mathbf{B} \cdot G\big)\big(\vec{O}(\vec{s})\big) + C(\vec{s}').^{6}
\end{aligned}
$$

Here, $C(\vec{s}')$ is some quantity that does not depend on $\vec{s}$. Now, fix $\vec{s}'$ as a reference sequence. Then for varying $\vec{s}$, $C(\vec{s}')$ is simply an additive constant. Consequently, up to an additive constant, this determines the return function that the learning system is incentivized to infer. We call it the *observation return function* since it is the return function based on the human's observations:

$$G_{\mathrm{obs}}(\vec{s}) := \big(\mathbf{B} \cdot G\big)\big(\vec{O}(\vec{s})\big).$$

This return function is not necessarily time-separable, but we assume that time-separability is not modeled correctly by the learning system. Now, define the resulting policy evaluation function $J_{\mathrm{obs}}$ by

$$J_{\mathrm{obs}}(\pi) := \mathop{\mathbf{E}}_{\vec{s} \sim P^\pi(\vec{s})} \big[G_{\mathrm{obs}}(\vec{s})\big].$$

This is the policy evaluation function that would be optimized if the learning system erroneously inferred the return function $G_{\mathrm{obs}}$.

## E.2 Interlude: When the Human Knows the Policy and is a Bayesian Reasoner, then $J_{\mathrm{obs}} = J$

In this section, we briefly consider what would happen if in $J_{\mathrm{obs}}$, the human's belief $B$ would make use of the true policy and be a rational Bayesian posterior as in Section D.1. We will show that under these conditions, we have $J_{\mathrm{obs}} = J$. Since these are unrealistic assumptions, no other section depends on this result.

For the analysis, we drop the assumption that the observation sequence kernel $P_{\vec{O}}$ is deterministic, and assume that $J_{\mathrm{obs}}$ is given as follows:

$$J_{\mathrm{obs}}(\pi) := \mathop{\mathbf{E}}_{\vec{s} \sim P^\pi(\vec{s})} \left[ \mathop{\mathbf{E}}_{\vec{o} \sim P_{\vec{O}}(\vec{o}|\vec{s})} \left[ \mathop{\mathbf{E}}_{\vec{s}' \sim B^\pi(\vec{s}'|\vec{o})} \big[G(\vec{s}')\big] \right] \right]. \tag{13}$$

In this formula, $B^\pi(\vec{s} \mid \vec{o}) := B(\vec{s} \mid \vec{o}, \pi)$ with $B$ being the joint distribution from Section D.1. Formally, this is the posterior of the joint distribution $B(\vec{s}, \vec{o} \mid \pi)$ that is given by the following hidden Markov model:

$$\tag{14}$$

In the case of non-deterministic observation kernels and choice probabilities given as in Equation (8), this argument does not work since the logarithm cannot be swapped with the outer expectation of the choice probabilities.

---

[6]Note that in the case of non-deterministic observation kernels and choice probabilities given as in Equation (8), this argument does not work since the logarithm cannot be swapped with the outer expectation of the choice probabilities.

Here, $\mathcal{T}^\pi(s' \mid s) := \sum_{a \in \mathcal{A}} \mathcal{T}(s' \mid s, a) \cdot \pi(a \mid s)$. $s_0$ is sampled according to the known initial distribution $P_0(s_0)$. The human's posterior $B^\pi(\vec{s}' \mid \vec{o})$ is then the true posterior in this HMM. We obtain:

**Proposition E.1.** *Let $\pi$ be a policy that is known to the human. Then $J_{\mathrm{obs}}(\pi) = J(\pi)$.*

*Proof.* By Equation (13), we have

$$
J_{\mathrm{obs}}(\pi) = \mathop{\mathbf{E}}_{\vec{s} \sim P^\pi(\vec{s})} \left[ \mathop{\mathbf{E}}_{\vec{o} \sim P_{\vec{O}}(\vec{o} \mid \vec{s})} \left[ \mathop{\mathbf{E}}_{\vec{s}' \sim B^\pi(\vec{s}' \mid \vec{o})} \left[ G(\vec{s}') \right] \right] \right]
$$

$$
\stackrel{(1)}{=} \sum_{\vec{s}} P^\pi(\vec{s}) \sum_{\vec{o}} P_{\vec{O}}(\vec{o} \mid \vec{s}) \sum_{\vec{s}'} B^\pi(\vec{s}' \mid \vec{o}) G(\vec{s}')
$$

$$
\stackrel{(2)}{=} \sum_{\vec{s}'} \left[ \sum_{\vec{o}} B^\pi(\vec{s}' \mid \vec{o}) \left[ \sum_{\vec{s}} P_{\vec{O}}(\vec{o} \mid \vec{s}) P^\pi(\vec{s}) \right] \right] G(\vec{s}')
$$

$$
\stackrel{(3)}{=} \sum_{\vec{s}'} \left[ \sum_{\vec{o}} B^\pi(\vec{s}' \mid \vec{o}) B^\pi(\vec{o}) \right] G(\vec{s}')
$$

$$
\stackrel{(4)}{=} \sum_{\vec{s}'} \left[ \sum_{\vec{o}} P^\pi(\vec{s}') P_{\vec{O}}(\vec{o} \mid \vec{s}') \right] G(\vec{s}')
$$

$$
\stackrel{(5)}{=} \sum_{\vec{s}'} P^\pi(\vec{s}') G(\vec{s}')
$$

$$
\stackrel{(6)}{=} \sum_{\vec{s}} P^\pi(\vec{s}) G(\vec{s})
$$

$$
\stackrel{(7)}{=} J(\pi).
$$

In step (1), we wrote the expectations out in terms of sums. In step (2), we reordered them. In step (3), we observed that the inner sum over $\vec{s}$ evaluates to the marginal distribution $B^\pi(\vec{o})$ of the observation sequence $\vec{o}$ in the HMM in Equation (13). In step (4), we used Bayes rule in the inner sum. This is possible since $B^\pi(\vec{s}' \mid \vec{o})$ is the true posterior when $\pi$ is known. In step (5), we pull $P^\pi(\vec{s}')$ out and notice that the remaining inner sum evaluates to 1. Step (6) is a relabeling and step (7) the definition of the true policy evaluation function $J$. $\square$

### E.3 Proof of Theorem 4.5

We first prove the following lemma.

**Lemma E.2.** *Let $\pi$ and $\pi_{ref}$ be two policies. If $J(\pi) < J(\pi_{ref})$ and $J_{\mathrm{obs}}(\pi) > J_{\mathrm{obs}}(\pi_{ref})$, then relative to $\pi_{ref}$, $\pi$ must exhibit deceptive inflation, overjustification, or both.*

*Proof.* We start by establishing a quantitative relationship between the average overestimation and underestimation errors $\overline{E}^+$ and $\overline{E}^-$ as defined in Definition 4.2, the true policy evaluation function $J$, and the observation evaluation function $J_{\mathrm{obs}}$ defined in Equation (4). Define $\Delta : \vec{\mathcal{S}} \to \mathbb{R}$ by $\Delta(\vec{s}) = G_{\mathrm{obs}}(\vec{s}) - G(\vec{s})$, where $G_{\mathrm{obs}}$ is as defined in Equation (3). Consider the quantity

$$
E^+(\vec{s}) - E^-(\vec{s}) = \max\left(0, \Delta(\vec{s})\right) - \max\left(0, -\Delta(\vec{s})\right).
$$

If $\Delta(\vec{s}) > 0$, then the first term is $\Delta(\vec{s})$ and the second one is 0. If $\Delta(\vec{s}) < 0$, then the first term is zero and the second one is $\Delta(\vec{s})$. If $\Delta(\vec{s}) = 0$, then both terms are zero. In all cases the right-hand side is equal to $\Delta(\vec{s})$. Unpacking the definition of $\Delta$ again, we have that for all $\vec{s}$,

$$
E^+(\vec{s}) - E^-(\vec{s}) = G_{\mathrm{obs}}(\vec{s}) - G(\vec{s}). \tag{15}
$$

For any policy $\pi$, if we take the expectation of both sides of this equation over the on-policy distribution admitted by $\pi$, $P^\pi$, we get

$$
\overline{E}^+(\pi) - \overline{E}^-(\pi) = J_{\mathrm{obs}}(\pi) - J(\pi). \tag{16}
$$

We now prove the lemma. Let $\pi$ and $\pi_{\text{ref}}$ be two policies, and assume that $J(\pi) < J(\pi_{\text{ref}})$ and $J_{\text{obs}}(\pi) \geq J_{\text{obs}}(\pi_{\text{ref}})$. Equivalently, we have $J_{\text{obs}}(\pi) - J_{\text{obs}}(\pi_{\text{ref}}) \geq 0$ and $J(\pi_{\text{ref}}) - J(\pi) > 0$, which we combine to state

$$\Big( J_{\text{obs}}(\pi) - J_{\text{obs}}(\pi_{\text{ref}}) \Big) + \Big( J(\pi_{\text{ref}}) - J(\pi) \Big) > 0. \tag{17}$$

Rearranging terms yields

$$\Big( J_{\text{obs}}(\pi) - J(\pi) \Big) - \Big( J_{\text{obs}}(\pi_{\text{ref}}) - J(\pi_{\text{ref}}) \Big) > 0.$$

These two differences inside parentheses are equal to the right-hand side of (16) for $\pi$ and $\pi_{\text{ref}}$, respectively. We substitute the left-hand side of (16) twice to obtain

$$\Big( \overline{E}^+(\pi) - \overline{E}^-(\pi) \Big) - \Big( \overline{E}^+(\pi_{\text{ref}}) - \overline{E}^-(\pi_{\text{ref}}) \Big) > 0.$$

Rearranging terms again yields

$$\Big( \overline{E}^+(\pi) - \overline{E}^+(\pi_{\text{ref}}) \Big) + \Big( \overline{E}^-(\pi_{\text{ref}}) - \overline{E}^-(\pi) \Big) > 0. \tag{18}$$

If $\overline{E}^+(\pi) - \overline{E}^+(\pi_{\text{ref}}) > 0$ then we have $\overline{E}^+(\pi) > \overline{E}^+(\pi_{\text{ref}})$ and, by assumption, $J_{\text{obs}}(\pi) > J_{\text{obs}}(\pi_{\text{ref}})$. By Definition 4.3, this means $\pi$ exhibits deceptive inflation relative to $\pi_{\text{ref}}$.

If $\overline{E}^-(\pi_{\text{ref}}) - \overline{E}^-(\pi) > 0$ then we have $\overline{E}^-(\pi) < \overline{E}^-(\pi_{\text{ref}})$ and, by assumption, $J(\pi) < J(\pi_{\text{ref}})$. By Definition 4.4, this means $\pi$ exhibits overjustification relative to $\pi_{\text{ref}}$.

At least one of the two differences in parentheses in (18) must be positive, otherwise their sum would not be positive. Thus $\pi$ must exhibit deceptive inflation relative to $\pi_{\text{ref}}$, overjustification relative to $\pi_{\text{ref}}$, or both. $\square$

We can now combine earlier results to prove Theorem 4.5, repeated here for convenience:

**Theorem E.3.** *Assume that $P_O$ is deterministic. Let $\pi^*_{\text{obs}}$ be an optimal policy according to a naive application of RLHF under partial observability, and let $\pi^*$ be an optimal policy according to the true objective $J$. If $\pi^*_{\text{obs}}$ is not $J$-optimal, then relative to $\pi^*$, $\pi^*_{\text{obs}}$ must exhibit deceptive inflation, overjustification, or both.*

*Proof.* Because $P_O$ is deterministic, $\pi^*_{\text{obs}}$ must be optimal with respect to $J_{\text{obs}}$ by Proposition 4.1 (proved in Appendix E.1). Thus $J_{\text{obs}}(\pi^*_{\text{obs}}) \geq J_{\text{obs}}(\pi^*)$. Since $\pi^*$ is $J$-optimal and $\pi^*_{\text{obs}}$ is not, $J(\pi^*) < J(\pi^*_{\text{obs}})$. By Lemma E.2, relative to $\pi^*$, $\pi^*_{\text{obs}}$ must exhibit deceptive inflation, overjustification, or both. $\square$

### E.4 Further Examples Supplementing Section 4.4

In this section, we present further mathematical examples supplementing those in Section 4.4. We found many of them before finding the examples we discuss in the main paper, and show the same and additional conceptual features with somewhat less polish. We again assume that $P_{\vec{O}}$ is deterministic.

**Example E.4.** *In the main paper, we have assumed a model where the human obeys Eq. (2) and showed that a naive application of RLHF can lead to suboptimal policies, and the specific failure modes of deceptive inflation and overjustification. What if the human makes the choices in a different way? Specifically, assume that all we know is that $P^R(\vec{o} \succ \vec{o}') + P^R(\vec{o}' \succ \vec{o}) = 1$. Can the human generally choose these choice probabilities in such a way that RLHF is incentivized to infer a reward function whose optimal policies are also optimal for $R$? The answer is no.*

*Take the following example:*

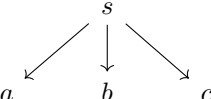

*In this example, there is a fixed start state $s$ and three actions $a, b, c$ that also serve as the final states. The time horizon is $T = 1$, so the only state sequences are $sa, sb, sc$. Assume $\mathcal{T}(a \mid s, a) = 1$,*

$\mathcal{T}(b \mid s, b) = 1, \mathcal{T}(c \mid s, c) = 1 - \epsilon, \mathcal{T}(a \mid s, c) = \epsilon$, *i.e., selecting action $c$ sometimes leads to state $a$. Also, assume $a = O(a) \neq O(b) = O(c) =: o$ and $R(a) = R(b) < R(c)$.*

*Since $b$ and $c$ have the same observation $o$, the human choice probabilities do not make a difference between them, and so RLHF is incentivized to infer a reward function $\tilde{R}$ with $\tilde{R}(b) = \tilde{R}(c) =: \tilde{R}(o)$. If $\tilde{R}(o) > \tilde{R}(a)$, then the policy optimal under $\tilde{R}$ will produce action $b$ since this deterministically leads to observation $o$, whereas $c$ does not. If $\tilde{R}(o) < \tilde{R}(a)$, then the policy optimal under $\tilde{R}$ will produce action $a$. In both cases, the resulting policy is suboptimal compared to $\pi^*$, which deterministically chooses action $c$.*

In the coming examples, it will also be useful to look at the *misleadingness* of state sequences:

**Definition E.5** (Misleadingness). *Let $\vec{s} \in \vec{\mathcal{S}}$ be a state sequence. Then its* misleadingness *is defined by*

$$\mathrm{M}(\vec{s}) := G_{\mathrm{obs}}(\vec{s}) - G(\vec{s}) = \underset{\vec{s}' \sim B(\vec{s}' | \vec{O}(\vec{s}))}{\mathbf{E}} \left[ G(\vec{s}') - G(s) \right].$$

*We call a state sequence* positively misleading *if $M(\vec{s}) > 0$, which means the sequence appears better than it is, and* negatively misleading *if $\mathrm{M}(\vec{s}) < 0$. The* misleadingness vector *is given by $\mathrm{M} \in \mathbb{R}^{\vec{\mathcal{S}}}$.*

Note that the misleadingness is related to $E^+$ and $E^-$, as defined in Definition 4.2: If $M(\vec{s}) > 0$ then $\mathrm{M}(\vec{s}) = E^+(\vec{s})$, and if $\mathrm{M}(\vec{s}) < 0$ then $\mathrm{M}(\vec{s}) = -E^-(\vec{s})$.

**Example E.6.** *In this example, we assume the human is a Bayesian reasoner as in Section D.1. Consider the MDP that is suggestively depicted as follows:*

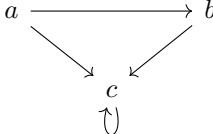

*The MDP has states $\mathcal{S} = \{a, b, c\}$ and actions $\mathcal{A} = \{b, c\}$. The transition kernel is given by $\mathcal{T}(c \mid a, c) = 1$ and $\mathcal{T}(b \mid a, b) = 1$, meaning that the action determines whether to transition from $a$ to $b$ or $c$. All other transitions are deterministic and do not depend on the action, as depicted. We assume an initial state distribution $P_0$ over states with probabilities $p_a = P_0(a), p_b = P_0(b), p_c = P_0(c)$. The true reward function $R \in \mathbb{R}^{\{a,b,c\}}$ and discount factor $\gamma \in [0, 1)$ are, for now, kept arbitrary. The time horizon is $T = 2$, meaning we have four possible state sequences $acc, abc, bcc, ccc$.*

*Furthermore, assume that $o := O(a) = O(b) \neq O(c) = c$, i.e., $c$ is observed and $a$ and $b$ are ambiguous.*

*Finally, assume that the human has a policy prior $B(\lambda)$, where $\lambda = \pi_\lambda(c \mid a)$ is the likelihood that the policy chooses action $c$ when in state $a$, which is a parameter that determines the entire policy.*

*We claim the following:*

1. *If $p_b \neq \gamma \cdot \mathbf{E}_{\lambda \sim B(\lambda)}[\lambda] \cdot p_a$, then $\ker \mathbf{B} \cap \operatorname{im} \mathbf{\Gamma} = \{0\}$, so there is no return function ambiguity under appropriately modeled partially observable RLHF, see Corollary D.4.*

2. *There are true reward functions $R$ for which optimizing $J_{\mathrm{obs}}$ leads to a suboptimal policy according to the true policy evaluation function $J$, a case of misalignment. Thus, a naive application of RLHF under partial observability fails, see Section 4.1.*

3. *The failure modes are related to hiding negative information (deception) and purposefully revealing information while incuring a loss (overjustifying behavior).*

*Proof.* Write $p := B(bcc \mid occ)$, the human's posterior probability of state sequence $bcc$ for observation sequence $occ$. We have $1 - p = B(acc \mid occ)$.

Consider the linear operators $\mathbf{\Gamma} : \mathbb{R}^{\{a,b,c\}} \to \mathbb{R}^{\{abc,bcc,ccc,acc\}}$ and $\mathbf{B} : \mathbb{R}^{\{abc,bcc,ccc,acc\}} \to \mathbb{R}^{\{ooc,occ,ccc\}}$ defined in the main paper. When ordering the states, state sequences, and observation

sequences as we just wrote down, we obtain

$$\mathbf{\Gamma} = \begin{pmatrix} 1 & \gamma & \gamma^2 \\ 0 & 1 & \gamma + \gamma^2 \\ 0 & 0 & 1 + \gamma + \gamma^2 \\ 1 & 0 & \gamma + \gamma^2 \end{pmatrix}, \quad \mathbf{B} = \begin{pmatrix} 1 & 0 & 0 & 0 \\ 0 & p & 0 & 1 - p \\ 0 & 0 & 1 & 0 \end{pmatrix}, \quad \mathbf{B} \circ \mathbf{\Gamma} = \begin{pmatrix} 1 & \gamma & \gamma^2 \\ 1 - p & p & \gamma + \gamma^2 \\ 0 & 0 & 1 + \gamma + \gamma^2 \end{pmatrix}.$$

By Corollary D.4, if $\mathbf{B} \circ \mathbf{\Gamma}$ is injective, then there is no reward function ambiguity. Clearly, this is the case if and only if $p \neq \gamma \cdot (1 - p)$. From Bayes rule, we have

$$p = \frac{B(bcc)}{B(acc) + B(bcc)}, \quad 1 - p = \frac{B(acc)}{B(acc) + B(bcc)}.$$

So the condition for injectivity holds if and only if

$$B(bcc) \neq \gamma \cdot B(acc).$$

Now, notice

$$B(bcc) = \int_\lambda B(\lambda) \cdot B(bcc \mid \lambda) d\lambda = \int_\lambda B(\lambda) \cdot p_b d\lambda = p_b$$

and

$$B(acc) = \int_\lambda B(\lambda) B(acc \mid \lambda) d\lambda = \int_\lambda B(\lambda) \cdot p_a \cdot \lambda d\lambda = p_a \cdot \mathop{\mathbf{E}}_{\lambda \sim B(\lambda)} [\lambda].$$

This shows the first result.

For the second statement, we explicitly compute $J_{\text{obs}}$ up to an affine transformation, which does not change the policy ordering. Let $R$ be the true reward function, $G = \mathbf{\Gamma}(R)$ the corresponding return function, and $\mathbf{B}(G)$ the resulting return function at the level of observations. For simplicity, assume $R(c) = 0$, which can always be achieved by adding a constant. We have:

$$
\begin{aligned}
J_{\text{obs}}(\lambda) &= \mathop{\mathbf{E}}_{\vec{s} \sim P^\lambda(\vec{s})} \Big[ \mathbf{B}(G)\big(\vec{O}(\vec{s})\big) \Big] \\
&= P^\lambda(abc) \cdot \mathbf{B}(G)(ooc) + P^\lambda(bcc) \cdot \mathbf{B}(G)(occ) + P^\lambda(ccc) \cdot \mathbf{B}(G)(ccc) + P^\lambda(acc) \cdot \mathbf{B}(G)(occ) \\
&= p_a \cdot (1 - \lambda) \cdot G(abc) + p_b \cdot \mathbf{B}(G)(occ) + p_c \cdot G(ccc) + p_a \cdot \lambda \cdot \mathbf{B}(G)(occ) \\
&\propto \lambda \cdot \Big[ \mathbf{B}(G)(occ) - G(abc) \Big].
\end{aligned}
$$

We have

$$G(abc) = R(a) + \gamma R(b), \quad \mathbf{B}(G)(occ) = (1 - p) \cdot G(acc) + p \cdot G(bcc) = (1 - p) \cdot R(a) + p \cdot R(b).$$

Thus, the condition $\mathbf{B}(G)(occ) > G(abc)$ is equivalent to

$$R(a) < \frac{p - \gamma}{p} \cdot R(b).$$

Thus, we have

$$\mathop{\arg\max}_{\lambda \in [0,1]} J_{\text{obs}}(\lambda) = \begin{cases} 1, & \text{if } R(a) < \frac{p - \gamma}{p} \cdot R(b), \\ 0, & \text{else.} \end{cases}$$

Now consider the case $R(b) > 0$. In this case, $\lambda = 0$ gives rise to the optimal policy according to $G$ since going to $b$ gives extra reward that one misses when going to $c$ directly. However, when $R(a) \ll 0$, then $J_{\text{obs}}$ selects for $\lambda = 1$. Intuitively, the policy tries to "hide that the episode started in $a$" by going directly to $c$, which leads to ambiguity between $acc$ and $bcc$. This is a case of deceptive inflation as in Theorem 4.5.

Now, consider the case $R(b) < 0$. In this case, $\lambda = 1$ gives rise to the optimal policy according to $G$. However, when $R(a) \gg 0$, then $J_{\text{obs}}$ selects for $\lambda = 0$. Intuitively, the policy tries to "reveal that the episode started with $a$" by going to $b$, which is positive information to the human, but negative from the perspective of optimizing $G$. As in Theorem 4.5, we see that this is a case of overjustification. $\quad \square$

**Example E.7.** *In this example, we consider an MDP that's similar to a multi-armed bandit with four states/actions $a, b, c, d$ and observation kernel $O(a) = O(b) \neq O(c) = O(d)$. Formally, we can imagine that it is given by the MDP*

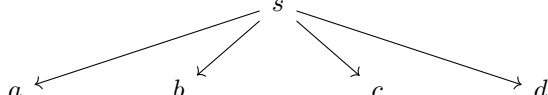

*with $R(s) = 0$ and a time-horizon of $T = 1$. In this example, we reveal that misleadingness and non-optimality (according to the true reward $R$, or $J$) are in principle orthogonal concepts. We consider the following four example cases. In each one, we vary some environment parameters and then determine $a^*_{\text{obs}}$, the action that results from optimizing $J_{\text{obs}}$ (corresponding to a naive application of RLHF under partial observability, see Section 4.1), its misleadingness $\mathrm{M}(a^*_{\text{obs}})$ (see Definition E.5), and the action $a^*$ that would result from optimizing $J$. If $a^*_{\text{obs}} = a^*$, then $J_{\text{obs}}$ selects for the optimal action. For simplicity, we can imagine that the human has a uniform prior over what action results eventually (out of the action taken and potentially a deviation defined by $\epsilon$, see below) is taken before making an observation, i.e. $B(a) = B(b) = B(c) = B(d) = \frac{1}{4}$.*

(a) *Assume $R(a) > R(c) > R(d) \gg R(b)$. Also assume that action $d$ leads with probability $\epsilon > 0$ to state $b$, whereas all other actions lead deterministically to the specified state. Then $a^*_{\text{obs}} = c$, $\mathrm{M}(c) < 0$ and $a^* = a$.*

(b) *Assume $R(d) > R(a) > R(c) \gg R(b)$. Again, assume there is a small probability $\epsilon > 0$ that action $d$ leads to state $b$. Then $a^*_{\text{obs}} = c$, $\mathrm{M}(c) > 0$, and $a^* = d$ or $a^* = a$, depending on the size of $\epsilon$.*

(c) *Assume $R(a) > R(b) > R(c) > R(d)$. Additionally, assume that there is a* large *probability $\epsilon > 0$ that action $a$ leads to state $d$, whereas all other actions lead to what's specified. If $\epsilon$ is large enough, then $a^* = b$. Additionally, we have $a^*_{\text{obs}} = b$ and $\mathrm{M}(b) > 0$.*

(d) *Assume $R(a) > R(b) > R(c) > R(d)$. Also, assume some probability $\epsilon > 0$ that action $b$ leads to state $d$, whereas all other actions lead deterministically to what's specified. Then $a^*_{\text{obs}} = a$, $\mathrm{M}(a) < 0$, and $a^* = a$.*

*Overall, we notice:*

- *Example (a) shows a high regret and negative misleadingness of $a^*_{\text{obs}} = c$. The action is better then it seems, but action $a$ would be better still but cannot be selected because it can be confused with the very bad action $b$.*

- *Example (b) shows a high regret and high misleadingness of $a^*_{\text{obs}} = c$. The action is worse than it seems and also not optimal.*

- *Example (c) shows zero regret and high misleadingness of $a^*_{\text{obs}} = b$. The action is worse than it seems because it can be confused with $a$, but it is still the optimal action because $a$ can turn into $d$.*

- *Example (d) shows zero regret negative misleadingness of $a^*_{\text{obs}} = a$. The action is chosen even though it seems worse than it is, and is also optimal.*

*Thus, we showed all combinations of regret and misleadingness of the action optimized for under $J_{\text{obs}}$.*

*We can also notice the following: Examples (a) and (b) only differ in the placement of $R(d)$. In particular, the* reason *that $a^*_{\text{obs}} = c$ is structurally the same in both, but the misleadingness changes. This indicates that misleadingness is not* on its own *contributing to what $J_{\text{obs}}$ optimizes for.*

The following is the smallest example we found with the following properties:

- There is a unique start state and terminal state.
- A naive application of RLHF fails in a way that shows deception and overjustification.
- Modeling partial observability resolves the problems.

**Example E.8.** *Consider the following graph:*

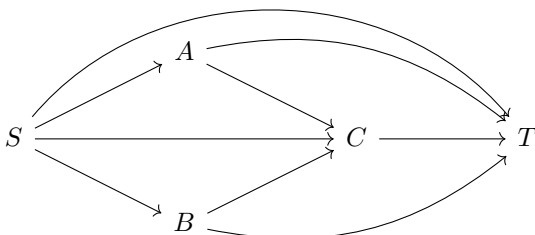

*This depicts an MDP with start state $S$, terminal state $T$ and possible state sequences $STTT, SATT, SACT, SCTT, SBCT, SBTT$ and no discount, i.e. $\gamma = 1$. Assume that $S, B, C$ are observed, i.e. $O(S) = S, O(B) = B, O(C) = C$, and that $A$ and $T$ are ambiguous: $O(A) = O(T) = X$. Then there are five observation sequences $SXXX, SXCX, SCXX, SBCX, SBXX$. Assume that the human can identify all observation sequences except $SXXX$, with belief $b = B(STTT \mid SXXX)$ and $1 - b = B(SATT \mid SXXX)$.*

*Then the return function is identifiable under these conditions when the human's belief is correctly modeled. However, for some choices of the true reward function $R$ and transition dynamics of this MDP, we can obtain deceptive or overjustified behavior for a naive application of RLHF.*

*Proof.* We apply Corollary D.4. We order states, state sequences, and observation sequences as follows:

$$\mathcal{S} = S, A, B, C, T,$$
$$\vec{\mathcal{S}} = STTT, SATT, SACT, SCTT, SBCT, SBTT,$$
$$\vec{\Omega} = SXXX, SXCX, SCXX, SBCX, SBXX.$$

As can easily be verified, with this ordering the matrices $\mathbf{B} \in \mathbb{R}^{\vec{\Omega} \times \vec{\mathcal{S}}}$ and $\mathbf{\Gamma} \in \mathbb{R}^{\vec{\mathcal{S}} \times \mathcal{S}}$ are given by:

$$\mathbf{B} = \begin{pmatrix} b & 1-b & 0 & 0 & 0 & 0 \\ 0 & 0 & 1 & 0 & 0 & 0 \\ 0 & 0 & 0 & 1 & 0 & 0 \\ 0 & 0 & 0 & 0 & 1 & 0 \\ 0 & 0 & 0 & 0 & 0 & 1 \end{pmatrix}, \quad \mathbf{\Gamma} = \begin{pmatrix} 1 & 0 & 0 & 0 & 3 \\ 1 & 1 & 0 & 0 & 2 \\ 1 & 1 & 0 & 1 & 1 \\ 1 & 0 & 0 & 1 & 2 \\ 1 & 0 & 1 & 1 & 1 \\ 1 & 0 & 1 & 0 & 2 \end{pmatrix}.$$

To show identifiability, we need to show that $\ker \mathbf{B} \cap \operatorname{im} \mathbf{\Gamma} = \{0\}$. Clearly, the kernel of $\mathbf{B}$ is given by all return functions in $\mathbb{R}^{\vec{\mathcal{S}}}$ that are multiples of $G' = (b - 1, b, 0, 0, 0, 0)$. Assume $G' \in \operatorname{im} \mathbf{\Gamma}$, meaning there is a reward function $R' \in \mathbb{R}^{\vec{\mathcal{S}}}$ with $\mathbf{\Gamma} \cdot R' = G'$. We need to deduce from this a contradiction. The assumption means we obtain the following equations:

$(i)\ \ R'(S) + 3R'(T) = b - 1,$
$(ii)\ \ R'(S) + R'(A) + 2R'(T) = b,$
$(iii)\ \ R'(S) + R'(A) + R'(C) + R'(T) = 0,$
$(iv)\ \ R'(S) + R'(C) + 2R'(T) = 0,$
$(v)\ \ R'(S) + R'(B) + R'(C) + R'(T) = 0$
$(vi)\ \ R'(S) + R'(B) + 2R'(T) = 0$

(iii) and (v) together imply $R'(A) = R'(B)$; (iv) and (vi) together imply $R'(B) = R'(C)$; (v) and (vi) together imply $R'(C) = R'(T)$; so together, we have $R'(A) = R'(T)$. Thus, replacing $R'(A)$ in (ii) by $R'(T)$ and comparing (i) and (ii), we obtain $b - 1 = b$, a contradiction. Overall, this shows $\ker \mathbf{B} \cap \operatorname{im} \mathbf{\Gamma} = \{0\}$, and thus identifiability of the return function by Corollary D.4.

Now we investigate the case of unmodeled partial observability.

For demonstrating overjustification, assume deterministic transition dynamics in which every arrow in the diagram can be chosen by the policy. Also, assume $R(A) \ll 0$, $R(T) > 0$, $R(S) = 0$,

$R(B) = 0$, and $R(C) = 0$. Then the optimal policy chooses the state sequence $STTT$. However, this trajectory has low observation value since $G_{\text{obs}}(STTT) = (\mathbf{B} \cdot G)(SXXX) = bG(STTT) + (1 - b)G(SATT)$, which is low since $R(A) \ll 0$. $J_{\text{obs}}$ then selects for the suboptimal policies choosing $SBTT$ or $SCTT$, which is overjustified behavior that makes sure that the human does not think state $A$ was accessed.

For demonstrating deception, assume that $R(A) \gg 0$, $R(T) < 0$, $R(S) = R(B) = R(C) = 0$ and that the transition dynamics are such that when the policy *attempts* to transition from $S$ to $A$, it will sometimes transition to $B$, with all other transitions deterministic. In this case, the optimal behavior attempts to enter state $A$ since this has very high value. $J_{\text{obs}}$, however, will select for the policy that chooses $STTT$. This is deceptive behavior. □

