# OpenReview forum: "When Your AIs Deceive You: Challenges of Partial Observability in Reinforcement Learning from Human Feedback"
_NeurIPS.cc/2024/Conference — NeurIPS 2024 poster_

### Official Review · Reviewer_rT4E · 2024-07-07

**Soundness:** 2
**Presentation:** 2
**Contribution:** 2
**Rating:** 5
**Confidence:** 3

**Summary:**

The authors present a current problem (research gap): if humans give model answer feedback in a partially observable environment, such feedback may lead to deceptive inflation and overjustification. For example, if humans purely depend on the error reporting result, this will encourage the model to hide the error intentionally, leading to deceptive behavior.

The authors then introduce the formalization of partial-observable RLHF, and formalize  Deceptive Inflation and Overjustification. For example, turning human beliefs into a matrix B, and replace Return(states) with B$\cdot$ Return(observation). After that, authors define the concept of ambiguity, a linear subspace. If the distance between human's return function and true return function fall into such subspace, the feedback will be the same.

**Strengths:**

[1]The proposed research question is practical, interesting, and in an early-investigated stage.  The authors formalize this question for human feedback under partial observability.

[2] The authors provide several informative figures, which should be encouraged.

**Weaknesses:**

[1] No experimental results and metrics

One main contribution of this paper is the proposed research question and its formalization. Given the formalization, the reader would expect the authors to (1)explain the question within the formalization, (2)give an experiment to empirically reveal the question or justify the proposed solution, and (3)give metrics to judge the effectiveness of methods that aim to solve this question. And the paper only did (1).
Though this paper gives proof of specific propositions, it's not reasonable for a formalization lacking (2) or (3).

[2] Writing

Despite the clear figures, some logic does not match. The purposes stated ahead of sub-sections are lost,  for example, due to the lack of translation from proven Propositions to the answers to proposed questions.

**Questions:**

The paper presents a formalization of  Deceptive Inflation and Overjustification and mentions some modifications to current RLHF methods.   Even with proof, the result may not stand without experiments. Thus this paper will grow much stronger if it can be supported from experiments.

**Limitations:**

The authors admitted that some of the assumptions that this method is based on may not hold, thus need further improvement.

Beyond that, authors may consider creating some metrics for deceptive inflation and overjustification, and then justifying them. In this way, this work could serve as a stepping stone for the community to follow.

---

> ### Author Rebuttal · Authors · 2024-08-07
>
> Thank you for the detailed review of our work!
>
> > Given the formalization, the reader would expect the authors to (1)explain the question within the formalization, (2)give an experiment to empirically reveal the question or justify the proposed solution, and (3)give metrics to judge the effectiveness of methods that aim to solve this question. And the paper only did (1). Though this paper gives proof of specific propositions, it's not reasonable for a formalization lacking (2) or (3).
>
> Our work is indeed theoretical. For our general philosophy justifying this choice, we refer to [our global rebuttal](https://openreview.net/forum?id=XcbgkjWSJ7&noteId=dwv3iH9Ccw).
>
> With respect to empirical justifications of proposed solutions, we want to additionally highlight that in Section 6, we sketch the beginnings of a method that is *theoretically justified*. In particular, if $B$ is known, then in Proposition C.5 we prove that there is a differentiable loss function whose minimum contains only feedback-compatible reward functions, which are safe if the ambiguity $\ker(B \circ \Gamma)$ disappears.
>
> > Despite the clear figures, some logic does not match. The purposes stated ahead of sub-sections are lost, for example, due to the lack of translation from proven Propositions to the answers to proposed questions.
>
> Thank you for this comment. We will extend the first paragraph in Section 4 to foreshadow Proposition 4.1 in Section 4.1, which, so far, was unmentioned in the intro of Section 4. Our new intro paragraph to Sec. 4 will read:
>
> “*We now analyze failure modes of a naive application of RLHF from partial observations, both theoretically and with examples. In Proposition 4.1, we show that under partial observations, RLHF incentives policies that maximize what we call $J_\text{obs}$, a policy evaluation function that evaluates how good the state sequences “look to the human”. The resulting policies can show two distinct failure modes that we formally define and call deceptive inflation and overjustification. In Theorem 4.5 we prove that at least one of them is present for $J_\text{obs}$-maximizing policies. Later, in Sections 5 and 6, we will see that an adaptation of the usual RLHF process might sometimes be able to avoid these problems.*”
>
> Furthermore, we think that the start of Section 5 might be confusing in its current form, where we ask “Assuming the human’s partial observability is known, could one do better?”. This question is only directly addressed in Section 6. To improve the logic, we will make the following changes:
> We will merge Sections 5 and 6 (i.e., Section 6 becomes the new Section 5.3), and change the second paragraph of the intro to Section 5 as follows:
>
> “*We start in Section 5.1 by analyzing how much information the feedback process provides about the return function when the human’s choice model under partial observations is known precisely. We show that the feedback determines the correct return function up to an additive constant and a linear subspace we call the ambiguity (see Theorem 5.2). If the human had a return function that differed from the true return function by an element in the ambiguity, they would give the exact same feedback — such return functions are thus feedback-compatible. In Section 5.2, we show an example where the ambiguity vanishes, and another where it doesn’t, leading to feedback-compatible return functions that have optimal policies with high regret under the true return function. Finally, in Section 5.3 we explore how one could in theory use Theorem 5.2 as a starting point to design reward learning techniques that work under partial observability.*”
>
> Please let us know if you see further problems in the clarity of our writing, and we are happy to address them!
>
> > The authors admitted that some of the assumptions that this method is based on may not hold, thus need further improvement.
>
> It is true that some assumptions on the human model may not hold. Nevertheless, see [our answer to reviewer LhS3](https://openreview.net/forum?id=XcbgkjWSJ7&noteId=FWxGuq61x2): we think a more realistic feature-map formulation of the human model (replacing the belief $B$) would effortlessly connect with our formalization. Thus, we think there is great potential for our formalization to remain relevant even when taking into account more realistic human models. We will add a discussion on this viewpoint in the final version.
>
> Additionally, it is a priori our belief that the results in Section 4 are robust to more realistic human models, since they are about failure modes that we show to be present even when the human has excellent rational expected-value thinking.
>
> Please let us know about any remaining concerns, and we are happy to discuss them.

---

> > ### Comment · Reviewer_rT4E · 2024-08-12
> >
> > Thank you for your response and clarification! The Global Rebuttal mitigates my major concern, and I would raise my score to a positive one.

---

### Official Review · Reviewer_LhS3 · 2024-07-12

**Soundness:** 3
**Presentation:** 3
**Contribution:** 3
**Rating:** 7
**Confidence:** 3

**Summary:**

This work studies the impact of partial observability in RLHF. Two failure cases are defined: deceptive inflation and overjustification, and under them, concrete conditions are provided to impact the learned policy. Moreover, the ambiguity induced by the partial observability is further measured that the feedback determines the correct return function up to an additive constant and a linear subspace, which does not vanishes in many circumstances. Finally, it is recommended that recommended performing exploratory research on RLHF for cases when partial observability is unavoidable.

**Strengths:**

- This work targets at one important issue in the current RLHF learning pipeline that the human annotators may not observe the full information of the model's generation processes. Taking such scenarios in consideration is vital to keep the trained models trustworthy. I appreciate the authors' efforts in this direction.

- One theoretical formulation is proposed to model how human deals with partial observability (i.e., through the belief matrix). Under this formulation, detailed discussions are provided on what kinds of conditions would lead to impacted policies and how much ambiguity is introduced. The cases of Deceptive Inflation and Overjustification are well aligned with many practical scenarios in my mind. And, the discussed ambiguity is enlightening, also highlighting the need to treat partial observability carefully.

- The overall presentation is very clear, with key messages illustrated both rigorously through theorems and intuitively through examples/figures. I enjoyed reading this work.

**Weaknesses:**

I am overall satisfied by this work. The followings are a few points that would be interesting if further discussed.

- This work discusses the partial observability mostly in the theoretical sense, with a few hypothetical examples. It would be nice to see or verify the impact of partial observability in real experiments. I understand this would require a large amount of further efforts, while encouraging the authors to contain some discussions (e.g., on experimental designs) to enlighten future works.

- As also mentioned in the limitations listed in Section 7, the formulation of the belief matrix is a bit strong in my mind. I fully understand that it is adopted to deliver the message of treating partial observability carefully, while also encouraging the authors to include some discussions to improve such a formulation.

**Questions:**

I would love to hear the authors' opinions on the points listed in weaknesses.

**Limitations:**

Yes.

---

> ### Author Rebuttal · Authors · 2024-08-07
>
> Thank you for your detailed review!
>
> > This work discusses the partial observability mostly in the theoretical sense, with a few hypothetical examples. It would be nice to see or verify the impact of partial observability in real experiments.
>
> Our work is indeed theoretical. For our general philosophy, we refer to [our global rebuttal](https://openreview.net/forum?id=XcbgkjWSJ7&noteId=dwv3iH9Ccw).
>
> ## Feature count formulation of the human model
>
> > As also mentioned in the limitations listed in Section 7, the formulation of the belief matrix is a bit strong in my mind.
>
> This is indeed appears to be a strong assumption. We think further work is needed to evaluate the strengths and shortcomings of this formalization. We want to explain one implicit viewpoint of ours that we think means this formalization might reach quite far.
>  Namely, we think it is compatible with how humans intuitively form judgments over events.
>
> Concretely, if you are a human tasked with judging the quality of an observation sequence $\vec{o}$, what you will do is admittedly **not** to compute an explicit posterior belief $B(\vec{s} \mid \vec{o})$ over $\vec{s}$ – indeed, if the environment is very complex, it may already be impossible to even think about entire state sequences $\vec{s}$. More likely, you would think about the presence of certain features in the state sequence; e.g., “how often was there a bug”, or “has the software been installed”, or “how efficient is the code”. One would then implicitly assign a reward to each such feature and then add up the rewards.
>
> This feature viewpoint could a priori be compatible with our formulation, namely if the feature-based returns implicitly come from a belief over state sequences. This could work as follows: Assume there is a feature map $\phi(s)$ that maps high-dimensional states $s$ to low-dimensional feature vectors $f$. Assume the reward only depends on feature vectors: $R(s) = R’(\phi(s))$ for some function $R’$. We obtain:
>
> $$\sum_{\vec{s}} B(\vec{s} \mid \vec{o}) G(\vec{s}) = \sum_{\vec{s}} B(\vec{s} \mid \vec{o}) \sum_{t = 0}^{T} \gamma^t R(s_t) = \sum_{f} \left( \sum_{\vec{s}} B(\vec{s} \mid \vec{o}) \sum_{t = 0}^{T} \gamma^t \delta_f(\phi(s_t)) \right) R’(f) \eqqcolon \sum_{f} N(f \mid \vec{o}) R'(f),$$
>
> Where the outer sum runs over possible features, and where $\delta_f(\phi(s_t))$ evaluates to $1$ if and only if $\phi(s_t) = f$. In the last step, we denoted the big coefficient of $R’(f)$ by $N(f \mid \vec{o})$, which is the expected discounted feature-vector count of $f$ upon observing the whole sequence $\vec{o}$.
>
> Then instead of modeling the human as coming with a reward function $R$ and a belief matrix $B$, we have found an alternative model: we can model the human as coming with a feature-based reward function $R’$ and an expected feature-count $N(f \mid \vec{o})$. This highlights that the whole formalism could also be built upon a human model where the human “counts the presence of features”, as long as these feature counts come with consistency properties that allow them to be compatible with a belief over state sequences. We think this viewpoint makes it natural to extend our work to more realistic human models that effortlessly connect with our work.
>
> We will add this additional discussion to our paper to explain the reach of our belief matrix formulation.
>
> Please let us know if you have remaining concerns, and we are happy to address them.

---

> > ### Comment · Reviewer_LhS3 · 2024-08-12
> >
> > Thank you for the response! It helps me maintain a positive opinion on this work.

---

### Official Review · Reviewer_zgLN · 2024-07-12

**Soundness:** 3
**Presentation:** 3
**Contribution:** 3
**Rating:** 6
**Confidence:** 2

**Summary:**

The paper discusses the challenges that arise when human annotations for RLHF are based on partial observations. They formally define deceptive inflation and overjustification as failure cases caused by partial observation, and theoretically prove that standard RLHF is guaranteed to result in either or both of them. They further analyze how much information the feedback process provides about the return function assuming that human’s partial observability is known and accounted for.

**Strengths:**

- The research question proposed in this paper is of significant value. The authors provide a clear formal definition of deceptive inflation and overjustification by introducing policy evaluation function and over-/underestimation error.
- The paper provides detailed and solid proof over their claims, not only discussing the limitations of standard methods but also showing whether the model can perform better when the human’s partial observability is known. The authors also use many examples and counterexamples to analyze and explain the proposed claims.

**Weaknesses:**

- The proofs are based on assumptions of a specific MDP structure and a particular human belief function, which might not be easily generalized to realistic, complex environments.
- No empirical evidence is included in this paper, though real-world examples are given.

(I am not quite familiar with relevant topics and am unable to assess this part.)

**Questions:**

Refer to the above comments.

**Limitations:**

The paper discusses some limitations and future work. In addition to the aspects mentioned by the authors, I am also looking forward to seeing empirical evidences regarding this topic.

---

> ### Author Rebuttal · Authors · 2024-08-07
>
> Thank you for your review!
>
> > The proofs are based on assumptions of a specific MDP structure and a particular human belief function, which might not be easily generalized to realistic, complex environments.
>
> **It is not true that we assume a specific MDP structure.** Our work applies to any MDP, encompassing all reinforcement learning problems. **Our human belief function is also assumed to be general**: The only restriction is that it sums to 1 over all state-sequences, for a given observation sequence. Future work could think about relaxing the human belief function to allow for "feature maps" since humans cannot think about the whole environment state of complex environments. We detail this “feature map” idea [in our answer to reviewer LhS3](https://openreview.net/forum?id=XcbgkjWSJ7&noteId=FWxGuq61x2).
>
> > No empirical evidence is included in this paper, though real-world examples are given.
>
> Our work is indeed theoretical. For our general philosophy, we refer to [our global rebuttal](https://openreview.net/forum?id=XcbgkjWSJ7&noteId=dwv3iH9Ccw).
>
> Please let us know if you have remaining concerns. We will be happy to address them!

---

> > ### Comment · Reviewer_zgLN · 2024-08-12
> >
> > Thank you for your response and clarification! I would maintain the current score. Thanks!

---

### Official Review · Reviewer_v92A · 2024-07-13

**Soundness:** 3
**Presentation:** 3
**Contribution:** 2
**Rating:** 4
**Confidence:** 3

**Summary:**

The paper addresses the problem of accounting for humans having only partial observability of their environment when providing feedback. They outline two natural issues that can arise from such partial observability - deceptive inflation and overjustification - and provide examples of both. They then explore what is possible if the human's partial observability is fully known and accounted for, specifically asking how much about the return function can be recovered from the feedback. They show that in many realistic cases, there is an irreducible ambiguity (formalized as a vector space) in determining the return function. Finally, they propose ideas for combating these issues with a small proof-of-concept theorem backing their ideas.

**Strengths:**

1. The paper tackles a new facet of partial observability in human feedback - humans not observing the entire state.
2. They formalize the intuitive issue of deceptive inflation, and also introduce the counterintuitive but very plausible idea of overjustification.
3. Their theoretical framework is very clean and allows them to completely characterize the ambiguity in obtaining the return function.
4. The study of examples in the appendix is very comprehensive.

**Weaknesses:**

1. While the problems outlined are illustrated using hypothetical examples, I think the paper would greatly benefit from carefully designed experiments that unambiguously demonstrate the existence of such problems in practice.
2. The paper does not propose any concrete solutions to the problem. While this is great work exploring the problem, I feel that that does not rise to the level of a NeurIPS paper. This would make an excellent workshop paper that can grow into a full conference paper after adding some concrete attempts at solving this problem, either by presenting algorithms with new theoretical guarantees in this context, practical experiments showing a quantifiable improvement in meaninigful metrics associated to the problem, or both.
3. Minor and easy to rectify: The paper does not discuss other work in RLHF that deals with partial observability and heterogeneity (albeit in a subtly different context), for example [1, 2, 3].

All in all, this paper is a great start, but needs more work to become a full-fledged conference paper. I am willing to reconsider my score if either 1 or 2 are provided.

Refs:

1. Direct Preference Optimization With Unobserved Preference Heterogeneity. Chidambaram et al, 2024.

2. A Theoretical Framework for Partially Observed Reward-States in RLHF. Kausik et al, 2024.

3. RLHF from Heterogeneous Feedback via Personalization and Preference Aggregation. Park et al, 2024.

**Questions:**

1. Are there easy ways to force the ML model to"expose the underlying state a bit more" from time to time? A naive example is forcing verbosity in the case of the deceptive inflation example, but of course one needs to design a more general and sustainable fix.

**Limitations:**

Yes

---

> ### Author Rebuttal · Authors · 2024-08-07
>
> Thank you for your thorough review!
>
> >The paper would greatly benefit from carefully designed experiments that unambiguously demonstrate the existence of such problems in practice.
>
> Our work is theoretical and we advocate for judging it on a theoretical standard. We explain our viewpoint in more detail in our [global rebuttal](https://openreview.net/forum?id=XcbgkjWSJ7&noteId=dwv3iH9Ccw).
>
> > The paper does not propose any concrete solutions to the problem [...] adding some concrete attempts at solving this problem, either by presenting algorithms with **new theoretical guarantees** [emphasis ours] in this context, practical experiments showing a quantifiable improvement in meaninigful metrics associated to the problem, or both.
>
> We want to highlight that Section 6 presents exploratory ideas for practical solutions, and in particular, that the discussed Appendix C.3 with Proposition C.5 provides **a first theoretical guarantee**: If the belief matrix B is known explicitly, one can design a differentiable loss function whose minima are feedback-compatible. In particular, this proves that if the ambiguity $\text{ker}(\text{B} \circ \Gamma)$ vanishes, then the minima of the loss function are guaranteed to be safe, in the sense that they have return functions that differ up to an additive constant from the true return function. Thus, future work has a theoretical guide for designing practical algorithms, which requires carefully designing the type of partial observability so that the ambiguity is “benign”, and potentially specifying an approximation of the belief matrix B.
>
> > The paper does not discuss other work in RLHF that deals with partial observability and heterogeneity (albeit in a subtly different context), for example [1, 2, 3].
>
> Thank you for making us aware of these recent works. We will add the following paragraph to the related work:
>
> “*The literature also discusses other cases of partial observability. [1] and [3] deal with the situation that different human evaluators can have different unobserved preference types. In contrast, we assume a single human evaluator with fixed reward function, which can be motivated by cases where the human choices are guided by a behavior policy, similar to a [constitution](https://www.anthropic.com/news/claudes-constitution) or a [model spec](https://cdn.openai.com/spec/model-spec-2024-05-08.html). [2] assumes that the choices of the human evaluator depend on an unobserved reward-state with its own transition dynamics, similar to an emotional state in a real human. In contrast, we assume the human to be stateless.*”
>
> We also add paragraphs detailing related work on the modeling choices for human preference selection, truthful AI, and other works that deal with missing information. Please let us know if you would like to see these paragraphs, then we will promptly add them in a subsequent answer.
>
> > Are there easy ways to force the ML model to"expose the underlying state a bit more" from time to time? A naive example is forcing verbosity in the case of the deceptive inflation example, but of course one needs to design a more general and sustainable fix.
>
> We think about this in terms of trade-offs. *In principle*, the environment state could be fully observed by the human: We could just send all the information that the AI receives also to the human evaluators. The reason why we still think our work is necessary is that we expect it to increasingly become prohibitively expensive to show human evaluators all content; after all, human evaluators have limited time, and so if they always fully observe a state sequence, that trades off against the number of labels they can provide within a given timeframe.
>
> Thus, the question becomes how to *design* the partial observability in exactly such a way that despite limited information, the feedback-process still leads to correct reward functions being learned. Your idea to “Expose the state from time to time” (e.g. randomly) is interesting, and we are indeed interested in further research that deeply analyzes such settings.
>
> Please let us know if you have remaining concerns. We will be happy to discuss them.

---

> ### Comment · Reviewer_v92A · 2024-08-12
> **Response to rebuttal**
>
> While I appreciate the response and acknowledge that I am no stranger to an emphasis on theory, I continue to have qualms.
>
> I believe that there is a difference between a theoretical work that introduces a new model and theoretical work that identifies a problem with existing work. Since your work falls into the latter, it needs more than just hypothetical justification to demonstrate the severity and impact of the problem. The experiments demonstrating the existence of this problem need not be your own, but they need to exist and be referred to.
>
> An example I would like to give is the work of Alon and Yahav introducing oversquashing in GNNs. Their paper lacked substantive theory, but it empirically demonstrated the existence of the problem they were postulating. I think if we identify a potential problem in our minds, it is our duty to make sure that we haven’t made it up.
>
> Further, I apologize for not acknowledging the theoretical result they had. The reason it didn’t satisfy me was because assuming that the belief matrix is known is too strong an assumption. This made the result too weak for me to appreciate, despite it seeming technically non-trivial to show.
>
> If either of these two concerns were addressed (strength of theoretical result or experiments demonstrating problem), then the other issue would be less major to me. Since neither has been addressed, I am afraid I cannot raise my score.
>
> I must emphasize that this is an important problem and a paper with potential, it just needs to be strengthened on at least one of these ends. While I know that it can be tempting to work on exciting theory, in this case I implore you to design an experiment that can demonstrate the potential existence of this problem. Work like that of Alon and Yahav is in the unrelated domain of GNNs, but nevertheless it should help you understand the broad flavor of experiments that could help you convince readers.

---

> > ### Author Response · Authors · 2024-08-14
> > **A reference for empirical evidence of deceptive inflation**
> >
> > Thank you for continuing the discussion.
> >
> > > The experiments demonstrating the existence of this problem need not be your own, but they need to exist and be referred to. [...] I think if we identify a potential problem in our minds, it is our duty to make sure that we haven’t made it up.
> >
> > We think you raise valid points on the need for empirical validation of theoretical concerns. One early example for deceptive inflation is the robot hand from the blogpost accompanying an early RLHF paper [1]. Unfortunately, there are not many details on this example. Another very recent paper [2] provides more detailed evidence for deceptive inflation.
> >
> > In detail: In [2], Section 3.3, there is a setup for a quintessential task involving deceptive inflation: A human user asks the model a question, and if it answers truthfully, it will get some low reward since the answer is undesirable to the user. If it simply answers with the desirable answer, it will also get a low reward. But if it answers in a desirable way and modifies a file in an unobserved way to make that answer appear to be true, then it gets a high reward. Figure 2 in [2] shows that this behavior can show up in a zero-shot way, and without being prompted to do so, after being trained on other tasks. Figure 13 shows that this behavior is successfully reinforced into the model with outcome-based PPO (with rewards based on observations, not full states) even when there is additional “helpful, honest, harmless” training happening (which only slows down the manifestation of deceptively inflating behavior). Finally, Figure 2 shows that there is also a more serious deceptively inflating behavior that can appear zero-shot (namely reward-tampering), albeit at a much lower rate, and as explained in Section 3.4, the authors do not attempt to demonstrate that this behavior can be strengthened via RL.
> >
> > To be clear, since Section 3.3 in [2] does not work with trajectory comparisons but directly looks at rewarding some unobserved behavior that produces favorable observations, it is mainly evidence that “maximizing $G_\text{obs}$” (in our language) can strengthen deceptive behavior (analogous to our Theorem 4.5). What’s still missing is evidence that such behavior also gets reinforced when there is an earlier reward modeling phase using trajectory comparisons. In other words, this paper does not provide direct empirical evidence for our Proposition 4.1.
> >
> > Overall, we think it is laudable that the authors of [2] could show the zero-shot emergence and reinforcement of deceptive behavior under partial observability, and we think that was likely a significant challenge. For example, the authors write:
> > > *Models’ ability to exploit misspecified reward processes, and the ease with which they can represent a reward-seeking policy, will grow as models get more capable. Without countermeasures we should expect reward-seeking behavior to become more likely, but we are far from the point at which such models are an active risk.*
> >
> > We think that we should be on the lookout for opportunities to empirically show realistic, dangerous behavior with models of increasing capabilities.
> >
> > To be more concrete, we propose to discuss this work in the related work section with the following, more compressed paragraph:
> >
> > *Our paper mainly provides theoretical evidence of failure modes of RLHF under partial observations. [2] provides first empirical evidence of deceptive inflation: a model zero-shot generalizes from more benign behavior to deceiving a synthetic human with unobserved actions that modify a file. This behavior is then subsequently reinforced in an RL stage by a reward function that does not “observe” the file tampering. The paper also shows that – very rarely – their model can zero-shot generalize from less serious behavior to outright unobserved reward-tampering. We are unaware of work showing empirical evidence for our second failure mode, overjustification.*
> >
> > Additionally, we will mention in our conclusion that we welcome future work with further empirical investigations of failure modes under partial observability.
> >
> > Please let us know whether this alleviates some of your concerns regarding empirical evidence of our proposed failure modes.
> >
> >
> > [1] Dario Amodei et al., Learning from human preferences, https://openai.com/index/learning-from-human-preferences/, 2017
> >
> > [2] Carson Denison et al., [Sycophancy to Subterfuge: Investigating Reward-Tampering in Large Language Models](https://arxiv.org/abs/2406.10162v3), arxiv e-prints, 2024

---

### Author Rebuttal · Authors · 2024-08-07

# Global Rebuttal

In this global answer, we want to advocate for the inclusion of our purely theoretical work. The reviewers highlight the following (and more!) positive aspects of our work:

- *On the problem setting*: Our setting is “vital” (LhS3) and has “significant value” (zgLN).
- *On deceptive inflation/overjustification*: These are “well aligned with many practical scenarios” (LhS3).
- *On the ambiguity in the return function*: We “Completely characterize the ambiguity” (v92A), which is “enlightening” (LhS3).
- *On the theory*: “very clean” (v92A), with “detailed and solid proofs” (zgLN), and “rigorous [...] theorems” (LhS3).
- *On examples*: Our “many examples and counterexamples” (zgLN) are “very comprehensive” (v92A), and “intuitive” (LhS3).
- *On the presentation*: We include “several informative figures” (rT4E) and have an “overall presentation [that] is very clear” and that they “enjoyed to read” (LhS3).

On the other hand, all four reviewers highlight the lack of experiments and metrics to empirically validate the failure modes of deceptive inflation and overjustification. Reviewers v92A and rT4E furthermore highlight the lack of proposed (practical) solutions or metrics to judge progress toward a solution.[a] We overall agree that these are valuable goals to have, but consider them future work, for our work is meant to lay the conceptual and theoretical groundwork for studying RLHF under partial observability. Our goal was to clearly formalize a problem that has remained unformalized despite being informally known since the inception of RLHF through the [robot-hand](https://openai.com/index/learning-from-human-preferences/), and later the [ELK report by Christiano et al](https://docs.google.com/document/d/1WwsnJQstPq91_Yh-Ch2XRL8H_EpsnjrC1dwZXR37PC8/edit#heading=h.kkaua0hwmp1d).

There is a large history of published work at NeurIPS, ICML and ICLR that is purely theoretical and comes without practical solutions. We highlight some that are related to our work: In [1], the authors theoretically analyze AI alignment in a setting in which the human’s reward function can change over time. In [2], it is shown that without knowledge of the algorithm with which humans choose their actions, one cannot learn their preferences. In [3], a theoretical umbrella of prior reward-learning work is built. In [4], the authors analyze what happens when the optimized reward function misses attributes important to humans. And in [5], the whole alignment problem is studied in a purely conceptual, pre-theoretic way. All these works are theoretical or conceptual, come without empirical investigations, and without new solutions to the proposed problems. **But crucially, many of these works are well-known and have guided the intuitions of researchers about the problems to tackle in our field.**

Given that similar work has been published before, it is our opinion that our theoretical work should be judged based on the quality of the theory we provide. With the large positive sentiment about all aspects of our work that we quoted above (on the problem setting, deceptive inflation/overjustification, the ambiguity in return functions, theoretical quality, examples, and presentation), we believe that we meet the standard for NeurIPS. We are happy to discuss further in the discussion phase, including in our answers to the individual reviews.

[1] Micah Carroll et al., *AI Alignment with Changing and Influenceable Reward Functions*, ICML, 2024

[2] Stuart Armstrong, Sören Mindermann, *Occam's razor is insufficient to infer the preferences of irrational agents*, NeurIPS, 2018

[3] Hong Jun Jeon et al., *Reward-rational (implicit) choice: A unifying formalism for reward learning*, NeurIPS, 2020

[4] Simon Zhuang, Dylan Hadfield-Menell, *Consequences of Misaligned AI*, NeurIPS, 2020

[5] Richard Ngo et al., *The Alignment Problem from a Deep Learning Perspective*, ICLR, 2024

[a]: While we do not have practical solutions, we **do** have theoretical beginnings of a solution. In particular, if B is known, then in Proposition C.5 we prove that there is a differentiable loss function whose minima contain only feedback-compatible reward functions, which are safe if the ambiguity $\text{ker}(\mathrm{B} \circ \Gamma)$ disappears. This is discussed in Section 6.

---

### Decision · Program_Chairs · 2024-09-25

**Decision:**

Accept (poster)

**Comment:**

This paper raised and formulated an interesting question in RLHF: what if the human feedback is only based only on partial observations. Two failure cases were defined and formalized, corresponding to "deceptive inflation" and "overjustification". It then analyzes how much information the feedback process provides about the return function, and under what cases human’s feedback can determine the return function. I appreciate the study of this new topic, which has been relatively overlooked in RLHF studies, and is important. The overall presentation is also clear. On the other hand, I also agree with some common concerns regarding the lack of experimental evidence of the issue: how serious it is, or does it really occur, in real-world applications of RLHF with real datasets. The paper would have been stronger if it started with such experimental observations, and the current version is a bit more conceptual (which in fact, can be a very good positional paper in my opinion). Given all the considerations, this is really a borderline paper, and the authors may benefit from the reviews this round. I hope the authors can incorporate the feedback in preparing the next version of the paper.

---

> ### Public Comment · ~Leon_Lang1 · 2024-12-01
>
> Thank you for the trust in our work! For the camera-ready version, we included simple empirical validations of our concerns in the toy environments that we also study conceptually. We hope that makes our claims more convincing to some readers.